# The organisation of subglacial drainage during the demise of the Finnish Lake District Ice-Lobe

Adam J. Hepburn[1,2*], Christine F. Dow[1], Antti Ojala[3], Joni Mäkinen[3], Elina Ahokangas[3], Jussi Hovikoski[4], Jukka-Pekka Palmu[4], and Kari Kajuutti[3]

[1]Department of Geography and Environmental Management, University of Waterloo, Waterloo, ON, Canada
[2]European Space Astronomy Centre, European Space Agency, Madrid, Spain
[3]Department of Geography and Geology, University of Turku, Turku, Finland
[4]Geological Survey of Finland, Espoo, Finland

**Correspondence:** Adam J. Hepburn, now at Aberystwyth University, Aberystwyth, Wales, UK (adam.hepburn@aber.ac.uk)

**Abstract.** Unknown basal characteristics limit our ability to simulate the subglacial hydrology of rapidly thinning contemporary ice sheets. Subglacial water is typically conceptualised as being routed through either distributed, inefficient, and high pressure systems, or channelised, efficient, and lower pressure systems, transitioning between the two as a function of discharge. Understanding the spatiotemporal transition in drainage modes is critical to modelling future ice mass loss. Sediment-based landforms generated beneath Pleistocene ice sheets, together with detailed digital elevation models, offer a valuable means of testing basal hydrology models, which describe the flow and dynamics of water in the subglacial system. However, previous work using geomorphological techniques together with models of subglacial hydrology has concentrated on landforms relating to channelised drainage (e.g., eskers) while using inherently channelised models, which are unable to capture transitions in drainage state. Landscapes relating to the distributed drainage system, and the hypothesised transitional zone of drainage between distributed and channelised drainage modes have therefore been largely ignored. To address this, we use the Glacier Drainage System model (GlaDS) to compare modelling output against predictions regarding the genesis conditions associated with 'murtoos', a distinctive triangular landform found throughout Finland and Sweden. Murtoos are hypothesised to form 40–60 km from the former Fennoscandian Ice Sheet margin at the onset of channelised drainage in a 'semi-distributed' system, in small cavities where water pressure is equal to or exceeds ice overburden pressure. Concentrating within a specific ice lobe of the former Fennoscandian Ice Sheet and using digital elevation models with a simulated former ice surface geometry, we forced GlaDS with transient surface melt and explored the sensitivity of our model outcomes to parameter decisions such as the system conductivity and bed topography. Our model outputs support many of the predictions for murtoo origin, including the location of water pressure equal to ice overburden, the onset of channelised drainage, the transition in drainage modes, and the predicted cavity size. Modelled channels also closely match the general spacing, direction and complexity of eskers and mapped assemblages of features related to subglacial drainage in 'meltwater routes'. Further, these conclusions are largely robust to a range of parameter decisions. Our results demonstrate that examining palaeo basal topography alongside subglacial hydrology model outputs holds promise for the mutually beneficial analyses of palaeo and contemporary ice sheets to assess the controls of hydrology on ice dynamics and subglacial landform evolution.

# 1 Introduction

Climatic warming is promoting more widespread and prolonged surface melting on both the Greenland and Antarctic Ice Sheets (van den Broeke et al., 2023). This surface meltwater is routed to the bed of ice through crevasses and moulins where it is supplemented by meltwater generated at the base of the ice through geothermal heat or friction (Davison et al., 2019). Meltwater delivered to the bed of ice sheets exerts a strong, but complex and non-linear control on ice flow and mass loss (Schoof, 2010; Wallis et al., 2023). In spring, at the onset of the melt season, individual glaciers in Greenland and the Antarctic

Peninsula have been observed to accelerate by between 10–300% relative to their winter velocity, with this signal extending tens of kilometres inland (Joughin et al., 2008; Bartholomew et al., 2012; Sole et al., 2011, 2013; Moon et al., 2014; Tuckett et al., 2019; Wallis et al., 2023). Such speedups result in enhanced mass loss into the ocean from increased run-off and iceberg calving, contributing to an increased rate of sea-level rise.

The changing configuration of the basal hydrological system beneath ice sheets throughout the melt season is primarily

responsible for modulating the response of ice flow to meltwater input (Schoof, 2010). In the winter, when meltwater production is limited, low volumes of water flow through an inefficient, highly-pressurised, *distributed* system consisting of thin films (Weertman, 1972), linked-cavities (Kamb, 1987), or as Darcian flow through a porous medium (Boulton and Jones, 1979). However, at the onset of the melt season, sudden, high-volume meltwater inputs can quickly overwhelm such a distributed system, raising water pressure over large areas of the glacier bed. Where water pressure is equal to or exceeds the pressure

of ice overburden, the overlying ice is hydraulically lifted, reducing the frictional resistance to ice motion and enhancing velocity (Schoof, 2010). However, beyond a critical discharge threshold (Schoof, 2010) sustained periods of high discharge and turbulent water flow promote channelisation, with wall-melt, and subglacial erosion, forming Nye channels into the underlying substrate (Nye, 1972) or Röthlisberger channels (R-Channels, Röthlisberger, 1972) and Hooke channels (Hooke, 1989) within overlying ice. In the channelised system low water pressure promotes steep pressure gradients which divert water flow away

from the distributed system towards channels, lowering water pressure and frictional resistance to flow over wide areas of the bed, dampening the velocity response to meltwater (Iken and Bindschadler, 1986; Iverson et al., 1999; Schoof, 2005). Within days velocity may slow below previous levels (Vijay et al., 2021) even as meltwater volume increases through the subsequent melt season.

An accurate representation of basal hydrology in models, and in particular the transition between distributed and channelised

drainage modes, is therefore critical in efforts to predict the rate and timing of ice sheet mass loss (Nienow et al., 2017). Models of basal hydrology, including those capable of resolving the transition between distributed and channelised drainage, have been widely used to investigate subglacial drainage beneath contemporary ice sheets (e.g., Schoof, 2005, 2010; Banwell et al., 2013; Werder et al., 2013; Flowers, 2018; Indrigo et al., 2021; Dow et al., 2022). However, despite recent work in both one and two-way coupling of these basal-hydrology models to ice flow models (e.g., Cook et al., 2019, 2020, 2022; Ehrenfeucht

et al., 2023), detailed treatment of basal hydrology is not yet widely integrated into ice sheet system models and predicting the rate and timing of glacial mass loss remains difficult (Dow, 2023). In part, this difficulty arises because the response of ice flow to melt depends on not only the volume, timing and duration of melt, but also, critically, on bed characteristics

including basal topography, underlying geology, and the hydraulic properties of the subsurface material (Chu, 2014). However, beneath contemporary ice sheets our knowledge of basal topography is limited to spatial resolutions in the order of $10^2$ m (e.g., Morlighem et al., 2017, 2020) and direct observations of hydraulic connectivity are sparse, especially at the ice sheet scale (Greenwood et al., 2016). In numerical models, given the absence of more detailed information, such characteristics and the processes which they govern are often reduced to parameterisations, or simplifications, of what is likely a more complex reality (e.g., Schoof, 2010; Werder et al., 2013; Flowers, 2018). In order to faithfully model the basal hydrology of contemporary ice sheets and reliably bound their future mass loss, it is important that we are to utilise all available sources of data to evaluate the efficacy of basal hydrology models (Dow et al., 2020; Doyle et al., 2022; Hager et al., 2022; McArthur et al., 2023).

Detailed geomorphological mapping across Europe and North America has revealed a complex record of glaciofluvial landforms (e.g., Figure 1B–E), which preserve information about the basal hydrology of Pleistocene ice sheets during their retreat since the last glacial maximum (e.g., Clark and Walder, 1994; Cofaigh, 1996; Rampton, 2000; Utting et al., 2009; Ojala et al., 2019; Coughlan et al., 2020; Dewald et al., 2021, 2022). Comparing the spatial distribution, hypothesised genesis, and geomorphology of these glaciofluvial landforms against the predictions of basal hydrology models may provide a crucial test of basal hydrology model ability not possible beneath contemporary ice sheets. However, this glaciofluvial landform record is a complex and likely time-integrated record dominated by evidence of channelised drainage, including eskers (e.g., Figure 1D), and tunnel valleys (e.g., Brennand, 2000; Storrar et al., 2014; Storrar and Livingstone, 2017; Lewington et al., 2019; Kirkham et al., 2022). By comparison, the distributed drainage system is less commonly described in landform records (Greenwood et al., 2016), although drumlins and ribbed moraine formations are thought to represent flow instabilities associated with the high water pressures within a distributed system (Chapwanya et al., 2011; Fowler, 2010; Fowler and Chapwanya, 2014; Stokes et al., 2013). Further, the ubiquity and scale of the channelised drainage record—Pleistocene eskers may be continuously traceable over tens to hundreds of kilometres (Storrar and Livingstone, 2017)—is largely without comparison in the contemporary record and similarly large channelised systems have only been described in isolated regions of the Antarctic (Dow et al., 2022). This scale mismatch, a paucity of information about distributed drainage, and the time-integrated nature of the palaeo record (Cofaigh, 1996; Greenwood et al., 2016), presents a challenge when seeking to evaluate the ability of current models to resolve the basal hydrology of palaeo ice sheets.

As a result, existing work using basal hydrology models in the palaeo setting has largely concentrated on channelised drainage, either over large areas of former ice sheet beds, or on the process of esker/tunnel valley formation at the individual landform scale, neglecting processes associated with distributed drainage and the transition between drainage modes. To investigate channelised drainage, previous work has made simplifying assumptions about water pressure, either prescribing a fixed water pressure at or near overburden pressure everywhere resulting in inherently channelised models (e.g., Livingstone et al., 2013a, b, 2015; Karlsson and Dahl-Jensen, 2015; Shackleton et al., 2018; Kirkham et al., 2022) or assuming channels form where water pressure is equal to ice overburden pressure (e.g., Boulton et al., 2007a, b, 2009). These models are unable to capture dynamic changes between distributed and channelised drainage, instead they are intended to represent long-term, inter-aunnual, 'steady-state' conditions (Banwell et al., 2013) applicable over millenial timescales—extremely relevant timescales

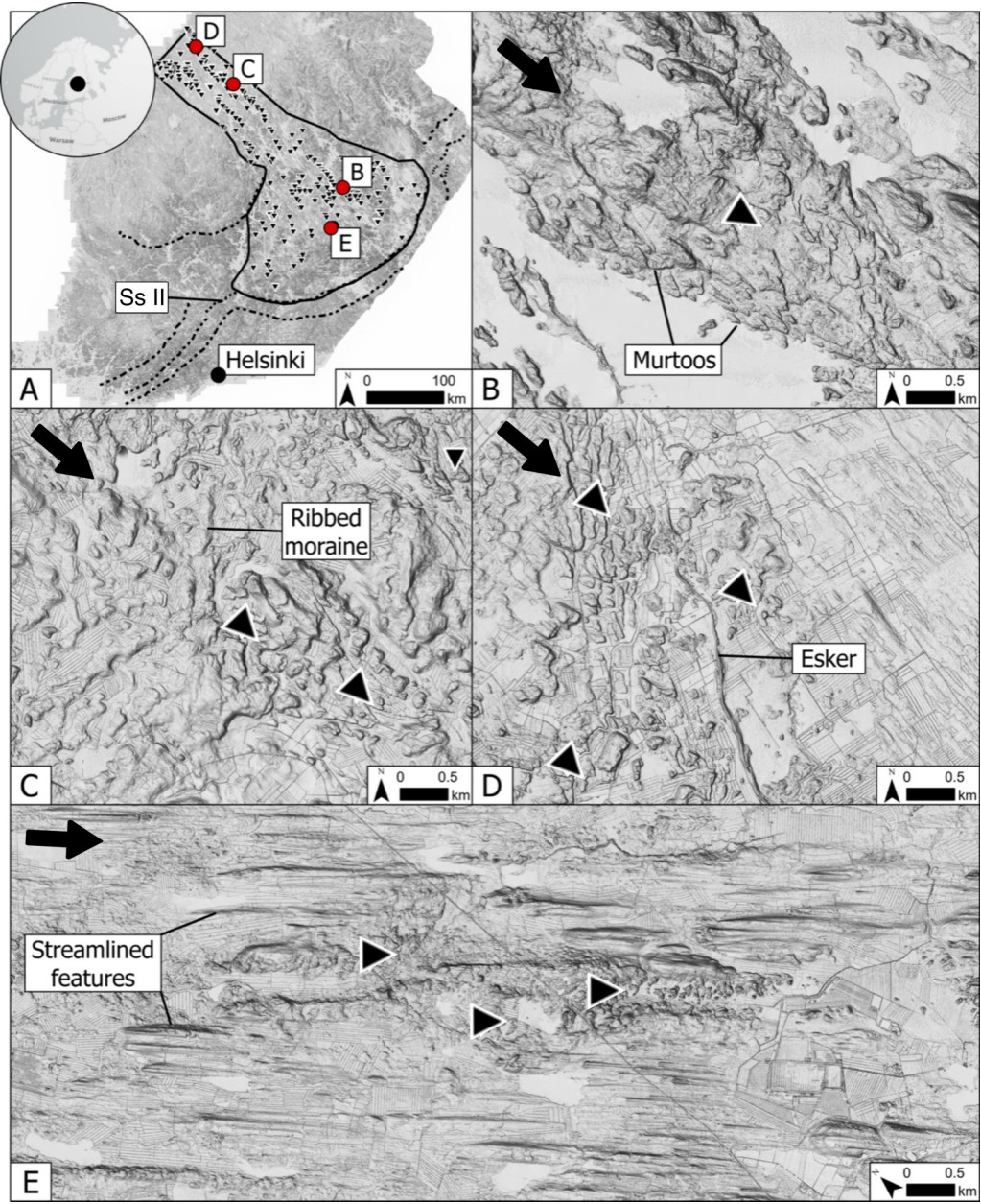

**Figure 1.** The study area. **A**) The extent of the GlaDS model domain (solid line) in the Finnish Lake District Ice Lobe (FLDIL) and the Salpausselkäs (dashed line). Salpausselkä II (Ss II) marks the 12 cal. ka ice extent. Murtoo fields identified by Ahokangas et al. (2021) within the FLDIL are shown as inverted triangles. **B**) A murtoo field. Adjacent to the murtoos, a large esker is visible in the lake. **C**) Murtoo fields amongst ribbed moraines in the north of the FLDIL. **D**) An abrupt downstream transition from murtoo fields to a large esker. Additional murtoo fields are found directly adjacent to the esker. **E**) Three murtoo fields amongst streamlined terrain within at the centre of the FLDIL. All panels show a multi-directional oblique weighted hillshade based on 2 m LiDAR data (see Ahokangas et al., 2021, for details). In panels B–E, black arrows in the upper left corner of each panel indicate the approximate ice flow direction and the inverted triangles are aligned with the orientation of murtoo fields.

when investigating large, time-transgressive features such as eskers and tunnel valleys (Clark and Walder, 1994; Clark et al., 2000; Mäkinen, 2003; Kirkham et al., 2022), which lack a discrete age-control but were likely formed over millennia.

More complex, 1D process-mechanism models have also been used to investigate esker genesis (e.g., Beaud et al., 2018; Hewitt and Creyts, 2019), however, these idealised models also typically lack a distributed component. Further, without including a representation of any specific bed topography or ice geometry, these 1D numerical process-mechanism models are difficult to compare directly to mapped geomorphology. Hewitt (2011) did include modelled drainage mode transitions by describing a continuum formulation of the distributed system coupled to a single channel which was allowed to evolve as a function of discharge. The length of the channel and its influence on the surrounding idealised glacial bed was used to determine channel length and spacing scaling relationships, finding good agreement with channel spacing predicted by Boulton et al. (2009) beneath the Fennoscandian Ice Sheet (FIS). However, in representing a single channel with fixed forcing, this work stopped short of directly comparing model outputs to realistic terrain data from palaeo beds. Glaciated Pleistocene terrains may provide valuable insight into subglacial hydrological processes (Greenwood et al., 2016), including those variable at the sub-annual scale and across the distributed–channelised transition, changes in which are known to be critically important in understanding contemporary ice mass loss during periods of rapid climate change (Dow, 2023). Models of basal hydrology capable of resolving distributed and channelised drainage are widely used to interrogate the dynamical response of contemporary glaciers to increasing melt (e.g., Werder et al., 2013; Sommers et al., 2018, 2022; Ehrenfeucht et al., 2023), yet despite the critical need to evaluate and improve these models (Dow, 2023) and the potential glaciated terrains hold for doing so, no work has applied a basal hydrology model capable of resolving distributed and channelised drainage to the palaeo setting.

In this paper, we present the first application of the Glacier Drainage System model (GlaDS Werder et al., 2013), a basal hydrology model capable of resolving transition between channelised and distributed drainage in the palaeo setting. To do so, we specifically focus on a recently described landform termed 'murtoos' (singular: murtoo, Mäkinen et al., 2017; Ojala et al., 2019)—small (30–100 m in width/length), low relief ($\sim$5 m high) features with a distinctive, broadly triangular morphology (See Figure 1, & Ojala et al., 2019; Ahokangas et al., 2021; Peterson Becher and Johnson, 2021; Ojala et al., 2021; Vérité et al., 2022; Van Boeckel et al., 2022). Murtoos were initially identified from LiDAR data in Finland (Mäkinen et al., 2017) and Sweden (Peterson Becher et al., 2017), and have since been extensively mapped throughout terrain formerly occupied by the FIS (Ojala et al., 2019; Ahokangas et al., 2021). Murtoos are hypothesised to form beneath warm-based ice during the rapid retreat of the FIS since the Younger Dryas (12.7 to 11.7 cal. ka), during which large volumes of meltwater were delivered to the bed in a glacial setting similar to contemporary Greenland (Ojala et al., 2019). Murtoo morphometry (Mäkinen et al., 2017; Ojala et al., 2021), their sedimentological architecture (Peterson Becher and Johnson, 2021; Hovikoski et al., 2023; Mäkinen et al., 2023), and close spatial association with eskers, ribbed tracts, and putative subglacial lakes (Ojala et al., 2021; Ahokangas et al., 2021; Vérité et al., 2022; Mäkinen et al., 2023) suggests that murtoo formation occurs within broad and low conduits, subject to repeated short pulses of meltwater against a backdrop of increasing water discharge throughout a melt season, at water pressures close to or exceeding ice overburden pressure, and with short sediment transport distances, such as might be found at the spatial onset of channelisation in a 'semi-distributed' transitional drainage system (Hovikoski et al., 2023).

Murtoos are therefore unique glaciofluvial landforms, and their short formation time, small size, and apparent location at the spatial onset of channelisation make murtoos potentially important components of the subglacial system. Murtoos are therefore ideal geomorphic targets against which basal hydrology models can be evaluated. Accordingly, in this paper, our aims are to model the subglacial hydrological conditions at the end of the Younger Dryas in order to:

- Compare the subglacial hydrological conditions proposed for murtoo genesis and their associated landforms against model outputs from GlaDS.

- Sensitivity test GlaDS across a range of possible parameter values to explore the influence of these parameters on our outcomes in order to evaluate the potential of such models to be used to interrogate palaeo-hydrological systems more broadly and in turn motivate future work in this area.

## 2 Study area and the significance of murtoos for basal hydrology

Our study area (Figure 1A) is the Finnish Lake District Ice-Lobe (FLDIL) province within southern Finland described by Putkinen et al. (2017) and Palmu et al. (2021), the FLDIL contains a high density of murtoo fields (Figure 1A, Ahokangas et al., 2021). The FLDIL is one ice lobe amongst several which comprised the eastern margin of the Fennoscandian Ice Sheet (FIS) at its Younger Dryas (12.7 to 11.7 cal. ka) extent, and the FLDIL encompasses an area of ~57,600 km$^2$, with a main
trunk upstream of a lobate expansion. The bedrock within the FLDIL and the wider FIS is predominantly crystalline bedrock, dominated by Precambrian schists, gneisses, and granitoids (Lehtinen et al., 2005) with a thin Quaternary overburden (Lunkka et al., 2021). The distribution of esker systems, ice-marginal complexes, streamlined bedforms, and moraines in south-central Finland suggests that the continental ice sheet at this time likely consisted of relatively rapidly flowing ice-lobe provinces, such as the FLDIL, interspersed with passive interlobate regions (Punkari, 1980; Salonen, 1986; Punkari, 1997; Boulton et al., 2001;
Lunkka et al., 2004; Johansson et al., 2005; Putkinen et al., 2017; Palmu et al., 2021).

The lobate portion of the FLDIL is particularly well demarcated at its distal margin by the first and second Salpausselkäs (Figure 1A)—large ice-marginal complexes, which mark the Younger Dryas extent of the FIS in the region (Donner, 2010; Lunkka et al., 2021). Our model domain is bound at its lateral margin by the younger, second Salpausselkä, which is the most recent ice-marginal feature in the region and marks the FLDIL extent at ~12 cal. ka (Putkinen et al., 2017). Shoreline data
indicates that the second Salpausselkä terminated in a shallow water body ranging in depth from <5 m to ~50 m (Lunkka and Erikkilä, 2012). The high-density of drumlins, mega-scale glacial lineations (MSGL), eskers and hummocky moraines (Figure 1C–E) within the FLDIL, their consistent orientation along a principal northwest–southeast axis, and the absence of ice-marginal features behind the second Salpausselkä suggest there was limited re-organisation of the major ice flow pathways during ice sheet retreat after ~12 cal. ka (Putkinen et al., 2017). Instead, the FIS is likely to have collapsed continuously and
rapidly following the Younger Dryas (Kleman et al., 1997), retreating northwest towards Norway and gone by 9–10 cal. ka (Hughes et al., 2016; Stroeven et al., 2016; Regnéll et al., 2019). The speed of the retreat, together with the complex and dense assemblage of glaciofluvial landforms (e.g., Palmu et al., 2021; Dewald et al., 2021), suggest that during deglaciation, the

FIS was characterised by high and spatially extensive atmospheric-driven surface melting delivered to the bed, accompanied by calving into the Baltic Sea Basin (Greenwood et al., 2017; Patton et al., 2017; Boswell et al., 2019). Conditions within the FLDIL, and the FIS more broadly, were likely comparable to conditions prevalent in land- or shallow-water terminating portions of the Greenland Ice Sheet today (Greenwood et al., 2016; Ojala et al., 2019).

Regional mapping across Finland has demonstrated a preferential clustering of murtoos in fields (e.g., Figure 1B) along meltwater routes—integrated assemblages of multiple landforms associated with subglacial meltwater (Lewington et al., 2020; Ahokangas et al., 2021; Dewald et al., 2022)—which are in turn concentrated in faster flowing, *warm-based* sectors of the FIS including the FLDIL (Ahokangas et al., 2021; Palmu et al., 2021; Dewald et al., 2022). Murtoos are absent within more passive, *cold-based* regions of the FIS (Ahokangas et al., 2021). Topographically, murtoos are associated with subglacial lake basins, bedrock depressions, and the lee side of some large bedrock protrusions (Ojala et al., 2021; Ahokangas et al., 2021). Meltwater routes containing murtoos, or 'murtoo routes' herein (Ahokangas et al., 2021) are often adjacent to or downstream of drumlin fields or ribbed moraines (e.g., Vérité et al., 2022, and Figure 1C), and murtoo routes may be located upstream of, and appear to transition into, eskers (Figure 1D, Ahokangas et al., 2021). The close association of landforms relating to channelised drainage with murtoo routes may therefore also mark the spatial and/or temporal transition from distributed to channelised drainage within a 'semi-distributed' drainage system, formed during the rapid deglaciation of the FIS (Ojala et al., 2019, 2021; Ahokangas et al., 2021; Peterson Becher and Johnson, 2021; Ojala et al., 2022; Vérité et al., 2022; Mäkinen et al., 2023).

Within our specific FLDIL study area, murtoo distribution is representative of their distribution across the wider FIS. In the upstream trunk, murtoo fields occur amongst ribbed and hummocky moraines (Figure 1C) in two longitudinal bands, each bounded by a dense assemblage of streamlined forms. In the northeastern longitudinal bands, eskers are particularly clearly associated with murtoo routes (Figure 1D). Downstream, where the FLDIL broadens into a lobe, murtoo distribution is more fragmented with less clustering evident. Murtoos are sparse in the centre of the ice lobe, however, the area's thin sediment cover (Figure A1) may limit the material for murtoo formation, and the high density of water bodies may act to mask existing murtoo fields (Ahokangas et al., 2021). Crucially, in the FLDIL, as elsewhere, murtoo routes have a characteristic distribution and are rarely found closer than 40–60 km from the second Salpausselkä margin at $\sim$12 cal. ka (Mäkinen et al., 2023), aligning well with the apparent limits of channelisation to within 50 km of the ice sheet margin in contemporary Greenland (Chandler et al., 2013, 2021; Dow et al., 2015).

Murtoo excavation and field study indicates that individual murtoos are composite landforms arranged parallel to ice flow, comprising a main body that is primarily depositional together with lateral margins and a lee-side head that are erosional features (Mäkinen et al., 2017, 2023; Hovikoski et al., 2023). Internal excavations within individual murtoos (e.g., Peterson Becher and Johnson, 2021; Mäkinen et al., 2023; Hovikoski et al., 2023) have revealed that murtoos consist of a core unit containing sorted sediments, overlain by a main body unit (referred to as Unit 2 by Mäkinen et al., 2023) that i) distally is comprised of alternating facies of heterogeneous diamicton, with strong fabrics interbedded with sorted gravelly and sandy sediment (Mäkinen et al., 2023) and ii) proximally is comprised of alternating sequences of glaciofluvial deposits, with current ripples (formed in low discharge, lower flow regimes) giving way to transitional cross-bedding (transitional flow regimes), and an-

tidunal sinsuoidal lamination (formed in higher discharge, upper flow regimes; Hovikoski et al., 2023). The murtoo body is, in turn, overlain by a mantling deposit heavily modified by soil forming processes, but nonetheless exhibiting weakly stratified diamicton and gravel beds as well as large boulders deposited at the ice-bed interface (Mäkinen et al., 2023). The sequence of murtoo formation is interpreted as follows (from Hovikoski et al., 2023);

1. In the first stage of murtoo formation, at the end of meltwater pulses the sorted sediment dominated core develops following sediment deposition within a rapidly enlarging broad and shallow subglacial conduit—possibly associated with pre-existing till ridges or cavities in the lee-side of protrusions. Sediment within this core evidences at least partial ice contact and periodic deformation by ice, and is superposed over existing, meltwater route deposits.

2. With the onset of spring melt, pulses of water deposit the murtoo body within an increasingly large conduit. As each pulse increases in discharge and then wanes they promote the deposition of sand lenses, sinusoidally stratified sand, and poorly-sorted gravel, with silt commonly draping ripple-scale features. In this phase of formation, cobbles are the largest clast size, which places an upper limit on water depth of $\sim 25\,\mathrm{cm}$ (Hovikoski et al., 2023).

3. As the melt season continues through summer, an increasingly enlarged pond forms in response to higher discharge. In turn, the increasing grain size indicates higher water velocity and sediments on the upper slope appear consistent with high velocity, upper-flow-regime deposits and the boulder size-distribution suggest a maximum flow space of $1\,\mathrm{m}$.

4. The development of this enlarged cavity/pond and subsequent water pressure drop encourages localised creep closure at the broadest part of the murtoo, evidenced by a disappearance of sorted sediment, and in some murtoos this is succeeded by compacted interbedded diamicton—indicating ice-bed recoupling. Meanwhile, closer to the margins of the murtoo body, meltwater flow continued and forced to pass an enclosed space, is routed obliquely towards the tip, forming boulder-rich proto-channels. These deposits indicate that the ice-bed recoupling at the broadest part of the murtoo coincided with intense and increasingly erosional channelised flow at the murtoo margins. The final stage of murtoo development is commonly represented by the development of boulder-rich marginal channels that finalise the triangular shape of the murtoos (Peterson Becher and Johnson, 2021).

5. Finally, murtoo deposition is abruptly terminated and marginal channels are abandoned. The final sedimentation within these marginal channels is characterised by suspension settling and laminated muds, indicating that the depositional space (0.6–0.8 m) remained open and water filled but no longer hydraulically connected to the wider meltwater system (Ojala et al., 2022; Hovikoski et al., 2023).

The sequence of murtoo deposits charts an overall increase in meltwater discharge throughout the melt season followed by an abrupt termination (Table 1), possibly within the same year (Hovikoski et al., 2023). Against this backdrop, the alternating sequences of glaciofluvial deposits in the main body of murtoos suggests that the system was also subject to repeated pulses of meltwater and rapid changes in flow regime, marking the rerouting and periodic isolation of cavities within a developing, semi-distributed drainage system over a single melt season or during rapid reorganisation associated with autogenic changes

within the wider meltwater system (Hovikoski et al., 2023; Mäkinen et al., 2023). The spatial distribution and sedimentological architecture of murtoos provides a testable set of predictions against which a basal hydrology model can be tested, including the location of a persistent area of high water pressure, the evolution of discharge through the year, and the onset of channelised drainage.

**Table 1.** Murtoo developmental stages (see Figure 10 in Hovikoski et al., 2023), their sedimentological signature, and anticipated model outcomes

| Murtoo developmental stage | Sedimentological evidence and interpretation | Expected model outcomes |
| --- | --- | --- |
| 1 | Sorted sediment core within rapidly enlarging cavity, partial ice contact | Sharp increases in $q_s$, increase in $overburden_\%$ |
| 2 | Onset of spring melt, cavity continuing to enlarge, with deposition of sinusoidal stratification and cobbles | Peak in $overburden_\%$, continued increase in $Q_c$ |
| 3 | Increasing grain size indicates high water velocity and boulder deposition indicates maximum cavity size | Peak in $V_W$, drop in $overbuden_\%$, peak in $Q_c$ approaching $1\,\mathrm{m^3\,s^{-1}}$ |
| 4 | Enlarged cavity leads to water pressure drop and ice-bed recoupling | Continued drop in $overbuden_\%$ and $Q_c$ |
| 5 | Abrupt termination of discharge with appearance of laminated mud | $overburden_\%$ approaching winter values |

## 3 Methods

To model the basal hydrology of the FLDIL, and compare this to the murtoo developmental stages, we used the Ice-sheet and Sea-Level System Model (ISSM, Larour et al., 2012, Revision 27448) and the implementation of the GlaDS model (Werder et al., 2013) contained therein. We first generated an input ice geometry by depressing a contemporary reanalysis temperature and precipitation dataset to approximate conditions $\sim 12\,\mathrm{cal.\,ka}$. Then, using GlaDS parameterised by this input ice geometry and a modified digital elevation model (DEM) of the region (see Section 3.1.1), we explored the evolution of basal hydrology beneath the FLDIL through time. A detailed description is provided below and model parameter values are given in Table 2.

### 3.1 Model description

The GlaDS model (described in full in Werder et al., 2013) is a 2D finite element model building upon earlier work (see Schoof, 2010; Hewitt, 2011) which has been widely applied to contemporary ice sheets in Greenland (e.g., Dow et al., 2018a; Cook et al., 2020, 2022; Ehrenfeucht et al., 2023) and Antarctica (e.g., Dow et al., 2018b, 2020; Indrigo et al., 2021; Dow et al., 2022; McArthur et al., 2023) as well as glaciers in Svalbard (e.g., Scholzen et al., 2021). The GlaDS model operates on an unstructured mesh and includes a model of distributed flow through linked cavities (Hewitt, 2011) represented by a continuous 'sheet' of water with variable thickness at mesh elements, and channelised flow—describing uniform, semi-circular Röthlisberger

**Table 2.** List of input values for GlaDS, values highlighted in bold indicate those used for sensitivity testing and a range of values is provided. Note, in all instances *sheet* refers to the subglacial drainage system.

| Symbol | Description | Default value | Tested range | Units |
|---|---|---|---|---|
| $\rho_i$ | ice density | 918 | | $\text{kg m}^3$ |
| $p_w$ | water density | 1000 | | $\text{kg m}^3$ |
| $g$ | gravitational acceleration | 9.81 | | $\text{m s}^{-2}$ |
| $n$ | Glen's flow law exponent | 3 | | |
| $a$ | basal friction coefficient | 0–120 | | $(\text{Pa a}^{-1})^{1/2}$ |
| $A$ | rate factor | $1.7\times10^{-24}$ | | $\text{s}^{-1}\,\text{Pa}^{-3}$ |
| $L$ | latent heat | $3.34\times10^{5}$ | | $\text{J kg}^{-1}$ |
| $c_t$ | pressure melt coefficient | $7.5\times10^{-8}$ | | $\text{KPa}^{-1}$ |
| $c_w$ | heat capacity of water | $4.22\times10^{3}$ | | $\text{Jkg}^{-1}\,\text{K}^{-1}$ |
| $\alpha$ | first sheet flow exponent | 5/4 | | |
| $\beta$ | second sheet flow exponent | 3/2 | | |
| $\alpha_c$ | first channel flow exponent | 5/4 | | |
| $\beta_c$ | second channel flow exponent | 3/2 | | |
| $k_s$ | **sheet conductivity** | $10^{-4}$ | $10^{-2}$–$10^{-5}$ | $\text{m}^{7/4}\,\text{kg}^{-1/2}$ |
| $k_c$ | **channel conductivity** | $10^{-1}$ | $5\times10^{-1}$–$10^{-3}$ | $\text{m}^{3/2}\,\text{kg}^{-1/2}$ |
| $E_{vr}$ | **englacial void ratio** | $10^{-4}$ | $10^{-3}$–$10^{-5}$ | |
| $l_c$ | sheet width below channel | 2 | | m |
| $A_m$ | moulin cross-sectional area | 10 | | $\text{m}^2$ |
| $l_r$ | cavity spacing | 2 | | m |
| $h_r$ | **basal bump height** | 0.085 | 0.05–0.1 | m |
| $b_{melt}$ | **basal melt rate** | $5\times10^{-3}$ | $1$–$7\times10^{-3}$ | $\text{m yr}^{-1}$ |
| $U_b$ | **mean annual velocity**[†] | 150 | 100–200 | $\text{m yr}^{-1}$ |
| $N_{moulins}$ | **number of moulins**[*] | 2500 | 1000–4000 | |

[†] We tested both a transient and temporally constant velocity within these given ranges for mean annual velocity

[*] We also ran an experiment in which melt was routed directly to the bed at each node (SHEET)

channels (R-channels) that are allowed to change diameter—along element edges (Schoof, 2010). A key advantage of GlaDS lies in its ability to capture the growth and restriction of these channels entirely due to drainage dynamics, without requiring a predetermined drainage system (Dow et al., 2020). Water flux, $q_s$, through the distributed system is driven by the hydraulic potential gradient, $\nabla \emptyset$, along with the sheet conductivity, $k_s$

$$q_s = -k_s h^\alpha |\nabla \emptyset|^\beta \nabla \emptyset, \tag{1}$$

where the first ($\alpha$) and second ($\beta$) sheet flow exponents describe fully turbulent flow in the Darcy-Weishbach law, and $h$ is the sheet thickness. The sheet thickness evolves through time given by

$$\frac{\delta h}{\delta t} = w - v, \tag{2}$$

for functions $w$ and $v$ which describe the cavity opening and closing rate respectively (Walder, 1986; Kamb, 1987). Basal sliding opens cavities at a rate given by the basal sliding speed, $U_b$ acting over basal bumps with a height, $h_r$ through

$$w(h) = \begin{cases} U_b (h_r - h) / l_r & \text{if } h < h_r \\ 0 & \text{otherwise} \end{cases} \tag{3}$$

where $l_r$ is the typical horizontal cavity spacing. In turn, viscous ice deformation leads to cavity closure, which is related to the effective pressure, $N$ by

$$v(h, N) = Ah|N|^{n-1}N, \tag{4}$$

where $A$ is the rate factor, or the rheological constant of ice, multiplied by a first order geometrical factor, and $n$ is the Glen's flow law exponent. Sheet elements exchange water with channels and the cross sectional area of these channels $S$, evolves through time due to the dissipation of potential energy, $\Pi$, sensible heat exchange, $\Xi$, and cavity closure rates due to viscous ice creep $v_c$

$$\frac{\partial S}{\partial t} = \frac{\Xi - \Pi}{\rho_i L} - v_c, \tag{5}$$

where $\rho_i$ is the ice density and $L$ is the latent heat of fusion. In GlaDS, channels are able to form along all element edges and channel discharge, $Q_c$, is always non-zero along these edges. Following Werder et al. (2013), we set a threshold discharge of $Q_c = 1\, \text{m}^3\, \text{s}^{-1}$ above which an element edge is classified as a 'meaningful' channel for our subsequent analysis. Surface melt can either be routed to the bed via a series of moulins, represented as cylinders with a fixed cross sectional area, $A_m = 10\, \text{m}^2$, or delivered directly to the bed at every node. An englacial void ratio term, $E_{vr}$ controls the volume of water stored in englacial aquifers to mimic the observed delay between daily maximal melt input and peak proglacial discharge (Werder et al., 2013). Finally, in the iteration used here, GlaDS is not coupled two-ways to a model of ice dynamics, and instead we prescribe an ice velocity and geometry that is not variable in response to hydrological forcing.

### 3.1.1 Boundary conditions and forcings

To model basal hydrology, GlaDS requires user inputs for melt forcing, bed elevation, $z_b$ and ice thickness, $H$ as well as boundary conditions and parameters (Table 2) detailed below. We anticipate that the modern topography is not representative of bed elevation $\sim$12 cal. ka. Therefore, as the baseline boundary condition, $z_b$, we account for changes, particularly in terrain associated with the second Salpausselkä ice-marginal formation, by subtracting Quaternary sediment thickness estimates (GTK, Finland, 2010) from the 25 m/pixel EU-Digital Elevation Model V1.1 (available at: https://www.eea.europa.eu/data -and-maps/data/copernicus-land-monitoring-service-eu-dem). Because lake bathymetry was only partially available we did not subtract this from our input DEM in the baseline model. We also did not adjust our model to account for differences in elevation due to glacial isostatic adjustment (GIA) since $\sim$12 cal. ka. Available sea-level markers from Rosentau et al. (2021) and Ojala et al. (2013) suggest that uplift of $\sim$80 m has occurred in the southeastern portion of our domain, and up to 200 m in the northwest since $\sim$12 cal. ka. Combined with eustatic sea level rise, these differences account for a maximum of 100 m difference in elevation relative to our DEM and a tilting of the basin towards the southeast during glaciation. Variable through time, accounting for GIA would result in a maximum increase to the mean annual air temperature of $\sim$0.75°C (based on our chosen lapse rate of 7.5°C km$^{-1}$, Section 3.1.1) across portions of our domain. To ensure the numerical stability of GlaDS the input DEM was smoothed using a low-pass filter. Finally, within steep terrain, an anisotropic mesh ($n_{nodes} \approx 19,000$) was refined based on $z_b$ such that element edges were shortest (to a minimum edge length of 400 m) in rougher terrain and longer where terrain was flatter (to a maximum edge length of 2 km). As boundary conditions, we imposed a zero flux condition on the domain edge everywhere except at the ice terminus, where given spatial variability in water depth (Lunkka and Erikkilä, 2012), an outlet Dirichlet condition equivalent to atmospheric pressure was prescribed in the baseline model. By enforcing zero input flux we neglect to include basal water input from beyond the model domain and we also do not account for any exchange of water between adjacent ice lobe provinces. To promote model stability, we used an adaptive timestep that was allowed to vary between one hour and $\sim$90 seconds and all of our transient models were run for 10,000 days, or $\sim$27 years.

An approximation for the FIS ice thickness, $H$, at $\sim$12 cal. ka within the FLDIL was generated using the 2D Shallow-Shelf Approximation (SSA, MacAyeal, 1989) within ISSM (Larour et al., 2012). Ice is assumed to be isothermal with a viscosity, $B$, equivalent to an ice temperature of -5 °C (from Cuffey and Paterson, 2010, p.73; rate factor, $A$, listed in Table 2). In reality, ice temperature is both spatially and temporally variable. However, without using a more detailed thermomechanical ice model, we follow the previous ad-hoc assumptions of Nick et al. (2013) for the Greenland Ice Sheet and Åkesson et al. (2018) for the FIS, by setting our ice temperature to -5°C. Basal motion was modelled using a viscous sliding law (Budd et al., 1979) and following Åkesson et al. (2018) we used a spatially variable basal drag coefficient, $a$, proportional to $z_b$, given by:

$$a = 120 \frac{\min\left(\max\left(0, z_b + 800\right), 2000\right)}{2000}. \tag{6}$$

To reach volumetric steady state, defined for our ice sheet model as differences in ice volume between successive iterations of less than $10^{-6}$ km$^3$, we ran the ice sheet model for 20,000 years with an adaptive timestep, allowed to vary between 1 day and 1 year. An initial estimate of ice surface elevation was given using a parabolic profile as a function of distance from the

terminus, and initialisation values for velocity were calculated using a stress balance solution for this ice surface. Dirichlet conditions were imposed at the mesh edges along the boundary with zero inflow.

We used climatic forcing both for our ice sheet model and for GlaDS. The $12\,\mathrm{cal.\,ka}$ climate was estimated using a modern (1981-2010) reanalysis dataset (see Abatzoglou et al., 2018). Precipitation was kept at the contemporary monthly value, but we depressed monthly temperature by $15°\mathrm{C}$, approximately the temperature differential indicated by NGRIP $\delta 18O$ records (Johnsen et al., 1997). In simply depressing the climate we are neglect the complex seasonality (short, warm summers with extreme winters) that characterised the Younger Dryas cold reversal in Fennoscandia (Schenk et al., 2018; Amon et al., 2022).

However in fixing our domain to the second Salpausselkä our model is representative of the end of the Younger Dryas at which time this extreme seasonality rapidly gave way to a markedly warmer climate with similar seasonality to the present day (Mangerud et al., 2023). To calculate surface mass balance efficiently in our long term ice sheet model we used a simple positive degree day (PDD) model (as in Cuzzone et al., 2019) allowed to vary about a fixed Gaussian distribution with standard deviation, $\sigma_{PDD} = 5.5°\,\mathrm{C}$ around the monthly mean and a lapse rate of $7.5°\mathrm{C\,km^{-1}}$. However, our focus here is on the basal

hydrology and we used a modified PDD scheme to estimate meltwater production for our GlaDS simulations.

    It is commonly assumed that the total monthly positive degree days can be represented by a fixed Gaussian distribution with $\sigma_{PDD} \approx 5.5°\,\mathrm{C}$ (e.g., Braithwaite and Olesen, 1989). However, field measurements suggest that this does not hold for the Greenland Ice Sheet (Wake and Marshall, 2015), particularly at temperatures $\geq -5°\,\mathrm{C}$. Instead, Wake and Marshall (2015) suggest monthly variability in temperature, $\sigma_M$, is more accurately described by a quadratic function:

$$\sigma_M = -0.0042T_M^2 - 0.3T_M + 2.64, \tag{7}$$

where $T_M$ is the mean monthly temperature. This function accounts for the observation that variability in temperature decreases with increasing temperatures (Gardner et al., 2009; Marshall and Sharp, 2009; Fausto et al., 2011) due to heat buffering, which promotes a more stable boundary layer (Wake and Marshall, 2015). We used the calculated $\sigma_M$ to add Gaussian noise to a daily temperature record estimated by linearly interpolating our depressed MAT record. The number of positive degree days

325 per month, $PDD_M$ was taken as $PDD_M \geq -5°\,\mathrm{C}$. We used $-5°\,\mathrm{C}$ as our threshold (rather than the more commonly used $0°\,\mathrm{C}$ threshold) to account for melt which may occur even for days with an average temperature of $0°\,\mathrm{C}$ (see van den Broeke et al., 2010). Finally, we used melt rate factors $\gamma_{ice} = 17.22\,\mathrm{mm\,per\,PDD}$ and $\gamma_{snow} = 2.65\,\mathrm{mm\,per\,PDD}$ following Cuzzone et al. (2019) keeping these consistent between our ice sheet model and GlaDS model. Monthly melt was kept fixed annually for each run. Melt varied in absolute terms between individual simulations but the mean melt and standard distribution remained

identical throughout.

    Total monthly melt was then converted to yearly melt rates and routed to the bed via a series of moulins. Without detailed ice sheet surface topography and following Werder et al. (2013) we divided our domain using Voronoi tessellation on a randomly distributed series of points. Within each Voronoi cell, acting as a 'catchment zone', the lowest elevation node was identified and used as the location for a moulin towards which all melt from all other catchment nodes flow. Surface melt rate was integrated

over each catchment and converted to instantaneous moulin discharge, $Q_m^k$.

### 3.1.2 Steady state and sensitivity testing

The GlaDS model has been extensively sensitivity tested for contemporary ice sheets where model results can be compared with geophysical evidence to determine the most plausible model output (e.g., Werder et al., 2013; Dow et al., 2018b, 2020, 2022; Indrigo et al., 2021; Scholzen et al., 2021). We set the parameters in our baseline model (default values listed in Table 2) following the default values in these studies which provide a reasonable approximation of contemporary ice sheet subglacial conditions. We then explored the sensitivity of our specific model outcomes to the available parameters (e.g., conductivity terms) in GlaDS throughout the range of values listed in Table 2, as well as the dependence of our results on our choice of forcing and boundary conditions (Section 4.2).

Before all model runs, we ran GlaDS to steady state with basal meltwater input but no surface melt. We did this to avoid overwhelming an unpressurised initial system with sudden surface meltwater inputs and to approximate a wintertime hydrology configuration characterised by a distributed system with high water pressures. To guarantee the majority of elements were pressurised at the end of each steady state run, we used a low, fixed velocity of $30\,\mathrm{m\,yr^{-1}}$ to limit the rate of cavity expansion (see Equation 3). Given uncertainty regarding the spatial variability of basal melt rates, which vary as a function of geothermal heat and frictional heating, we used a spatially and temporally constant basal water input (as in Dow et al., 2018a, c, 2020; Poinar et al., 2019). Basal melt rates beneath the Greenland Ice Sheet typically range between $1\text{–}7\times10^{-3}\,\mathrm{m\,yr^{-1}}$ (see Karlsson et al., 2021) and we used $5\times10^{-3}\,\mathrm{m\,yr^{-1}}$ for our steady state configuration and the majority of the subsequent transient runs. To test the influence of basal melt rates on our system we ran an additional low basal melt rate ($1\times10^{-3}\,\mathrm{m\,yr^{-1}}$) and high basal melt rate scenario ($7\times10^{-3}\,\mathrm{m\,yr^{-1}}$) to steady state. Here, steady state was reached once the median difference in sheet thickness between two successive steps was less than $10^{-6}\,\mathrm{m}$. All three basal melt scenarios reached this by 20,000 days, and nodes reached water pressures 90% of overburden pressure, or $overburden_{\%}\approx90\%$ [1] with no channel formation.

For the baseline transient model run, we used the final configuration of our steady state run as an initialisation state with the addition of transient surface melt routed to the bed via ~2500 moulins, a density of 0.04 moulins per $\mathrm{km^2}$. Measured moulin density varies between 0.02 to 0.09 moulins per $\mathrm{km^2}$ in Greenland (Yang and Smith, 2016). To test the sensitivity of our system to moulin density we also ran models with ~1000 (0.02 per $\mathrm{km^2}$), ~4000 (0.06 per $\mathrm{km^2}$), and two further randomly generated configurations of the default ~2500 (0.04 per $\mathrm{km^2}$). We also tested an additional configuration in which melt at every node was routed directly to the bed. Further sensitivity testing (parameters listed in bold in Table 2) was carried out for several poorly constrained parameters in GlaDS, as well as for the basal geometry and moulin density. The conductivity of both the sheet, $k_s$ and channels, $k_c$ are the key controls on the extent and spacing of channels, with the basal bump height, $h_r$, and basal velocity also important. For basal velocity, we tested values between $100\text{–}200\,\mathrm{m\,yr^{-1}}$ chosen to be comparable to GPS measurements of surface velocity across land-terminating sectors of the Greenland Ice Sheet (e.g., Tedstone et al., 2015). We tested both a temporally fixed and temporally variable velocity, with the transient velocity varying between 85% and 140% of the mean to approximate speed-ups at the onset of the melt season and winter slowdowns commonly observed in Greenland (e.g., Sole

---

[1]Borehole measurements of overwinter water pressure in the distributed drainage system have been measured at 80-90% of overburden pressure (e.g., Harper et al., 2021)

et al., 2013). Without a more detailed understanding of past ice dynamics, the magnitude of velocity was kept spatially uniform throughout.

Although the default configuration describes a terrestrial margin, we also tested the influence of a shallow body of water at the ice margin by prescribing Dirichlet conditions at the ice margin where water pressure is equivalent to that of a uniform 30 m water depth (a simplification of the variable 5–50 m water depth from Lunkka and Erikkilä, 2012). To explore the influence of our modified topography boundary condition, we ran tests with a uniformly flat bed, one representing contemporary terrain (without Quaternary sediment thickness removed), and one with the available partial lake bathymetry removed. Finally, we also explored the dependency of our results on mesh geometry, including using a coarser mesh (maximum edge length of 5 km), a mesh not refined by elevation in any way, and a mesh in which a coarse mesh (edge length between 5–8 km) was prescribed >80 km from the ice margin and a much finer mesh (edge length ≈300 m) was prescribed <80 km from the ice margin.

## 3.2  Model validation from geomorphological datasets

Finally, we compared the GlaDS output to the subglacial hydrological conditions proposed for murtoo genesis. We anticipate that our domain contains a time-integrated record of landforms formed throughout the retreat of the FIS since the end of the Younger Dryas. With a fixed domain bound at the second Salpausselkä, we are effectively representing a single time slice ~12 cal. ka. As such, we expect that much of the landform record will not be well-represented by our model outputs, particularly those landforms further from the ice margin. However, closer to the ice margin (within 0–50 km), where work in Greenland would suggest the basal hydrological system is more likely to be channelised during summer (e.g., Chandler et al., 2013), we made visual comparison of modelled channel spacing, length, and complexity against esker deposits mapped by Palmu et al. (2021).

Similarly, and assuming that the mapped murtoo distribution is also representative of a time-transgressive mode of origin, we examine the performance of our model within the hypothesised zone of murtoo formation (e.g., Ojala et al., 2019; Ahokangas et al., 2021) by specifically isolating model nodes falling within 40–60 km of our ice margin representative of the FLDIL extent ~12 cal. ka. We then queried these nodes according to whether they occurred within a mapped meltwater routes (as mapped by Ahokangas et al., 2021); sub-dividing these routes based on the presence or absence of murtoo fields into murtoo routes (meltwater routes *with* murtoos) and meltwater routes (meltwater routes *without* murtoos). Using 500 m buffers, we approximated the lateral extent of murtoo/meltwater routes along 2D polylines representing their central long-axis from Ahokangas et al. (2021). In total, 244 nodes occur in murtoo routes, 951 nodes in meltwater routes, and 1205 nodes are not associated with any meltwater routes but are between 40–60 km of the ice margin. Ahokangas et al. (2021) mapped eskers separately to meltwater routes, including these as "channelised routes" in their dataset. However, Ahokangas et al. (2021) go on to note that many channelised routes fall within or intercept meltwater routes and likely postdate meltwater routes. Without age-control, we do not make a distinction between murtoo/meltwater routes and channelised routes here.

## 4 Results

In total, 30 simulations are reported on here, all of which successfully converged. In Section 4.1 we discuss our baseline test, and in Section 4.2 we report the parameter and forcing dependencies. For each model run, we examined the subglacial water pressure, expressed as a percentage of the overburden pressure, $overburden_\%$, sheet discharge, $q_s$ on element faces; channel discharge $Q_c$ on element edges, and water velocity, $V_W$.

### 4.1 Baseline scenario

#### 4.1.1 Model behaviour

After an initial adjustment from steady state to transient forcing over 5 years, the baseline model reached a quasi-steady state configuration in which the system responded seasonally to summer meltwater input (Figure A2). Figure 3 shows the median summer and winter state (excluding the initial adjustment time) of the baseline run in terms of $overburden_\%$ (Figure 3A), discharge within sheet elements, $q_s$ (Figure 3B), water velocity, $V_W$ (Figure 3C), and channel discharge, $Q_c$ (Figure 3D) as well as mapped murtoo fields from Ahokangas et al. (2021). In summer, modelled channels, shown as black solid lines (Figure 3A–C) typically extend up to 40 km from the ice margin creating valleys of low $overburden_\%$ (Figure 4A). Between 40 km and up to 60-70 km from the ice margin, $overburden_\%$ approaches and exceeds 100%. Beyond 70 km from the ice margin, with zero modelled channels, $overburden_\%$ is uniformly at $\sim 80\%$.

Throughout the year, $q_s$ sharply decreases 60 km from the margin (Figure 3B). In summer, areas of high $q_s$ (approaching $10^{-1}\,\mathrm{m^2\,s^{-1}}$) are found between channels 30–40 km from the ice margin which we interpret as arising due to channels draw down water from surrounding areas. In winter, $q_s$ is lower throughout the domain, and the highest values of $q_s$ ($\sim 1 \times 10^{-2}\,\mathrm{m^2\,s^{-1}}$ Figure 3B) are found in patchy areas within 60 km of the ice margin. Throughout the year, $V_W$ remains high ($\sim 15 \times 10^{-4}\,\mathrm{m\,s^{-1}}$) at the ice margin. During summer, $V_W \sim 8 \times 10^{-4}\,\mathrm{m\,s^{-1}}$ persist up to 50 km from the ice margin. In winter, the drop in velocity away from the ice margin is more pronounced, and higher water velocities ($> 5 \times 10^{-4}\,\mathrm{m\,s^{-1}}$) are limited to less than 50 km from the margin. Finally, Figure 3D shows $Q_c$ with $\sim 35$ parallel–sub-parallel channels visible in summer, during which median $Q_c$ reached a maximum of $100\,\mathrm{m^3\,s^{-1}}$. A number of channels have an anabranching structure as well as small tributaries. In the winter, $\sim 10$ channels persist in the winter median system with a maximum median discharge of $3\,\mathrm{m^3\,s^{-1}}$ (Figure 3D).

Figure 4C shows the system state during September of model year 19, arbitrarily chosen as a representative example of the model state at the end of a melt season. Beyond 100 km, upglacier of any significant surface melt inputs to the bed, only limited seasonal evolution of the hydrological system is evident (e.g., Figure 4A). Here, the system is effectively inert, with $overburden_\%$ remaining $\approx 80\%$ with only small periodic perturbations in $q_s$, $Q_c$, and $V_W$. Closer towards the ice margin, a sub-parallel pattern of channels emerges in summer months (Figure 4C), with channels arranged perpendicular to, and extending up to $\sim 50$ km inland of the ice margin and comparable in structure and spacing to the location of esker deposits in the FLDIL (Figure 2 Palmu et al., 2021). Approximately 40 main channels are evident in late summer (Figure 4), evenly spaced every $\sim 5$ km laterally, each of which is also fed by one to two levels of anabranching lower-order tributaries. Following the

initial period of adjustment to transient forcing, peak discharge for these channels is $200\,\mathrm{m^3\,s^{-1}}$ during summer ($\sim$June) each
year with a maximum cross-sectional channel area of $42\,\mathrm{m^2}$ (equivalent to a half-circle with radius, $r \approx 5\,\mathrm{m}$). Many channels
exhibit a biannual pattern of growth and decay (e.g., Figure 4B) persisting throughout the winter between two consecutive
summers. Following initial channel growth after the onset of the melt season (preceded by a sharp increase in $overburden_\%$),
$Q_c$ in these persistent channels drops towards $1\,\mathrm{m^3\,s^{-1}}$ but does not fall below the channelisation threshold before the following
summer. As a result, subsequent meltwater input through these persistent channels is quickly accommodated with only a small
increase in $overburden_\%$ and little change in $q_s$ (Figure 4B). An alternating spatial pattern of overwinter channel persistence
is evident. In any given year, channels will persist through winter in either the central third of the lobe or in the remaining two
thirds of the lobe (Movie A1).

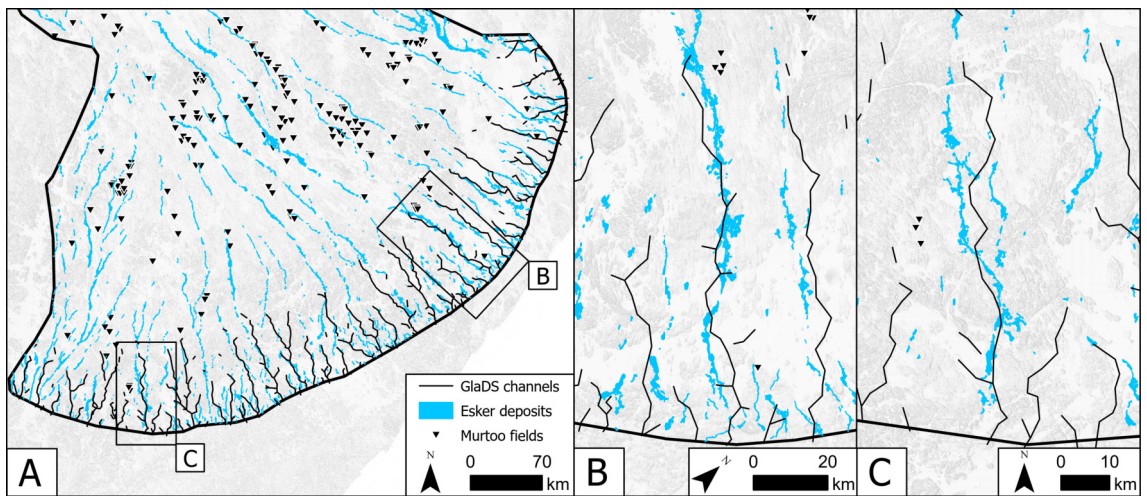

**Figure 2.** Modelled channel location compared to esker deposits mapped by Palmu et al. (2021) at the ice margin of the FLDIL. **A)** Modelled
channels in the baseline run (black lines) across the full width of the domain compared to esker deposits (blue polygons). **B & C)** Detailed
comparison of two large esker systems against model channels.

In the ice interior, at the head of modelled channels, a persistent area of high $overburden_\% \approx 100\%$ develops each melt
season (Figure 3A) following the onset and migration up-glacier of surface meltwater inputs. Figure 4D–E demonstrates the
seasonal evolution of two nodes in this area, each nearby to channel systems. Both nodes undergo a rapid seasonal increase in
$overburden_\%$ up to a maximum of approximately $120\%$ with a more gradual decrease thereafter. At node 3,842, chosen to be
representative of surrounding nodes at the onset of a channel (Figure 4E), this pattern repeats annually—every year the increase
and decrease in $overburden_\%$ is accompanied by peaks in $q_s$, $Q_c$, and $V_W$ and the development of channels throughout the
meltwater season. However, a more complex biannual signal is evident at node 16,402 (Figure 4D), which is located $\sim$0.7 km
from a murtoo field between the onset of adjacent modelled channels. Here, every other year, the evolution of $overburden_\%$
and $Q_c$ follow the expected model outcomes described in Table 1. Each year, there is a sharp increase in $overburden_\%$ at
the start of the melt season to $overburden_\% \geq 100\%$, however, the subsequent drop in $overburden_\%$ varies every other year.

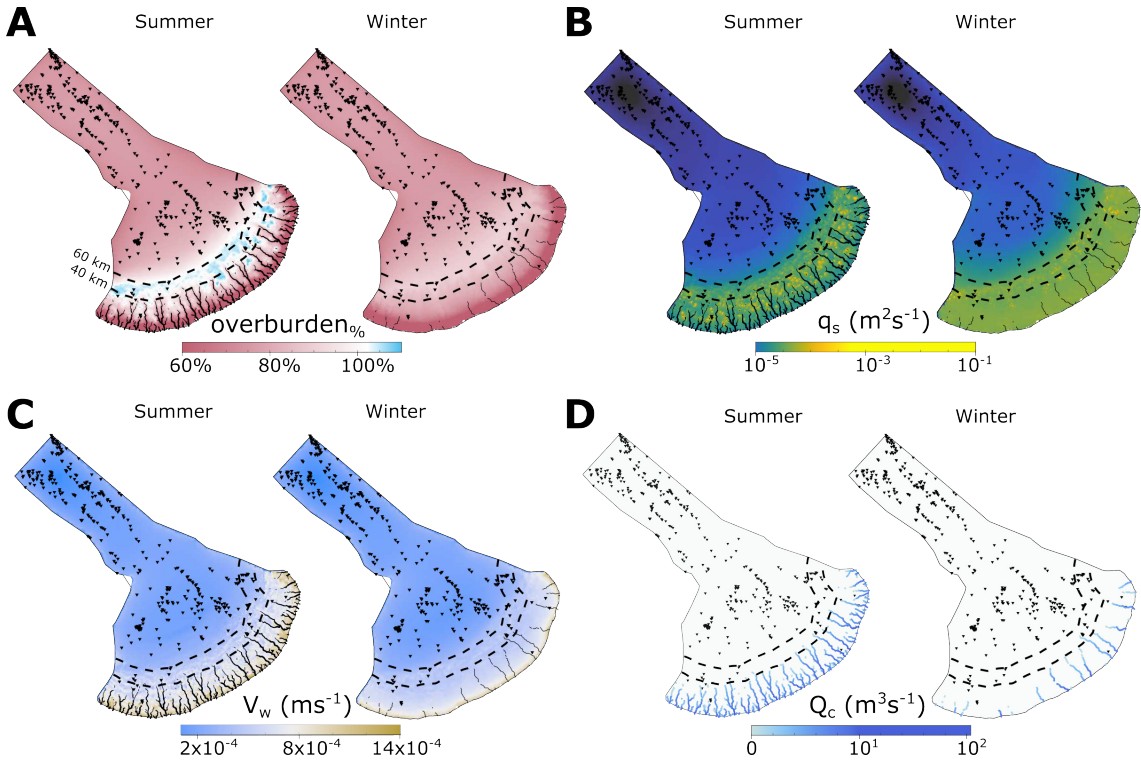

**Figure 3.** Median summer and winter system states in the baseline model run. **A**) Water pressure as a percentage of overburden pressure, $overburden_\%$. In summer and winter, **B**) Sheet discharge, $q_s$. **C**) Water velocity, $V_W$. **D**) Channel discharge, $Q_c$. For each output, we took the median from model years 5–27 disregarding the initial period of adjustment to transient forcing. *Summer* extends from May to September, all other points fall into *winter*. Note that the scales for panels B and D are logarithmic. Dashed lines in all panels indicate contours of 40 and 60 km from the ice margin. Murtoo fields (Ahokangas et al., 2021) are shown as inverted triangles in all plots. Channels are shown as black solid lines in panels A–C.

Either $overburden_\%$ spikes and then drops rapidly over 1-2 months to the winter value ($\sim$80%) until the following melt season, or the drop in $overburden_\%$ is initially shallower before quickly dropping to an elevated $overburden_\%$ ($\sim$90%) relative to the previous winter. Years in which the drop in $overburden_\%$ is more gradual are also associated with lower $Q_c$ and higher $q_s$. In contrast, years that have a rapid drop in $overburden_\%$ after the melt season are associated with values of $Q_c$ approaching 455 $1\,\mathrm{m^3\,s^{-1}}$.

### 4.1.2 Hydrology in the hypothesised zone of murtoo formation

We explored behaviours potentially associated with murtoo formation by focusing on nodes 40–60 km from the ice margin, within the zone thought to be associated with murtoo formation $\sim$12 cal. ka (Ojala et al., 2019). We grouped all nodes within 40–60 km by their relation to meltwater routes mapped by Ahokangas et al. (2021). The three groups included nodes i) within

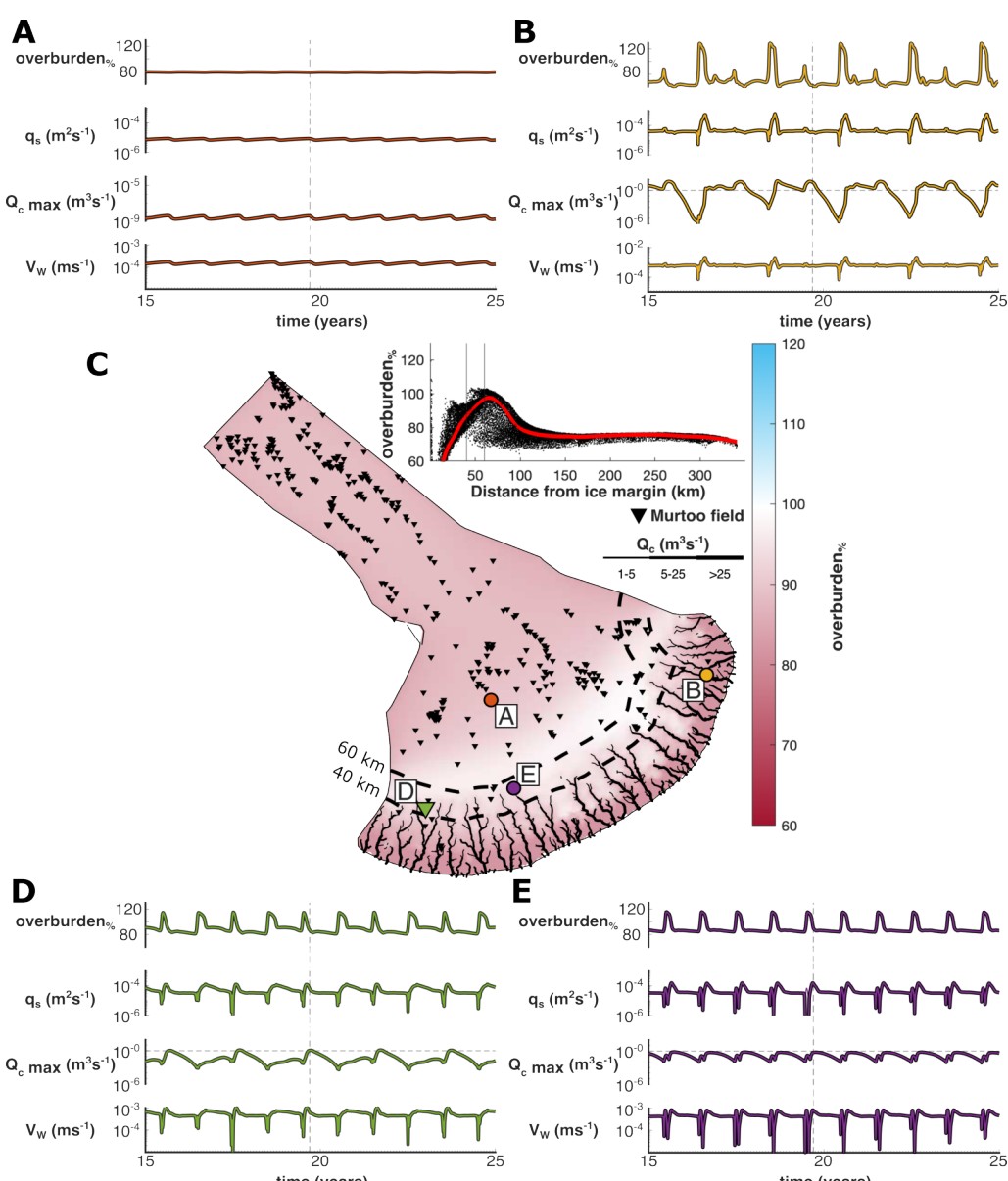

**Figure 4.** The evolution of water pressure as a percentage of overburden pressure, ($overburden_\%$), sheet discharge, ($q_s$), water velocity, ($V_W$), and maximum channel discharge, ($Q_{cmax}$) at four nodes over model years 15–25 in the baseline model run. **A**) Node No. 6,277 located ~120 km from the ice margin. **B**) Node No. 18,517 located ~17 km from the ice margin. **C**) Overburden at the end of the melt season in model year 19 (arbitrarily selected). Channels are represented as black lines, murtoo fields as inverted black triangles, and the location of panels A, B, D, and E as coloured points. Inset shows overburden at every node as a function of distance, $D$ from the ice margin with a smoothing function shown in red and vertical lines at 40 and 60 km from the ice margin. **D**) Node No. 16,402 located 0.7 km from a murtoo field and ~45 km from the ice margin. **E**) Node No. 3,842 located ~54 km from the ice margin at the head of a channel system without an adjacent murtoo field. The time slice shown in panel C is represented as a vertical dashed line in panels A, B, D, and E. Note the logarithmic scale for $q_s$ and $Q_{cmax}$.

the boundary of a murtoo route (a meltwater route containing murtoos $n = 241$), ii) within the boundary of a meltwater routes (that *does not* contain a murtoo field $n = 955$), and iii) beyond any mapped meltwater routes (all other nodes, $n = 1205$) (Figures 5 & 6). As noted in Section 3.2, group i and ii may also include eskers ('channelised routes' in Ahokangas et al., 2021) as these are often coincident with meltwater routes.

     Nodes that fall within a murtoo or meltwater route (Groups i and ii) show strong seasonal variation and at every point

throughout the year have higher $overburden_\%$, $q_s$, $V_W$, and $Q_c$ than nodes that do not fall within a murtoo/meltwater route. The median signal of nodes within murtoo and meltwater routes is one of sharp increases at or just following the onset of the melt season, followed by a more gradual decline into winter (Figure 5). One-way ANOVA analysis in which the values of $overburden_\%$, $q_s$, $V_W$, and $Q_c$ between groups i–iii were considered without respect to time, indicates that there is a statistically significant difference in the population marginal means (or the mean within each grouping) of the three categories

($p < 0.05$ at the 95% confidence interval).

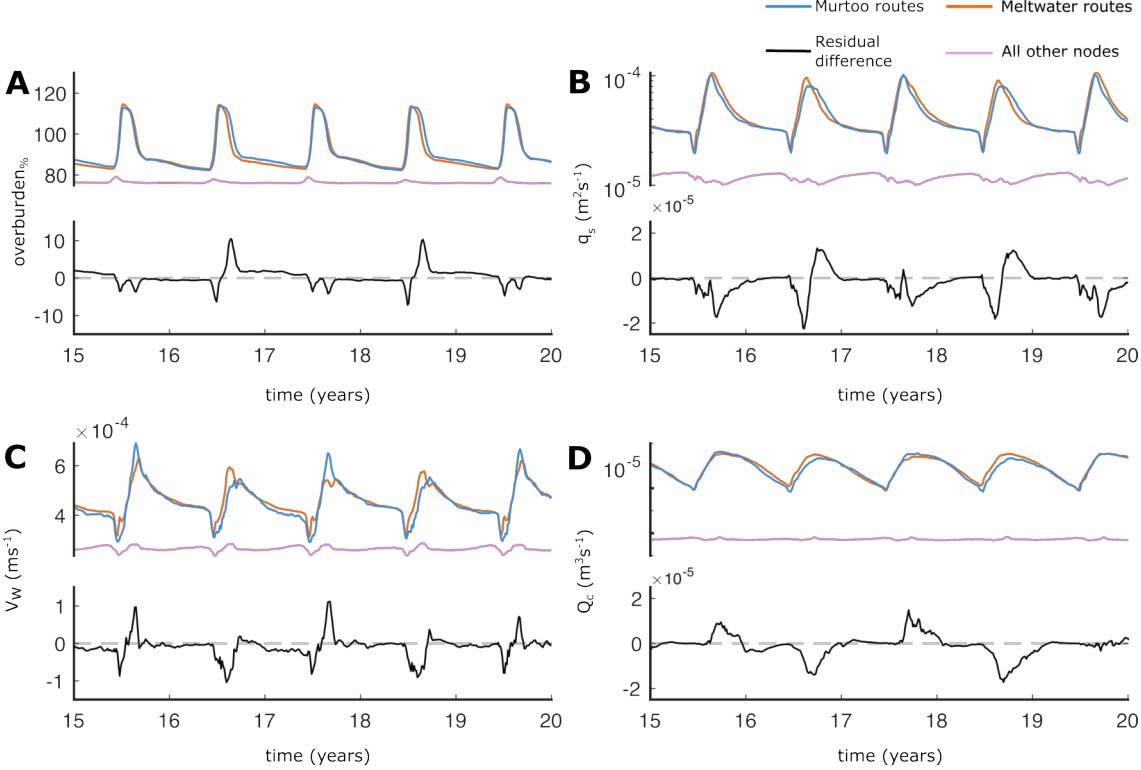

**Figure 5.** Median output during model years 15–20 at nodes between 40–60 km from the ice margin. In all panels, nodes that fall within murtoo routes are shown in blue, those which fall within meltwater routes (without murtoos) are shown in orange, and all other nodes are shown in purple. The black line in each panel represents the residual difference between the median of murtoo routes and meltwater routes. A positive residual indicates higher median values in murtoo routes, and vice versa for negative residual values. **A**) Water pressure expressed as a percentage of overburden, $overburden_\%$ **B**) Sheet discharge, $q_s$. **C**) Water velocity, $V_W$. **D**) Channel discharge, $Q_c$. Note panels B and D have logarithmic scales.

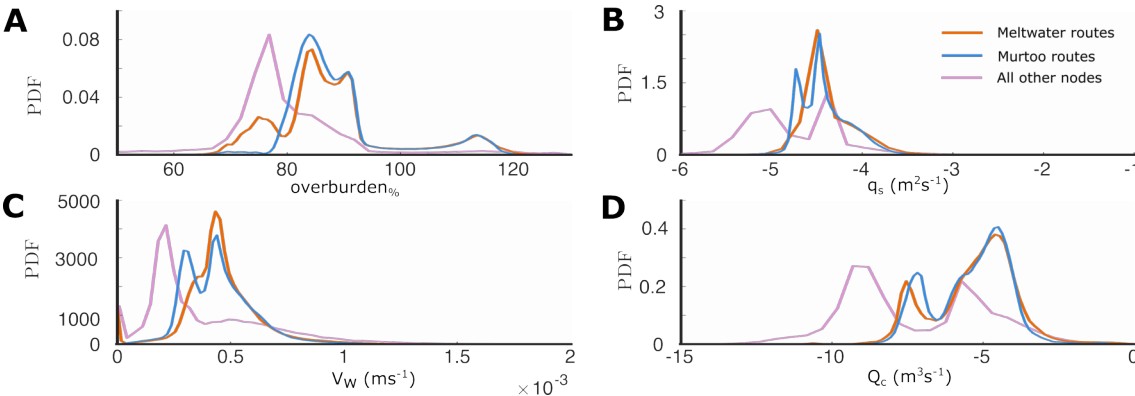

**Figure 6.** Probability density estimates from kernel smoothing of output parameters during all model years at nodes between 40–60 km from the ice margin. As in Figure 5, nodes that fall within murtoo routes are shown in blue, those which fall within meltwater routes (without murtoos) are shown in orange, and all other nodes are shown in purple. **A**) Water pressure expressed as a percentage of overburden, $overburden_\%$ **B**) Sheet discharge, $q_s$. **C**) Water velocity, $V_W$. **D**) Channel discharge, $Q_c$. Note panels B and D have logarithmic scales.

Additionally, we grouped each node observation within calendar months (Figure A32). Comparison of groups i–iii within each month—using two-way ANOVA analysis and the Tukey-Kramer HSD Test for multiple comparisons of unequal group sizes—indicate that there are significant differences ($p < 0.05$) between the population marginal means throughout the year for each of the four parameters discussed here (Tables A1–A4). In terms of $overburden_\%$, both murtoo routes and meltwater routes are significantly higher (10–30%) than nodes outside of murtoo/meltwater routes during every month. However, both murtoo and meltwater route groups are also significantly different from each other every month. Between January and April, $overburden_\%$ is significantly higher by 1–3% in murtoo-hosting meltwater routes nodes than in murtoo-free meltwater routes. Following the onset of the melt season, during June and July, $overburden_\%$ is lower by 1–5% in murtoo route nodes than in meltwater route nodes, before again returning to higher $overburden_\%$ (1–3%) in murtoo route nodes relative to meltwater nodes between August–December. In terms of $q_s$ (Table A2) both groups of murtoo and meltwater routes are higher than nodes beyond murtoo/meltwater routes all year by $\sim 1 \times 10^{-5}\,\mathrm{m^2\,s^{-1}}$. Between January and May there are no significant differences between murtoo routes and meltwater routes. However, between June and December, $q_s$ is significantly lower in murtoo routes than in meltwater routes by $\sim 1 \times 10^{-5}\,\mathrm{m^2\,s^{-1}}$. There is no significant difference between any group in terms of $Q_c$ with the exception of between June–October (Table A3), during which $Q_c$ is significantly higher within murtoo and meltwater routes than beyond by $\sim 1 \times 10^{-5}\,\mathrm{m^3\,s^{-1}}$. Finally, throughout the year, $V_W$ is higher in murtoo and meltwater routes than beyond them (Table A4) by $\sim 1 \times 10^{-3}\,\mathrm{m\,s^{-1}}$. In murtoo routes $V_W$ is significantly lower for each month than in murtoo free meltwater routes by $\sim 1 \times 10^{-4}\,\mathrm{m\,s^{-1}}$.

## 4.2 Sensitivity tests

The parameter sensitivity of basal drainage within GlaDS has already been extensively explored by Werder et al. (2013) and others (see Dow et al., 2018b, 2020, 2022; Indrigo et al., 2021; Scholzen et al., 2021) and as such we do not conduct a

detailed review here. However, because several parameters in GlaDS have uncertain physical values, we did test the robustness of our findings from the baseline scenario throughout the ranges indicated in Table 2 and describe the outcome of these changes below. In GlaDS, the spacing and lengths of channels, and in turn the influence of these channels on water pressure at the channel limits, is most sensitive to $k_s$ and $k_c$, describing the sheet and channel conductivity respectively. At the highest sheet conductivity ($k_s = 10^{-2}\,\mathrm{m}^{7/4}\,\mathrm{kg}^{-1/2}$, Figure A3) no channels longer than 1 km are formed and water is instead more readily transmitted through the distributed system at relatively low water pressures ($overburden_\% < 60\%$). With a lower sheet conductivity ($k_s = 10^{-3}\,\mathrm{m}^{7/4}\,\mathrm{kg}^{-1/2}$, Figure A4), around 60 channels were largely limited to within $\sim 10\,\mathrm{km}$ of the ice margin, and an area of high $overburden_\%$ 40–60 km of the ice margin was limited to a short one month period at the peak of the melt season. At a minimum sheet conductivity ($k_s = 10^{-5}\,\mathrm{m}^{7/4}\,\mathrm{kg}^{-1/2}$, Figure A5) $\sim$40 channels extend up to 50 km back from the ice margin. These channels appear more complex than those in the baseline run, with a more sinuous geometry and higher number of tributaries. Additionally, an extended area of high (but not 100%) $overburden_\%$ extends up to 150 km from the ice margin.

At the maximum channel conductivity ($k_c = 5 \times 10^{-1}\,\mathrm{m}^{3/2}\,\mathrm{kg}^{-1/2}$, Figure A6), $\sim$30 relatively linear channels extend up to 60 km from the ice margin. The area of highest $overburden_\%$ is pushed back to between 60–80 km from the ice margin, however $overburden_\%$ remains consistently below $\sim$ 90 %. At $k_c = 5 \times 10^{-2}\,\mathrm{m}^{3/2}\,\mathrm{kg}^{-1/2}$ (Figure A7 $\sim$ 30 channels are limited to $\sim$40 km of the ice margin. Within the 40–60 km distance from the ice margin, $overburden_\%$ is consistently in excess of 100% throughout summer. At the lowest value of channel conductivity tested here ($k_c = 10^{-3}\,\mathrm{m}^{3/2}\,\mathrm{kg}^{-1/2}$, Figure A8) a number of high $overburden_\%$ channels are restricted to within 1 km of the ice boundary, with a zone of $overburden_\% > 100\%$ extending 70 km from the ice margin. Excessively long ($>50\,\mathrm{km}$) or short ($<10\,\mathrm{km}$) channels compared to contemporary channels in Greenland and major changes in system pressure at the tested limits of $k_s$ and $k_c$ suggest our baseline conductivity terms are the most plausible parameters.

Changing the moulin density also alters the density, length, and complexity of channels as well as $overburden_\%$ beyond the upper limit of channel length. At the minimum moulin density tested ($N_{moulins} = 1000$, Figure A9), approximately 25 channels extend up to 50 km from the ice margin. The location of these channels closely follow the position of high discharge moulins near to the glacier terminus. A consequence of this is that $overburden_\%$ is less spatially continuous 40–60 km from the ice margin, though areas where $overburden_\% \approx 100\%$ are still in strong agreement with murtoo field location. A higher moulin density ($N_{moulins} = 4000$, Figure A10) with lower discharge has a similar impact on the spatial distribution of channels as increasing the conductivity of the sheet, with $\sim$60 channels reaching a maximum of 10–20 km from the ice margin, the location of which appears limited to the lowest elevation moulins closest to the margin at which higher discharges are prescribed. With lower discharge moulins also existing upglacier however, an area of high $overburden_\%$ extends up to 70 km from the ice margin. Routing water directly to the bed at every node instead of concentrating discharge through moulins (Figure A11) increases the frequency of short channels ($\sim$5 km) but does not alter the spacing of larger channels, which extend up to $\sim$40 km from the ice margin. Without water input at specific moulins, the area of high $overburden_\%$ associated with larger channels does extend further, with a clear pressure influence extending a further 10 km beyond each channel head. Two different random

variations of the default moulin density (Figures A12 & A13) altered the exact location of channels and pressure around these channels, but did not alter the overall pattern of pressure or drainage.

Changing the basal melt rate between $1$–$7 \times 10^{-3}$ m yr$^{-1}$ (Figures A14 & A15) had little impact on the pattern of channelisation and on pressure 40–60 km of the ice margin, likewise neither did altering the basal bump height between 0.1–0.05 m (Figures A16 & A17). Changing the mesh characteristics by not refining the mesh with respect to elevation alters the absolute position and detailed expression of channels but does not alter their spacing, length, or drainage in terms of pressure and discharge (Figure A18). Using a coarser mesh (Figure A19) lengthens channels, but this likely reflects the accompanying change in catchment areas and resultant drainage patterns through fewer moulins. Increasing the resolution within 80 km of the ice margin (Figure A20), results in major channels (those with a length longer than 20 km) extending up to 5 km further, but with a lower discharge over their full length, supplemented by more frequent small channels within 10 km of the margin. The absolute position of large channels changes compared to the default mesh, but their horizontal spacing remains consistent.

Modelling with a flat bed (Figure A21), using a modern bed (without removing Quaternary sediment thickness, Figure A22), and including the available lake bathymetry (Figure A23) has limited impact on channel density, length or drainage. Changing the terminus boundary conditions to approximate drainage into a shallow ($\sim$30 m deep) water body (Figure A24) also has limited influence on our results. Raising the englacial void ratio ($E_{vr} = 10^{-3}$, Figure A25) results in more complex channel geometry, and by increasing storage englacially, confines pressure variability nearer to channels. Lowering the englacial void ratio ($E_{vr} = 10^{-5}$, Figure A26) has no clear influence on channel geometry or pressure.

Finally, changing the basal ice velocity to a fixed value of 100 m yr$^{-1}$ (Figure A27) lowers $overburden_\%$ 40–60 km from the ice margin by $\sim$10% but does not alter channel spacing or length, while raising the basal ice velocity to a fixed value of 200 m yr$^{-1}$ (Figure A28) lowers $overburden_\%$ within channels <40 km from the ice margin. Introducing annual transient variability in velocity with a mean velocity of 150 m yr$^{-1}$ (Figure A29) limits the maximum length channels attain to $\sim$40 km from the ice margin relative to a fixed velocity. In addition, the transient velocity results in a spatial distribution of $overburden_\% \approx 100\%$ in stronger agreement with the contours of 40–60 km from the ice margin compared to the baseline scenario. A transient velocity with a mean of 100 m yr$^{-1}$ (Figure A30) does not clearly impact channels or $overburden_\%$ <60 km from the ice margin but does lower $overburden_\%$ further from the ice margin by $\sim$5%. A transient velocity with a mean of 200 m yr$^{-1}$ (Figure A31) has the opposite influence on $overburden_\%$ >60 km from the ice margin.

## 5   Discussion

### 5.1   Catchment-scale hydrological configuration

Murtoos are unique amongst glaciofluvial landforms in both their geomorphology and hypothesised genesis (see; Mäkinen et al., 2017, 2023; Peterson Becher and Johnson, 2021; Ojala et al., 2019, 2021, 2022; Ahokangas et al., 2021; Vérité et al., 2022; Hovikoski et al., 2023). In contrast to previously examined landforms such as eskers and tunnel valleys, which as channelised features have been previously investigated with relatively simple and inherently channelised basal hydrology models (e.g., Livingstone et al., 2013a, b, 2015; Kirkham et al., 2022), murtoos are thought to form within a transitional, semi-

distributed, drainage regime where widespread distributed systems give way to efficient and channelised drainage (Ojala et al., 2019). The weak to moderate deformation of murtoo sediments periodically evident within murtoo vertical exposures indicates

that water pressure remained close to overburden for sustained periods of time during murtoo formation (Peterson Becher and Johnson, 2021; Vérité et al., 2022; Mäkinen et al., 2023; Hovikoski et al., 2023). The broad and low geomorphology of murtoos, together with internal horizons that are arcuate at a similar curvature to the surface slope, suggest that murtoo deposition occurred within a low and broad cavity reaching a maximum water depth of 1 m (Hovikoski et al., 2023; Mäkinen et al., 2023). As discharge increases through a melt season, the cavity enlarges and is able to accommodate more sediment and

water flow before closing as water discharge decreases late in the melt season (Peterson Becher and Johnson, 2021; Mäkinen et al., 2023; Hovikoski et al., 2023). Crucially, the cavity never enlarges enough to form a channel of any appreciable size (Mäkinen et al., 2023). Murtoos are rarely found closer than 40 km from the ice margin (Ahokangas et al., 2021). At similar distances from the ice margin in Greenland, shallow surface gradients engender low hydraulic potential gradients, while low crevasse density limits meltwater input to the bed (Gagliardini and Werder, 2018), which together prevent the water supply

necessary to grow large, low water pressure channels (Dow et al., 2015; Bartholomew et al., 2011; Chandler et al., 2013; Ojala et al., 2019; Hooke and Fastook, 2007). Accordingly, murtoo formation is suggested to occur 40–60 km from the ice margin, within a high water pressure *semi*-efficient drainage system (e.g., Hovikoski et al., 2023; Mäkinen et al., 2023).

The predicted conditions associated with murtoo genesis therefore provide a unique set of criteria against which we can test GlaDS, a model of basal hydrology capable of resolving the transition in drainage modes between distributed and channelised

water flow (Werder et al., 2013). Our modelling output here, in both the baseline model (Section 4.1) and many of the sensitivity test (Section 4.2) closely match the predictions for murtoo genesis. Our baseline model predicts channels extending up to 40–50 km from the ice margin during summer (Figures 3, 4 and A), extending in to but not beyond the hypothesised zone of murtoo formation 40–60 km from the ice margin, supporting the idea that murtoos form in the transition in drainage from distributed systems to channelised systems (Ojala et al., 2019). During the melt season, modelled channels align well with murtoo fields

in the southwest and northeast portions of the FLDIL (e.g., Figure 4C). The median cross-sectional area of these channels 40–60 km from the margin is 2.8 m$^2$ (equivalent to a semi-circle with radius of 1.3 m), close to the maximum cavity height of 1 m inferred from boulder distributions in the upper slope of murtoos (Hovikoski et al., 2023). In GlaDS, channels are assumed to be semi-circular R-channels, not broad and low canals (as described by Walder and Fowler, 1994). However, the close agreement in approximate radius suggests that the limited cavity expansion or restricted channel floor width within which murtoo form is

captured within our model. At the head of channels in our baseline model, our modelling also reproduces the expected window of high $overburden_\%\approx100\%$ within 40–60 km of the ice margin associated with the presence of a semi-distributed system during the melt season (Figures 3, 4 and A). If we accept the hypothesis that murtoos form where $overburden_\%\approx100\%$ our modelling supports the idea that the murtoos mapped >70 km from the ice margin postdate 12 cal. ka and that murtoo formation is time-transgressive (Ahokangas et al., 2021). Within 40 km of the ice margin, $overburden_\%$ remains lower than

100% in both winter and summer, reflecting i) the presence of channelised drainage efficiently evacuating water close to the ice margin (<40 km) during summer and ii) the limited meltwater supply across the domain during winter. Further than 70 km from the ice margin, with low or zero atmospheric meltwater input, $overburden_\%$ remains constant throughout the year, rarely

dropping below 75% (Figure 4C–E) within a constantly distributed system. Similarly, $q_s$ and $V_W$ are both low >60 km from the ice margin and increase closer to the ice margin. This modelled spatial pattern of $overburden_\%$ and the expression of modelled channels holds for most but not all of our sensitivity tests. At the upper and lower magnitude limit of the conductivity terms, the channel length changes as the efficiency with which either the distributed or channelised system could transmit water was limited so that either no channels formed or a high density ($n = 50$) of very short channels (5–10 km) formed close to the ice margin. Similarly, when moulin density was highest, the reduced discharge associated with any one moulin resulted in a higher density of short channels. In each of these models, the pattern of $overburden_\%$ reflects the length of channels and an area of high $overburden_\%$ is either not present at all, or extends nearly the full extent of the FLDIL lobe where moulin density is extremely high and numerous low discharge moulins result in a highly pressurised distributed system. However, the pattern of $overburden_\%$, channel length and spacing remains largely insensitive to all other parameter changes relative to the baseline model, suggesting our conclusions are largely robust to specific parameter choices and that the baseline model is a plausible representation of the FLDIL drainage system.

Without extant ice in the FLDIL against which to test the validity of our model inputs and outputs, we are unable to fully determine the correct model parameters to describe our FLDIL domain. As a result, the baseline model was parameterised following existing work on contemporary ice sheets (see Section 3.1.2). As expected, the baseline model provides a range of seasonal water pressure and channel lengths that are similar to models of contemporary ice sheets validated with geophysical methods (e.g., Dow et al., 2020). At the catchment scale, our results closely match the apparent spatiotemporal expression of channelisation in land-terminating sectors of the Greenland Ice Sheets. Tracer transit times (e.g., Chandler et al., 2013) and basal hydrology modelling indicate efficient channelisation extends up to 40–50 km from the ice margin in Greenland, transitioning between channelised and distributed drainage modes at ice where ice is ∼900–1200 m thick (De Fleurian et al., 2016; Dow et al., 2015) as it does here. However, the pressure conditions within large channels close to the ice margin is notably different in our model results when compared to observations beneath the contemporary Greenland Ice Sheet (e.g., Van de Wal et al., 2015). In Greenland, subglacial channels form seasonally in response to meltwater discharge and exist at lower water pressures than the surrounding distributed system (Davison et al., 2019). The resultant hydraulic potential gradient forces large volumes of water from the surrounding distributed system towards channels, in turn lowering water pressure in the distributed system and increasing basal traction (Schoof, 2010). Even as meltwater delivery to the bed increases through the melt season, these channels can act to reduce ice velocity (Nienow et al., 2017) and reduce ice mass loss. In contrast, the channels modelled here remain at relatively high $overburden_\%$ throughout the year ($> 60\%$), with a lower hydraulic potential gradient between channelised and distributed systems. The FLDIL is relatively low-relief compared to the steep margins of the Greenland Ice Sheet (e.g., Wright et al., 2016), and the shallow topography may act to reduce the hydraulic gradient between distributed and channelised drainage. In a system such as the FLDIL, with low relief bed topography and high-pressure channels, it is likely that the influence of channelisation on velocity would be relatively limited, as lower rates of water exchange between distributed and channelised drainage permit more of the bed to remain closer to $overburden_\% \approx 100\%$, sustaining higher velocities for extended periods of time as a result (Dow et al., 2022).

## 5.2 Comparison with glaciofluvial landforms

We can also evaluate our model outputs by making comparisons to other glaciofluvial landforms. Modelled channels in our baseline model (Figure 2) and many of the sensitivity tests have similar locations as eskers mapped by Palmu et al. (2021), particularly in terms of their lateral spacing, length, and the observation that smaller esker deposits are alternately found between large features (Figure 2). The horizontal spacing ($\sim$15 km) of our channels is in close agreement with the theoretical spacing of eskers derived from the modelling results of Boulton et al. (2009) and Hewitt (2011). In the baseline model specifically, at several locations, modelled channel outputs closely track the location of several particularly large esker deposits (Figure 2B–C). We caveat this by noting that because our model operates on a mesh, the resolution of which is a balance of suitable fidelity against the increased computational cost of resolving finer details, the exact location of these modelled channels is sensitive to mesh geometry. Channels cannot form where no element edge exists. Differences in the exact channel location also arise because of moulin density and location, bed topography, velocity, and basal bump height. Nonetheless, the spacing and length of channels remains robust against the parameters tested here, and compares favourably to previous work, suggesting GlaDS is faithfully capturing the broad patterns of drainage beneath the FLIDL.

## 5.3 Comparison between model outputs and mapped murtoo locations

In order to directly compare the model outputs and location of mapped murtoos we grouped nodes according to whether they fell within a murtoo/meltwater route (both mapped by Ahokangas et al., 2021, see Section 3.2). For this analysis, we isolated nodes within 40–60 km of the ice margin. In doing so we accept the hypothesis that murtoos formed time-transgressively within 40–60 km of the retreating ice margin (Ojala et al., 2019; Ahokangas et al., 2021) an assumption that our baseline model and sensitivity runs would suggest is valid (Section 5.1). A general pattern emerges within these murtoo and meltwater route nodes that is absent in other nodes (Figure 5), one which largely agrees with the timeline of murtoo formation in a semi-distributed drainage system (Table 1, Peterson Becher and Johnson, 2021; Mäkinen et al., 2023; Hovikoski et al., 2023). At the onset of the melt season, $overburden_\%$ and $q_s$ sharply increases and peaks followed by peaks in $Q_c$ and $V_W$. The peak in $Q_c$ promotes a rapid drop in $overburden_\%$ and $q_s$, as cavity expansion promotes lower water pressure and more efficient discharge that, in turn, is able to redirect more water from the sheet elements along pressure gradients. In comparison, $Q_c$ drops more slowly, not reaching a minimum until the end of winter in the following year, at which point there is an abrupt drop in $V_W$ and $q_s$ coinciding with the minimum $Q_c$. Although the seasonal evolution is consistent with the formation of a murtoo, we fail to reproduce the sharp drop in discharge at the end of the melt season or the rapidly changing flow regimes within a single melt season (see Mäkinen et al., 2023; Hovikoski et al., 2023, and Section 2). However, in GlaDS the subglacial system is assumed to be pervasively hydraulically connected, and there is no mechanism which can lead to the hydraulic isolation of specific areas of the bed (e.g., Rada and Schoof, 2018; Hoffman et al., 2016). As a result, we do not expect to be able to reproduce the rapid changes in meltwater discharge necessary to form upper and lower flow regime deposits (e.g., Hovikoski et al., 2023, see Section 2) or laminated muds in marginal murtoo channels (e.g., Ojala et al., 2022). Nonetheless, the overall evolution of the system through time, and the sharp difference between murtoo/meltwater routes and areas of the bed without geomorphological

evidence of meltwater suggests that the baseline model is successfully reproducing many of the expected conditions of murtoo formation.

Our baseline model also makes a statistically meaningful distinction between murtoo routes and every other meltwater route (Section 4.1.2). The differences in probability distribution functions are largely similar between murtoo routes and meltwater routes, with the exception of noticeable differences particularly at the lower end of the distribution (Figure 6). Murtoo routes

in particular have a $overburden_\%$ distribution with a more tightly constrained lower tail with fewer nodes dropping below $overburden_\% = 80\%$ and a more variable $V_W$ than meltwater routes. Through time, the difference for each parameter is statistically significant (Tables A1– A4) during the melt season and also variable biannually (Figure 5). In terms of $overburden_\%$ this takes the form of higher water pressures within murtoo routes than in meltwater routes after the start of the melt season during even-numbered years. For the other outputs ($q_s$, $Q_c$, and $V_W$), every year, at the start of the year values are lower at

the start of the melt season in murtoo routes. In odd-numbered years, this is followed by a brief peak in murtoo routes not replicated in meltwater routes. In the following, even-numbered years, the values in murtoo routes are lower, for longer. This biannual signal can also be seen at individual nodes. Node No. 16,402, located $0.7\,\mathrm{km}$ from a murtoo field and $\sim45\,\mathrm{km}$ from the ice margin (Figure 4D), chosen to be representative of nodes in the immediate vicinity, undergoes an evolution through time similar to that within all meltwater routes and to the murtoo formation sequence, but only in odd-numbered years. In

even-numbered years, $overburden_\%$ peaks for longer and is elevated throughout the following winter, with much lower $Q_c$ as a result. This is in contrast to the similarly located Node No. 3,842, $\sim45\,\mathrm{km}$ from the ice margin and not neighbouring any murtoo fields, which undergoes a repetitive evolution year on year (Figure 4E).

It is difficult to say whether or not these differences are truly the model capturing subtle differences between water flow in meltwater routes and murtoo routes or if they arise due to our model setup. There is, for an example, a spatial component to

the biannual signal in our murtoo route outputs, potentially linked to the observation of winter channels persisting after the end of the melt season. Modelled channels, and the conditions at their headward extension do not always coincide with murtoo fields, particularly within the centre of our domain, 40–60 km from the ice margin. Here, our baseline scenario also reproduces apparent conditions for murtoo formation, including the termination of low-discharge channels and $overburden_\% \approx 100\%$ over a broad area during summer (e.g., Figure 5E). Despite this, no murtoo fields have been mapped in this area (Ahokangas

et al., 2021). Winter channels meanwhile follow a pattern in which channels in the central third of the FLDIL lobe persist in alternating winters to those in the northernmost and southernmost outer thirds. The presence of these winter channels likely influences the nearby system through the following summer, with preexisting channels dampening the influence of the initial melt input by providing an already established efficient drainage pathway. With an absence of murtoos in the central third of the lobe, the significant biannual difference between murtoo routes and meltwater routes (distributed more evenly across the

FLDIL, Figure A1A) may be an artefact of the spatial expression of winter channels. Murtoos appear to form within a semi-distributed drainage environment, and sedimentological studies indicate the movement of sediment is important in murtoo formation (Peterson Becher and Johnson, 2021; Mäkinen et al., 2023; Hovikoski et al., 2023). The reason that murtoos are not present in the centre of the FLDIL where our modelling suggests they should form may be a preservation issue or due to limited sediment supply. Sediment cover in this area is very thin, and the large areas of exposed bedrock likely limited

the upstream supply of sediment from which murtoos could form (Figure A1B), an interaction not yet accounted for in our modelling. Modern lakes are also abundant in the centre of the FLDIL and these may also act to mask murtoo routes.

However, the factors giving rise to the spatial pattern of winter channels themselves are more complex. On the Greenland Ice Sheet, winter slowdowns following high-melt summers have been linked to the sustained persistence of larger and more extensive channels into winter months (Sole et al., 2013) and their existence alone in our baseline model is not necessarily
surprising. There is a spatial variability to our meltwater inputs, arising from heterogeneity in the climate reanalaysis used to estimate the Younger Dryas climate (Section 3.1.1). However, our model forcing, though cyclical, has no interannual variability and no melt seasons are any more elevated than others. Additionally, as described in Section 3.1 and following Werder et al. (2013), an arbitrary minimum threshold $Q_c \geq 1\,\mathrm{m^3\,s^{-1}}$ was defined, above which $Q_c$ along an element edge was classified as a meaningful 'channel'. Channels persisting through winter months tend to operate at very low discharges of $1$–$3\,\mathrm{m^3\,s^{-1}}$, and
would not be categorised as channels with a higher threshold. Nonetheless, despite their low discharge and our fixed cyclical forcing, these channels do have a discontinuous spatial distribution with a biannual signal, which together with their winter persistence must arise, at least in part, for reasons besides our choice of external climate forcing or our choice of $Q_c$ threshold.

We anticipate that the lobate geometry of our model domain, chosen to be representative of the extent of the FLDIL $\sim 12\,\mathrm{cal.\,ka}$ may in part control the spatial expression of winter channels in our baseline model, in turn contributing to the
significant difference in the drainage characteristics of murtoo and meltwater routes. The divergence of ice flow vectors within the lobe appear to act as an initial perturbation which, together with spatial variability in the climate signal, results in an initially non-uniform concentration of meltwater within the lobe. We hypothesise that this local concentration of meltwater promotes large enough channels in portions of our model that are able to resist closure during winter, which subsequently act to more efficiently remove meltwater and lead to an earlier peak discharge the following summer, and an earlier closure in the following
winter. Such a repetitive biannual signal, and with it the resulting significant difference between murtoo routes and meltwater routes, is unlikely to persist in a more realistic model setup in which basal velocity can change in response to water pressure and subject to more realistic variability in meltwater forcing. Nonetheless, the landform record in the FLDIL does suggest that the divergence of flow in the ice lobe is an important control on glaciofluvial landform generation. In the FLDIL, flow parallel lineations (e.g., Figure 1E) indicate a largely uniform flow direction within the primary trunk that diverges radially within the
lobe. As a result, landforms within the FLDIL have previously been divided into three sub-lobes. The boundaries between these three sub-lobes are demarcated by particularly large esker deposits suggesting a concentration of meltwater here (Palmu et al., 2021). These eskers and the sub-lobes they bound align approximately with the distinct alternating pattern of over-winter channel persistence.

## 5.4 Limitations and future work

We make a number of simplifying assumptions to ensure models could run to completion with a walltime of 1–2 days while remaining numerically stable across the range of parameters tested for model sensitivity. These include smoothing of the bed topography below the maximum resolution available, and using a relatively large mesh. However, sensitivity testing indicates our conclusions are largely insensitive to topography, including its absence, and that the ice surface gradient instead imposes

the dominant control on basal hydrology. Similarly, changing the mesh resolution also appears to have limited impact on our conclusions. We did not account for changes in elevation due to glacial isostatic adjustment (GIA) since 12 cal. ka. Accounting for an anticipated uplift and tilting in this area reaching a maximum of $\sim$100 m (Ojala et al., 2013; Rosentau et al., 2021) is likely to increase the volume of melt delivered to the bed by elevating the mean annual air temperature by up to 0.75°C where uplift rates are highest, which will result in higher discharge channels that persist further upglacier of those high-uplift areas. Additional uncertainty arises from our estimated (and constant) meltwater and basal melt inputs, lack of diurnal forcing, fixed basal velocity, fixed conductivity parameters (in both space and time), fixed semi-circular channel geometry, assumed water turbulence, pervasive hydraulic connectivity, lack of water flux from abutting ice, and randomly seeded moulin inputs. Changes in geometry are known to be important in synthetic experiments of GlaDS (see Hayden and Dow, 2023) whereas we kept ice geometry fixed here. Finally, we note that in its uncoupled configuration, GlaDS does not account for a reduction in the frictional resistance to ice flow where $overburden_\%$ exceeds 100% or the increase in cavity closure rates that would accompany the increase in basal velocity associated with such a change in friction. In reality, sustained summer $overburden_\% \geq 100\%$ would result in the decoupling of the ice from the underlying bed as is suggested to be the reason for the limited observations of deformational structures within murtoo sediment exposures (e.g., Peterson Becher and Johnson, 2021; Mäkinen et al., 2023; Hovikoski et al., 2023). Future work should seek to address some of these limitations by including, for example, a more variable climate or coupled ice dynamics whereby the frictional resistance to ice flow is allowed to vary in response to changes $overburden_\%$ (e.g., as in Ehrenfeucht et al., 2023). Initial sensitivity testing of velocity forced to change seasonally does indicate that changes in velocity throughout the year is important for repressurising the system each winter to more closely match borehole records (e.g., Doyle et al., 2018, 2022).

## 6 Conclusions

In this paper we present the first application of the Glacier Drainage System model (GlaDS)—a dynamic basal hydrology model capable of resolving transitions between distributed and channelised drainage—to the palaeo setting. In doing so, we compared model outputs against the predicted conditions associated with murtoo genesis. Murtoos are a unique glaciofluvial landform, identified throughout Finland and Sweden in terrain formerly occupied by the Fennoscandian Ice Sheet (FIS). The alternating sequence of upper and lower flow regimes preserved within murtoos suggest that they formed amongst a network of small channels and cavities subject to rapid changes in water discharge and where water pressure met or exceeded ice overburden pressure. Further, their spatial distribution, rarely found closer than 40 km from the ice margin and often found downstream of ribbed moraines and upstream of eskers, suggests that murtoos represent the glaciofluvial imprint of a spatial and/or temporal transition between distributed and channelised drainage. We modelled this system using a setup representative of the Finnish Lake District Ice Lobe (FLDIL) at the end of the Younger Dryas, $\sim$12 cal. ka. Our model was forced with a positive degree model representative of the palaeo climate, as well as a modified digital elevation model and reconstructed ice surface elevation representative of the same time period.

Our model outputs reproduce many of the conditions predicted for murtoo genesis including:

i. An extensive area of water pressure at or equal to ice overburden pressure 40–60 km from the ice margin, largely robust to the range of parameters tested here.

ii. The annual evolution of a semi-distributed drainage system, which matches many of the anticipated conditions for murtoo genesis.

iii. The limited expansion of small cavities within the area of high water pressure to a maximum diameter of $\sim$1 m.

iv. Modelled channels which extend 40–50 km from the ice margin extending headward into the hypothesised transitional drainage zone associated with murtoo formation. These channels also have a similar spacing and geometry to mapped eskers in the region.

v. A statistically meaningful difference between areas of the bed without any indication of meltwater flow and areas of the bed with meltwater routes or murtoo routes.

Murtoo fields are not universally present where the conditions for their formation are predicted in our model, particularly within the centre of the FLDIL lobe, and we interpret this as a lack of upstream sediment supply further compounded by the high-density of terrain-obscuring lakes in this area. Additionally, we also find a statistically meaningful difference in water pressure, velocity, and discharge, between meltwater routes and murtoo routes, we interpret this as a combination of patchy murtoo distribution and internal model dynamics relating to the radial geometry of the lobe. Nonetheless, many of our model outcomes from the baseline model, in particular the area of high water pressure 40–60 km from the ice margin, are robust across the majority of 29 additional sensitivity tests carried out here, in which various values for model parameters and boundary conditions were tested within a range of numerical stability. At extremely high and low values of conductivity, parameters controlling how readily water flows through the distributed or channelised system, water was evacuated from the system too easily or slowly to form meaningful channels. However, across all other tests, including random mesh geometries, alternate bed topographies, changing ice velocity, and changing moulin density, similar patterns of modelled channels and water pressures emerge. Although our system is necessarily an idealised representation of the study area—not including adjacent and abutting ice lobes, an upstream catchment area, or a coupled representation of ice dynamics and basal hydrology—this work nonetheless demonstrates the potential application of state of the art basal hydrology models to the palaeo setting, where model outputs can be directly compared to geomorphology and specific models of landform genesis.

*Code and data availability.* All geophysical data used to parameterise the modelling (e.g., Quaternary sediment thickness, geothermal heat flux, lake bathymetry) is available from Finnish Geological Survey's 'Hakku' service (https://hakku.gtk.fi/?locale=en, last accessed on 06-09-2023). The Copernicus DEM used as basal elevation is available from: https://spacedata.copernicus.eu/collections/copernicus-digital -elevation-model (last accessed on 06-09-2023). For our modelling we used the Ice-sheet and Sea-level System Model (Larour et al., 2012) `revision 27448` available from: https://issm.jpl.nasa.gov/ (last accessed on 06-09-2023). Murtoo field locations from Ahokangas et al. (2021), glacial landforms shapefile data from Palmu et al. (2021), model results, and example input scripts used to produce and plot those results are available at the repository linked to this manuscript (https://doi.org/10.5281/zenodo.8344208, Hepburn et al., 2023)

*Video supplement.* Movie A1 is available at the online repository linked to this article (https://doi.org/10.5281/zenodo.8344208, Hepburn et al., 2023).

*Author contributions.* A.O, J.M, and C.F.D conceived the study, A.J.H designed and carried out the study and wrote the manuscript, all authors commented on the writing and helped with the analysis and interpretation.

*Competing interests.* The authors declare that no competing interests are present

*Acknowledgements.* This work forms part of the RewarD project (MUST consortium, University of Turku), funded by the Academy of Finland (grant numbers 322243/J.M and 322252/A.O). A.J.H is funded by the European Space Agency Internal Fellowship program, C.F.D is funded by the Canada Research Chair program (950-231237). All simulations were run on the Digital Research Alliance of Canada compute cluster, and we thank the European Union and the Finnish Geological Survey for enabling access to the data used to parameterise our model. We thank M.Werder for making the GlaDS model available, and we also thank M. Morlighem, J. Quinn, and J. Cuzzone for their help with ISSM.

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

## Appendix A: Contents

This file contains supplementary information for '*Reorganisation of subglacial drainage processes during rapid melting of the Fennoscandian Ice Sheet*'

**Movie A1.** Evolution of the system with respect to $overburden_\%$ through time in the baseline model run. Model years 15–17 were arbitrarily chosen to illustrate the transient state of the system through several melt season cycles. Channels are shown as black lines where $Q_c$ exceeds $1\,\mathrm{m^3\,s^{-1}}$.

**Figure A1.** The distribution of meltwater routes, murtoo routes, and sediment in the Finnish Lake District Ice Lobe. **A)** Meltwater routes and murtoo routes as mapped by Ahokangas et al. (2021). There is a general absence of murtoos in the centre of the lobe 40–60 km from the ice margin. **B)** Sediment cover (GTK, Finland, 2010) showing thin sediment thickness in the terrain from which murtoos appear absent.

**Figure A2.** Median $overburden_\%$, channel discharge, $Q_c$, and sheet discharge, $q_s$ per timestep over the full length of the baseline model run.

**Figure A3–A31.** Comparison of the median summer system for the range of sensitivity parameters against the baseline model run. **A)** Water pressure expressed as a percentage of overburden pressure, $overburden_\%$. Channels are shown as black lines where median discharge exceeds $1\,\mathrm{m^3\,s^{-1}}$. **B)** Baseline median summer $overburden_\%$ minus the tested median summer $overburden_\%$. The same figure caption applies for Figures A3–A31.

**Figure A32.** Boxplots of model parameters grouped by month for overburden ($overburden_\%$, **A**), sheet discharge ($q_s$, **B**), water velocity ($V_W$, **C**), and channel discharge ($Q_c$, **D**) during all model years at nodes between 40–60 km from the ice margin. As in Figure 5, nodes that fall within meltwater routes which do host murtoos (murtoo routes) are shown in blue, nodes which fall within mapped meltwater routes that do not contain murtoo fields (meltwater routes) are shown in orange, and all other nodes are shown in purple. Medians for each group are shown as black circles, and 'outliers'—defined as points more than 150% of the interquartile range away from the upper and lower quartile—are shown as crosses.

**Tables A1–A4.** Tukey-Kramer HSD test results for $overburden_\%$ (Table A1), $q_s$ (Table A2), $Q_c$ (Table A3), and $V_w$ (Table A4) in meltwater routes, murtoo routes, and non-meltwater routes between 40–60 km from the ice margin. The upper and lower limits describe the 95% confidence intervals for the true mean difference, A-B is the difference between group means.

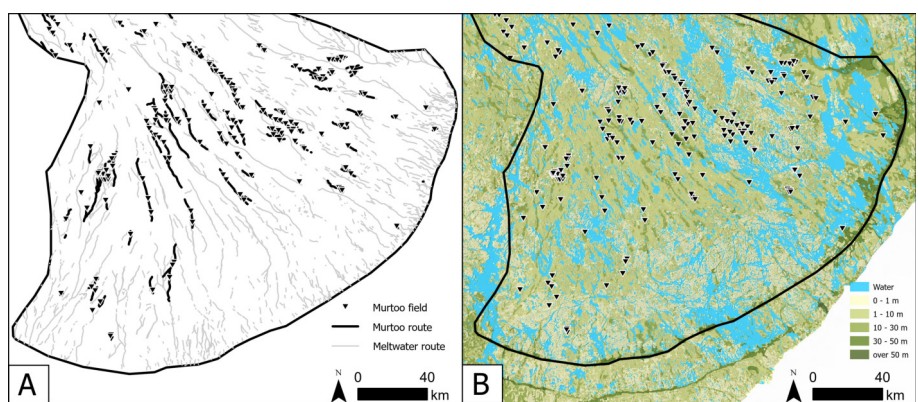

**Figure A1.** The distribution of meltwater routes, murtoo routes, and sediment in the Finnish Lake District Ice Lobe. **A)** Meltwater routes and murtoo routes as mapped by Ahokangas et al. (2021). There is a general absence of murtoos in the centre of the lobe 40–60 km from the ice margin. **B)** Sediment cover (GTK, Finland, 2010) showing thin sediment thickness in the terrain from which murtoos appear absent.

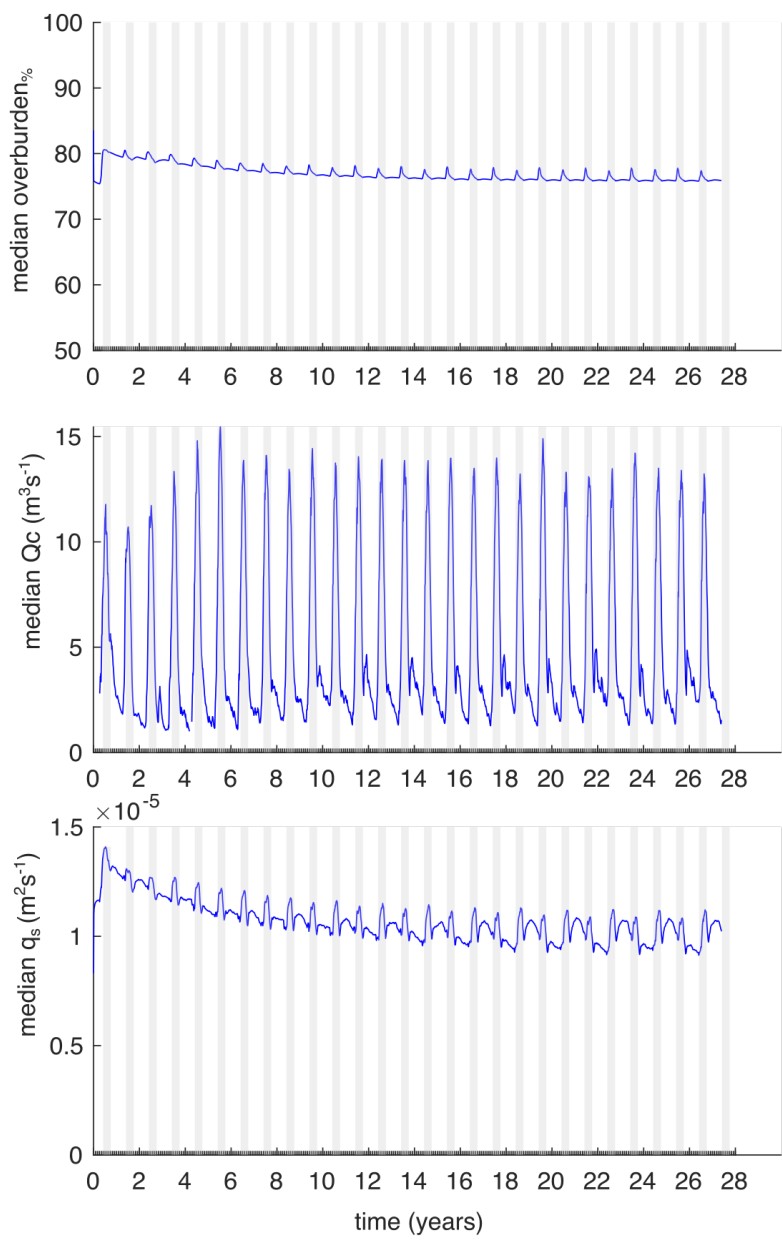

**Figure A2.** Median $overburden_\%$, channel discharge, $Q_c$, and sheet discharge, $q_s$ over the full length of the baseline model run.

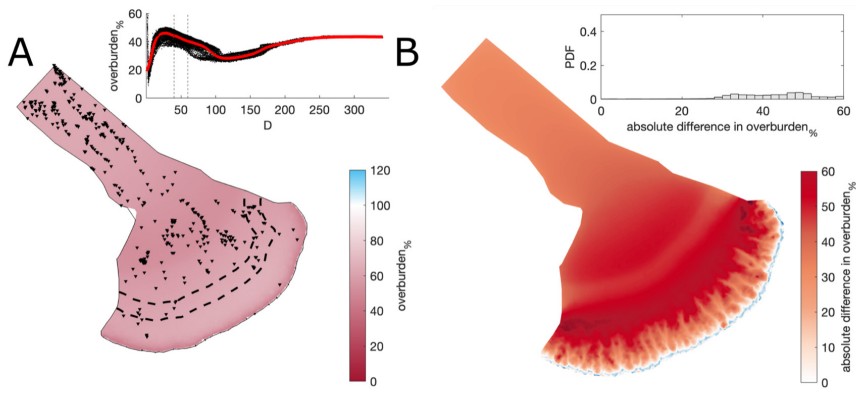

**Figure A3.** Comparison of the median summer system for sheet conductivity, $k_s = 10^{-2}\,\mathrm{m}^{7/4}\,\mathrm{kg}^{-1/2}$ against the baseline model run ($k_s = 10^{-4}\,\mathrm{m}^{7/4}\,\mathrm{kg}^{-1/2}$). **A)** Water pressure expressed as a percentage of overburden pressure, $overburden_\%$. Channels are shown as black lines where median discharge exceeds $1\,\mathrm{m}^3\,\mathrm{s}^{-1}$. **B)** Baseline median summer $overburden_\%$ minus the $k_s = 10^{-2}\,\mathrm{m}^{7/4}\,\mathrm{kg}^{-1/2}$ median summer $overburden_\%$. The same figure caption applies for Figures A3–A31

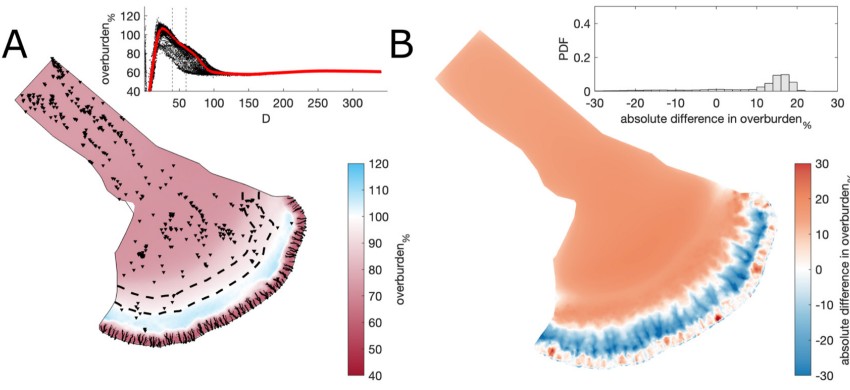

**Figure A4.** Comparison of the median summer system for sheet conductivity, $k_s = 10^{-3}\,\mathrm{m}^{7/4}\,\mathrm{kg}^{-1/2}$ against the baseline model run ($k_s = 10^{-4}\,\mathrm{m}^{7/4}\,\mathrm{kg}^{-1/2}$). The same figure caption as Figure A3 applies.

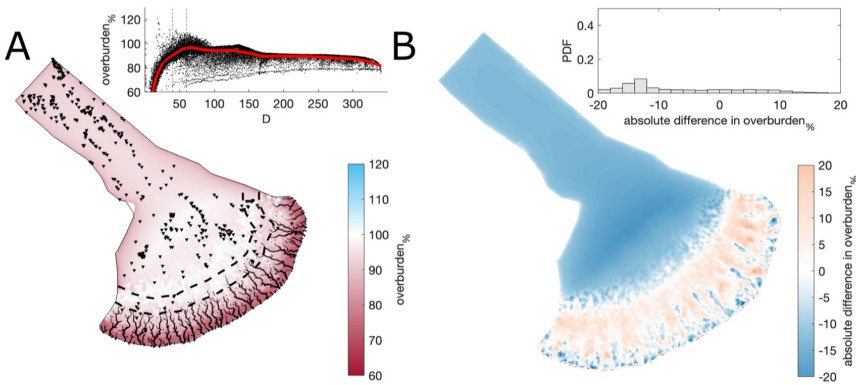

**Figure A5.** Comparison of the median summer system for sheet confuctivity, $k_s = 10^{-5} \, \mathrm{m}^{7/4} \, \mathrm{kg}^{-1/2}$ against the baseline model run ($k_s = 10^{-4} \, \mathrm{m}^{7/4} \, \mathrm{kg}^{-1/2}$). The same figure caption as Figure A3 applies.

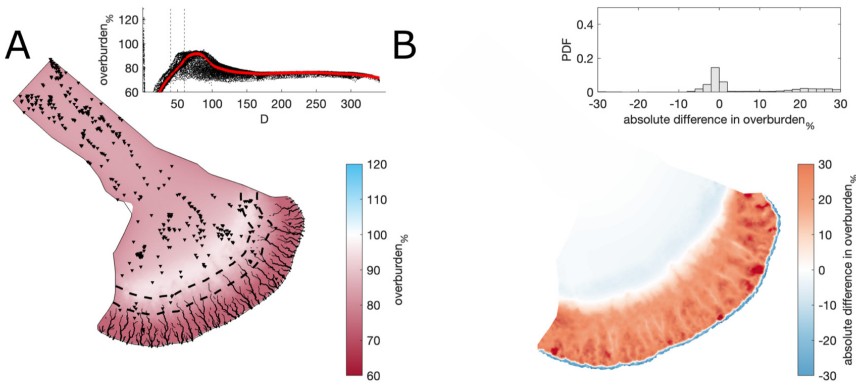

**Figure A6.** Comparison of the median summer system for channel conductivity, $k_c = 5 \times 10^{-1} \, \mathrm{m}^{3/2} \, \mathrm{kg}^{-1/2}$ against the baseline model run ($k_c = 10^{-1} \, \mathrm{m}^{3/2} \, \mathrm{kg}^{-1/2}$). The same figure caption as Figure A3 applies.

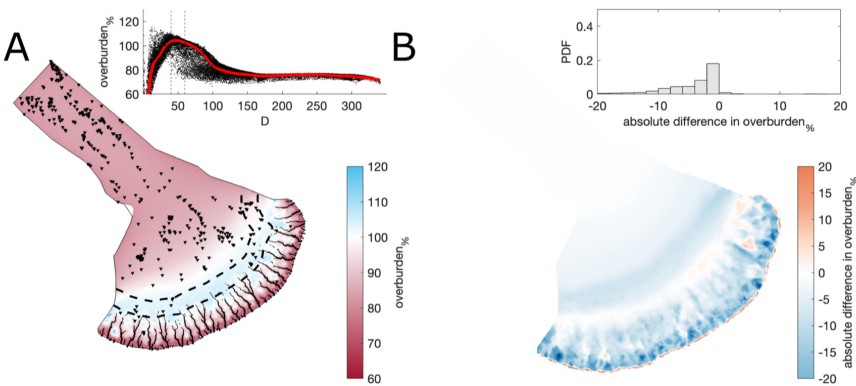

**Figure A7.** Comparison of the median summer system for channel conductivity, $k_c = 5 \times 10^{-2} \, \mathrm{m}^{3/2} \, \mathrm{kg}^{-1/2}$ against the baseline model run $(k_c = 10^{-1} \, \mathrm{m}^{3/2} \, \mathrm{kg}^{-1/2})$. The same figure caption as Figure A3 applies.

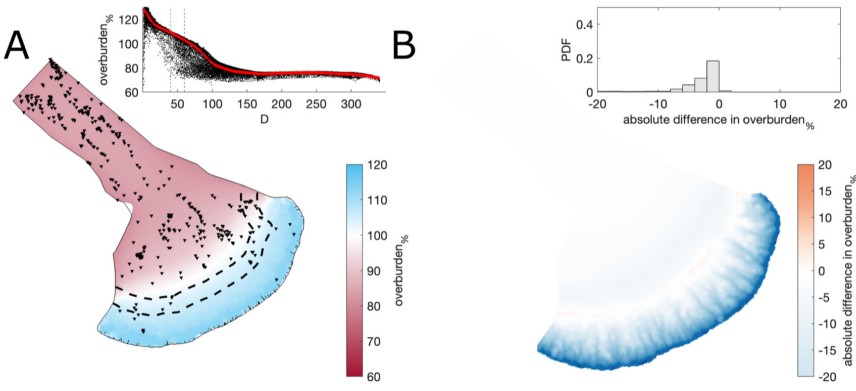

**Figure A8.** Comparison of the median summer system for channel conductivity, $k_c = 10^{-3} \, \mathrm{m}^{3/2} \, \mathrm{kg}^{-1/2}$ against the baseline model run $(k_c = 10^{-1} \, \mathrm{m}^{3/2} \, \mathrm{kg}^{-1/2})$. The same figure caption as Figure A3 applies.

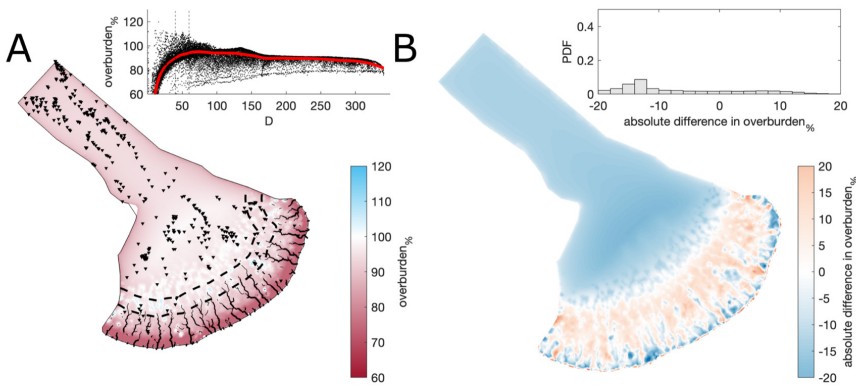

**Figure A9.** Comparison of the median summer system for moulin frequency, $N_{moulins} = 1000$ against the baseline model run ($N_{moulins} = 2500$). The same figure caption as Figure A3 applies.

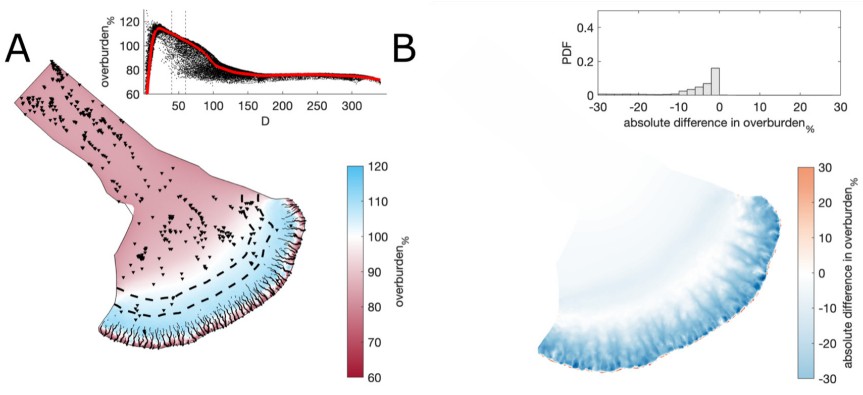

**Figure A10.** Comparison of the median summer system for moulin frequency, $N_{moulins} = 4000$ against the baseline model run ($N_{moulins} = 2500$). The same figure caption as Figure A3 applies.

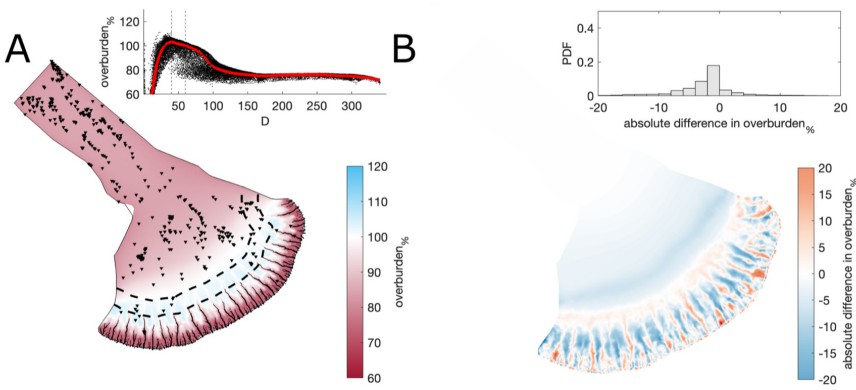

**Figure A11.** Comparison of the median summer system for where water was directly input at every nodes against the baseline model run ($N_{moulins} = 2500$). The same figure caption as Figure A3 applies.

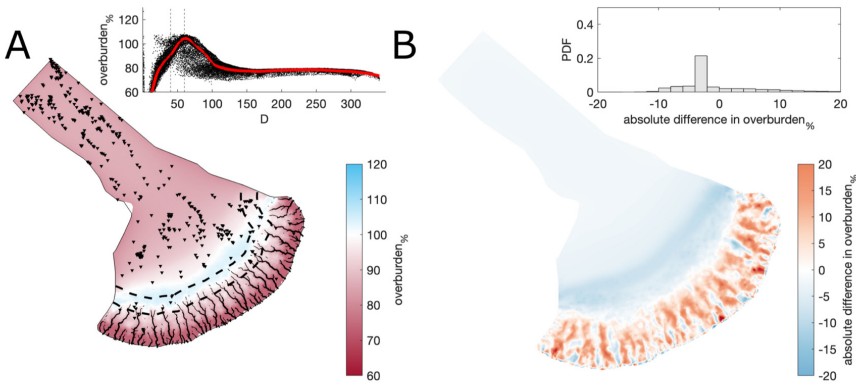

**Figure A12.** Comparison of the median summer system for a second random distribution of moulin frequency, $N_{moulins} = 2500$ against the baseline model run ($N_{moulins} = 2500$). The same figure caption as Figure A3 applies.

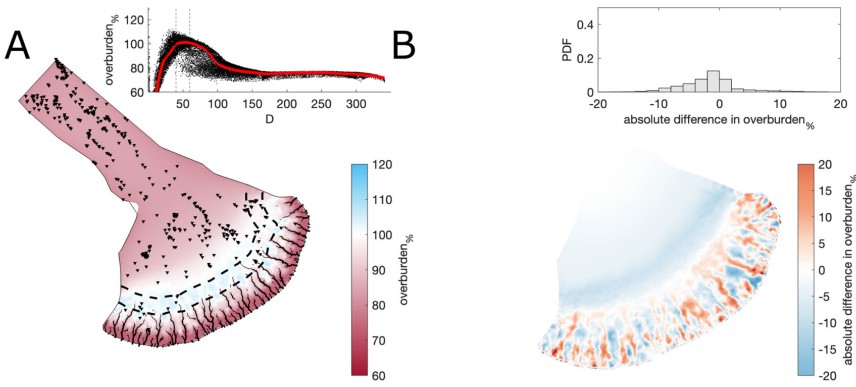

**Figure A13.** Comparison of the median summer system for a third random distribution of moulin frequency, $N_{moulins} = 2500$ against the baseline model run ($N_{moulins} = 2500$). The same figure caption as Figure A3 applies.

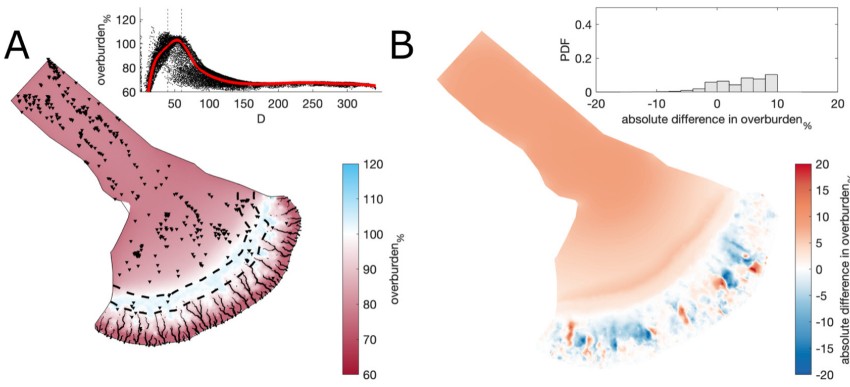

**Figure A14.** Comparison of the median summer system for basal melt rate, $b_{melt} = 1 \times 10^{-3}$ m yr$^{-1}$ against the baseline model run ($b_{melt} = 1 \times 10^{-3}$ m yr$^{-1}$). The same figure caption as Figure A3 applies.

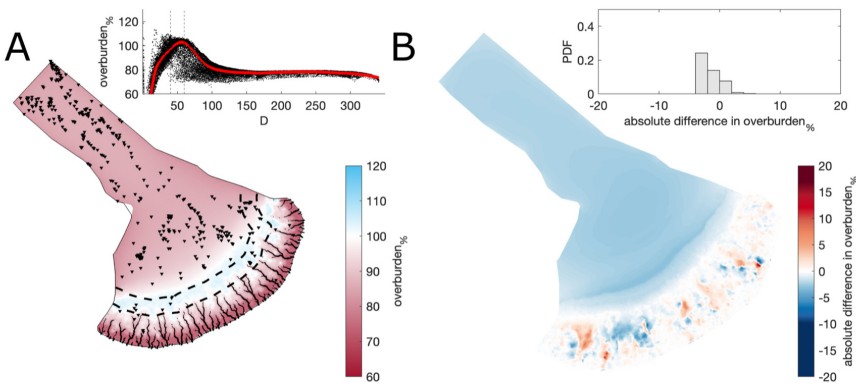

**Figure A15.** Comparison of the median summer system for basal melt rate, $b_{melt} = 1 \times 10^{-3}\mathrm{m\,yr}^{-1}$ against the baseline model run ($b_{melt} = 7 \times 10^{-3}\mathrm{m\,yr}^{-1}$). The same figure caption as Figure A3 applies.

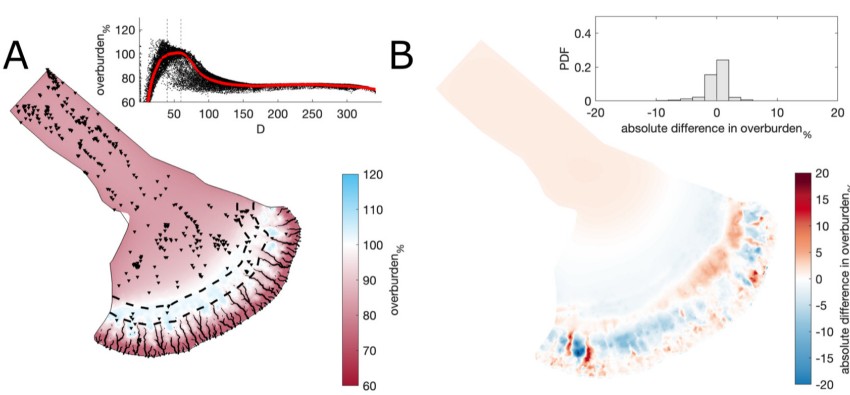

**Figure A16.** Comparison of the median summer system for basal bump height, $h_r = 0.1\,\mathrm{m}$ against the baseline model run ($h_r = 0.085\,\mathrm{m}$). The same figure caption as Figure A3 applies.

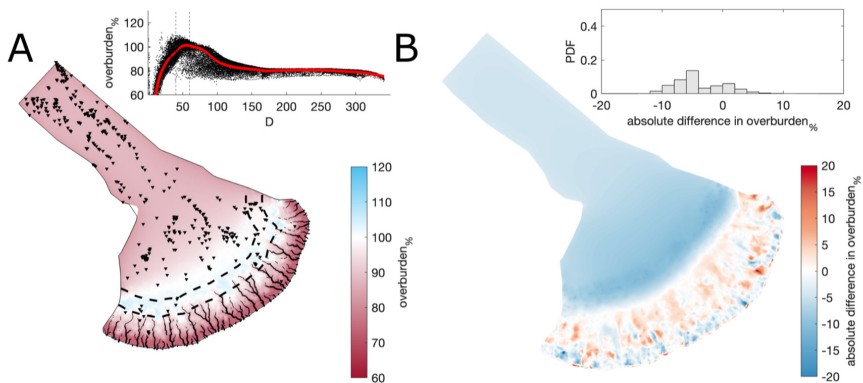

**Figure A17.** Comparison of the median summer system for basal bump height, $h_r = 0.05\,\mathrm{m}$ against the baseline model run ($h_r = 0.085\,\mathrm{m}$). The same figure caption as Figure A3 applies.

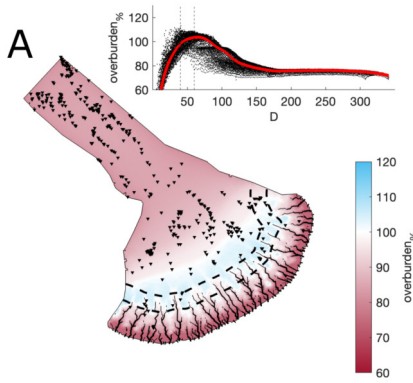

**Figure A18.** Comparison of a mesh that is not refined with respect to elevation against the baseline model run. The same figure caption as Figure A3 applies.

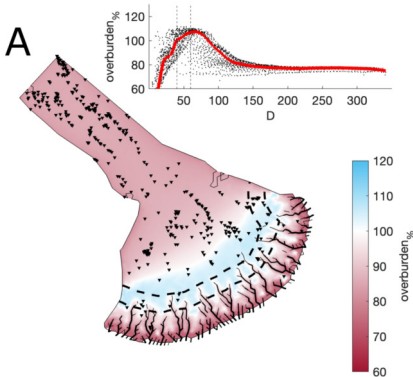

**Figure A19.** Comparison of a coarser mesh (edge length ∼5 km) against the baseline model run. The same figure caption as Figure A3 applies.

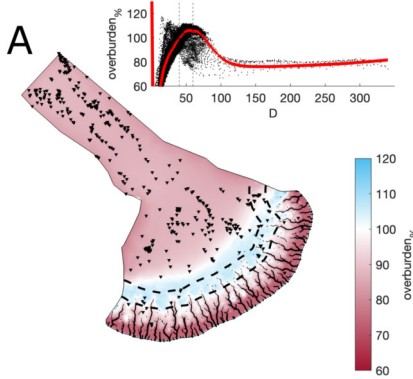

**Figure A20.** Comparison of a refined mesh (minimum edge length ≈300 m) <80 km from the ice margin against the baseline model run. The same figure caption as Figure A3 applies.

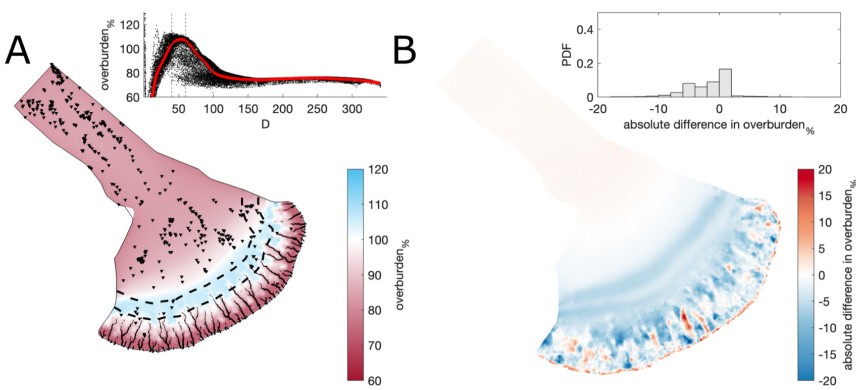

**Figure A21.** Comparison of a flat bed against the baseline model run. The same figure caption as Figure A3 applies.

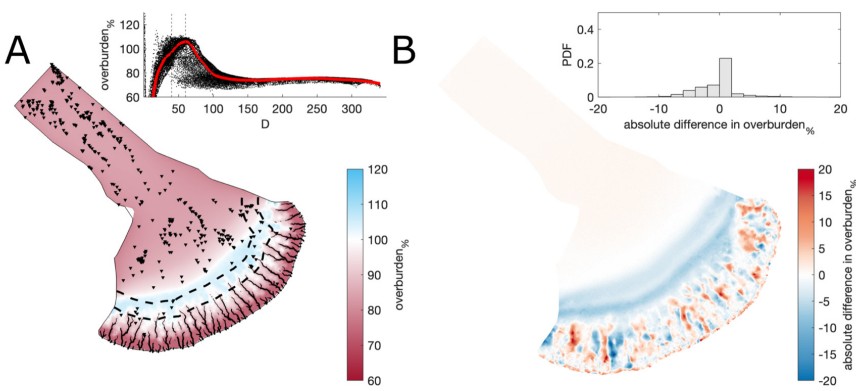

**Figure A22.** Comparison of a modern mesh (without subtracting Quaternary sediment thickness) against the baseline model run. The same figure caption as Figure A3 applies.

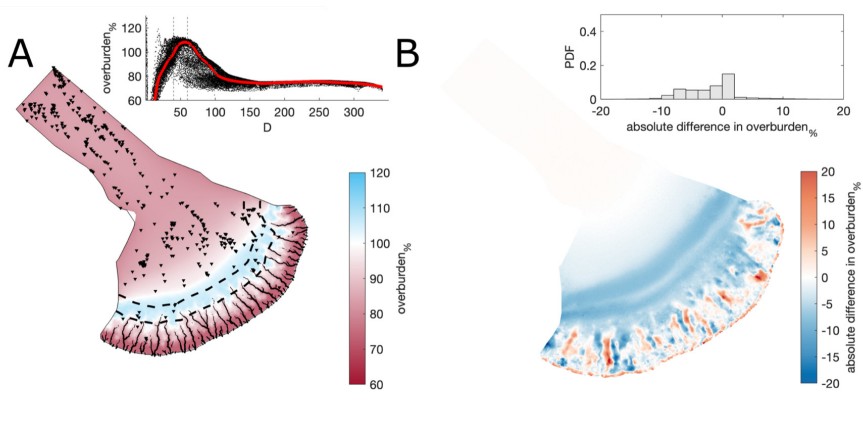

**Figure A23.** Comparison including lake bathymetry against the baseline model run. The same figure caption as Figure A3 applies.

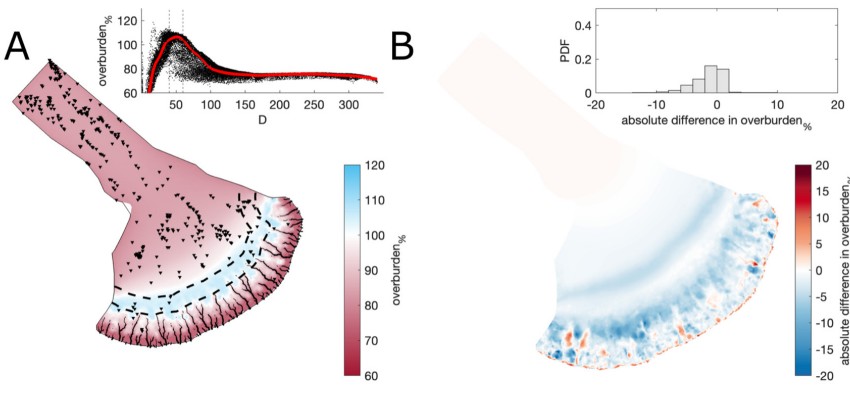

**Figure A24.** Comparison of a 30 m deep water body at the ice margin boundary against the baseline model run (land-terminating). The same figure caption as Figure A3 applies.

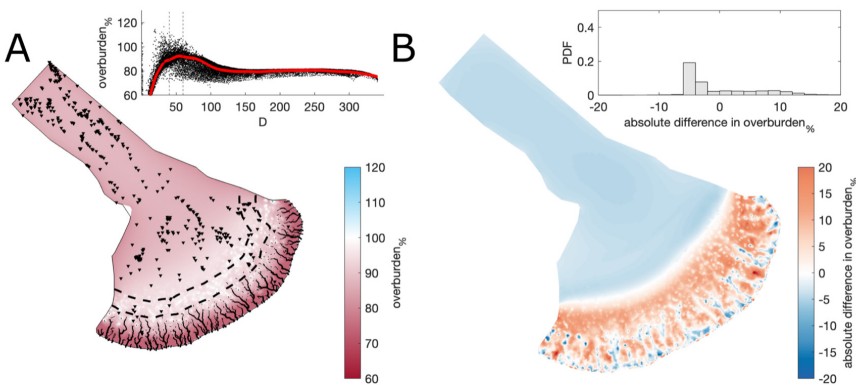

**Figure A25.** Comparison of the median summer system for an englacial void ratio, $E_{vr} = 10^{-3}$ against the baseline model run ($E_{vr} = 10^{-4}$). The same figure caption as Figure A3 applies.

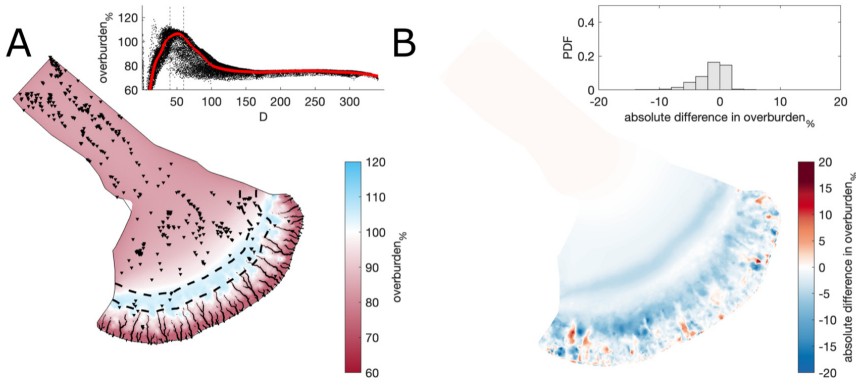

**Figure A26.** Comparison of the median summer system for an englacial void ratio, $E_{vr} = 10^{-5}$ against the baseline model run ($E_{vr} = 10^{-4}$). The same figure caption as Figure A3 applies.

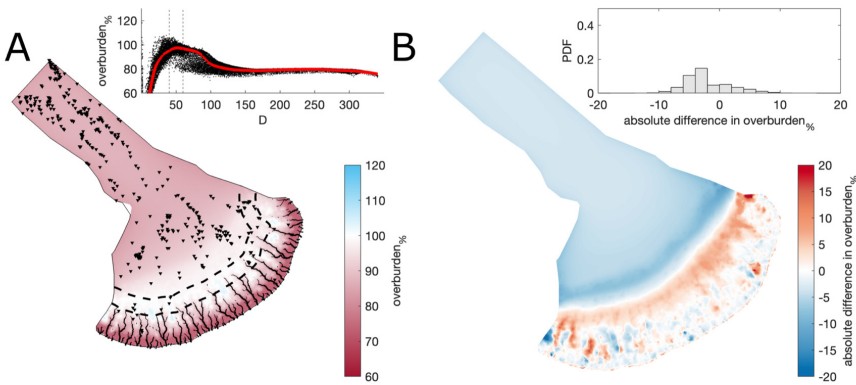

**Figure A27.** Comparison of the median summer system for a fixed velocity, $U_b = 100\,\text{m}\,\text{yr}^{-1}$ against the baseline model run ($U_b = 150\,\text{m}\,\text{yr}^{-1}$). The same figure caption as Figure A3 applies.

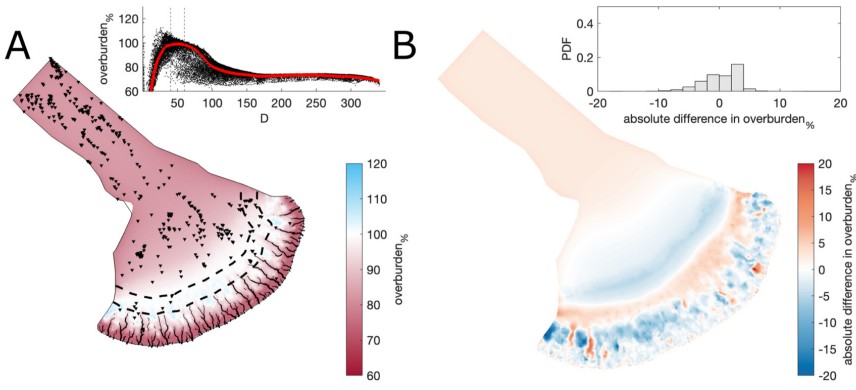

**Figure A28.** Comparison of the median summer system for a fixed velocity, $U_b = 200\,\text{m}\,\text{yr}^{-1}$ against the baseline model run ($U_b = 150\,\text{m}\,\text{yr}^{-1}$). The same figure caption as Figure A3 applies.

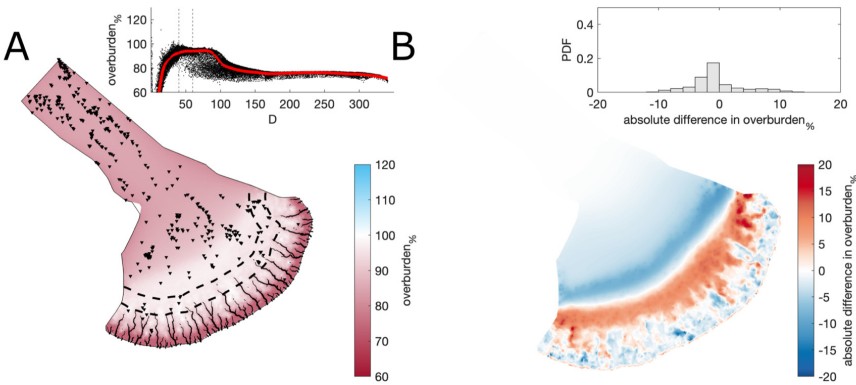

**Figure A29.** Comparison of the median summer system for a transient velocity, $U_b$ with a median $U_b = 150\,\mathrm{m\,yr^{-1}}$ against the fixed baseline model run ($U_b = 150\,\mathrm{m\,yr^{-1}}$). The same figure caption as Figure A3 applies.

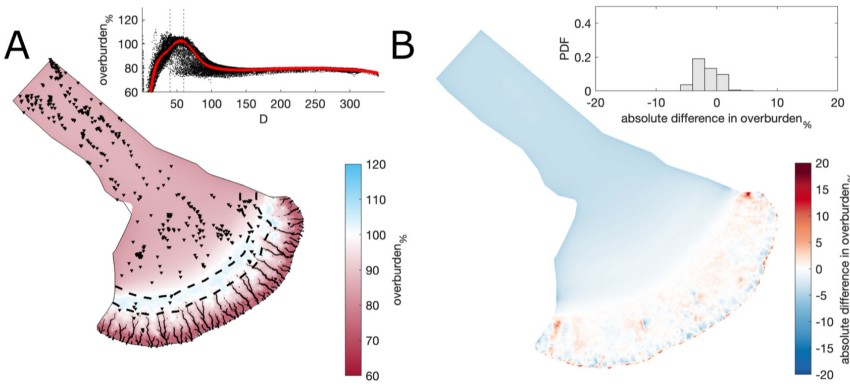

**Figure A30.** Comparison of the median summer system for a transient velocity, $U_b$ with a median $U_b = 100\,\mathrm{m\,yr^{-1}}$ against the fixed baseline model run ($U_b = 150\,\mathrm{m\,yr^{-1}}$). The same figure caption as Figure A3 applies.

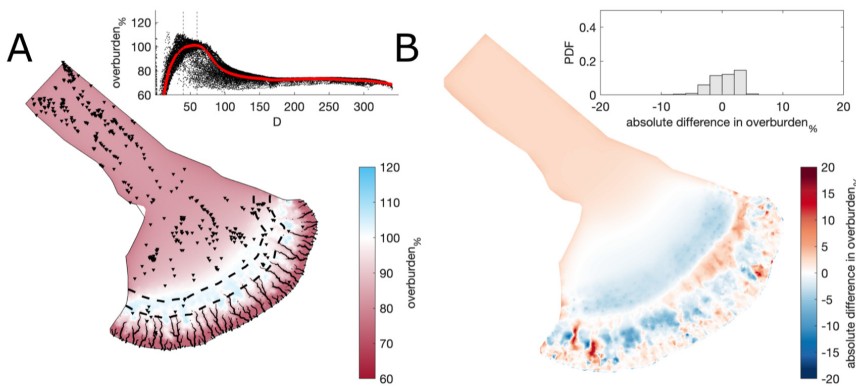

**Figure A31.** Comparison of the median summer system for a transient velocity, $U_b$ with a median $U_b = 200 \, \mathrm{m \, yr^{-1}}$ against the fixed baseline model run ($U_b = 150 \, \mathrm{m \, yr^{-1}}$). The same figure caption as Figure A3 applies.

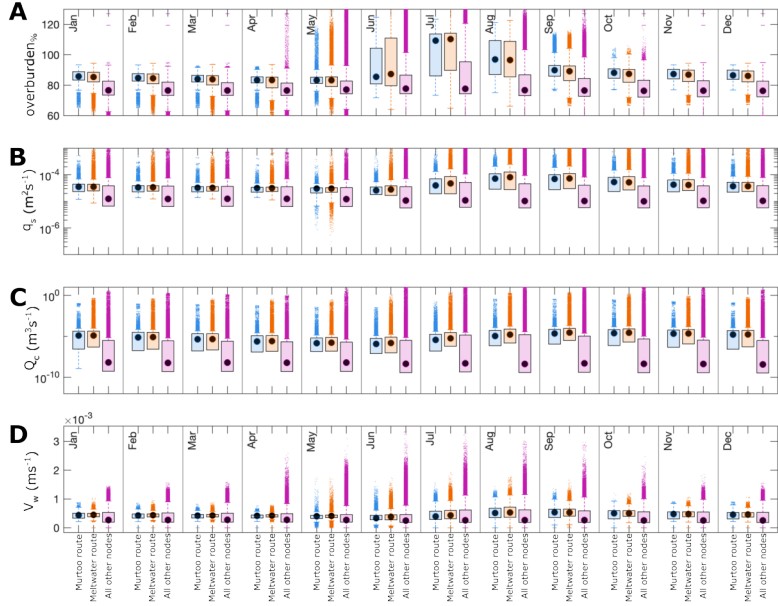

**Figure A32.** Boxplots of model parameters grouped by month for overburden ($overburden_\%$, **A**), sheet discharge ($q_s$, **B**), water velocity ($V_W$, **C**), and channel discharge ($Q_c$, **D**) during all model years at nodes between 40–60 km from the ice margin. As in Figure 5, nodes that fall within meltwater routes which do host murtoos (Murtoo free MRs) are shown in blue, nodes which fall within mapped meltwater routes that do not contain murtoo fields (Murtoo hosting MRs) are shown in orange, and all other nodes are shown in purple. Medians for each group are shown as black circles, and 'outliers'—defined as points more than 150% of the interquartile range away from the upper and lower quartile—are shown as crosses.

**Table A1.** Tukey-Kramer HSD test of $overburden_\%$ in meltwater routes, murtoo routes, and non-meltwater routes between 40–60 km from the ice margin. The upper and lower limits describe the 95% confidence intervals for the true mean difference, A-B is the difference between group means.

| Month | Group A | Group B | Lower Limit | A-B | Upper limit | P-Value |
|---|---|---|---|---|---|---|
| | meltwater route | murtoo route | -1.71 | -1.30 | -0.89 | 0.00 |
| January | all other nodes | murtoo route | -10.49 | -10.12 | -9.75 | 0.00 |
| | meltwater route | all other nodes | 8.58 | 8.82 | 9.06 | 0.00 |
| | meltwater route | murtoo route | -1.62 | -1.21 | -0.80 | 0.00 |
| February | all other nodes | murtoo route | -9.87 | -9.50 | -9.12 | 0.00 |
| | meltwater route | all other nodes | 8.05 | 8.29 | 8.52 | 0.00 |
| | meltwater route | murtoo route | -1.58 | -1.17 | -0.76 | 0.00 |
| March | all other nodes | murtoo route | -9.30 | -8.93 | -8.55 | 0.00 |
| | meltwater route | all other nodes | 7.52 | 7.75 | 7.99 | 0.00 |
| | meltwater route | murtoo route | -1.55 | -1.14 | -0.73 | 0.00 |
| April | all other nodes | murtoo route | -8.33 | -7.96 | -7.58 | 0.00 |
| | meltwater route | all other nodes | 6.58 | 6.82 | 7.06 | 0.00 |
| | meltwater route | murtoo route | -1.10 | -0.69 | -0.28 | 0.00 |
| May | all other nodes | murtoo route | -5.26 | -4.89 | -4.51 | 0.00 |
| | meltwater route | all other nodes | 3.96 | 4.20 | 4.43 | 0.00 |
| | meltwater route | murtoo route | 0.70 | 1.12 | 1.54 | 0.00 |
| June | all other nodes | murtoo route | -8.56 | -8.17 | -7.79 | 0.00 |
| | meltwater route | all other nodes | 9.05 | 9.30 | 9.54 | 0.00 |
| | meltwater route | murtoo route | 0.10 | 0.52 | 0.93 | 0.00 |
| July | all other nodes | murtoo route | -18.72 | -18.34 | -17.96 | 0.00 |
| | meltwater route | all other nodes | 18.61 | 18.85 | 19.10 | 0.00 |
| | meltwater route | murtoo route | -2.01 | -1.59 | -1.18 | 0.00 |
| August | all other nodes | murtoo route | -18.81 | -18.43 | -18.04 | 0.00 |
| | meltwater route | all other nodes | 16.59 | 16.83 | 17.08 | 0.00 |
| | meltwater route | murtoo route | -2.30 | -1.88 | -1.47 | 0.00 |
| September | all other nodes | murtoo route | -14.47 | -14.09 | -13.71 | 0.00 |
| | meltwater route | all other nodes | 11.97 | 12.21 | 12.45 | 0.00 |
| | meltwater route | murtoo route | -2.14 | -1.72 | -1.31 | 0.00 |
| October | all other nodes | murtoo route | -12.49 | -12.11 | -11.73 | 0.00 |
| | meltwater route | all other nodes | 10.15 | 10.39 | 10.63 | 0.00 |
| | meltwater route | murtoo route | -1.99 | -1.57 | -1.15 | 0.00 |
| November | all other nodes | murtoo route | -12.06 | -11.68 | -11.30 | 0.00 |
| | meltwater route | all other nodes | 9.87 | 10.11 | 10.36 | 0.00 |
| | meltwater route | murtoo route | -1.88 | -1.46 | -1.04 | 0.00 |
| December | all other nodes | murtoo route | -11.56 | -11.17 | -10.79 | 0.00 |
| | meltwater route | all other nodes | 9.47 | 9.72 | 9.96 | 0.00 |

**Table A2.** Tukey-Kramer HSD test of $q_s$ in meltwater routes, murtoo routes, and non-meltwater routes between 40–60 km from the ice margin. The upper and lower limits describe the 95% confidence intervals for the true mean difference, A-B is the difference between group means.

| Month | Group A | Group B | Lower Limit | A-B | Upper limit | P-Value |
|---|---|---|---|---|---|---|
| | meltwater route | murtoo route | $-5.07\times10^{-7}$ | $2.89\times10^{-6}$ | $6.28\times10^{-6}$ | 0.27 |
| January | all other nodes | murtoo route | $-1.57\times10^{-5}$ | $-1.26\times10^{-5}$ | $-9.54\times10^{-7}$ | 0.00 |
| | meltwater route | all other nodes | $1.36\times10^{-5}$ | $1.55\times10^{-5}$ | $1.75\times10^{-5}$ | 0.00 |
| | meltwater route | murtoo route | $-1.23\times10^{-6}$ | $2.19\times10^{-6}$ | $5.6\times10^{-6}$ | 0.88 |
| February | all other nodes | murtoo route | $-1.39\times10^{-5}$ | $-1.08\times10^{-5}$ | $-7.69\times10^{-6}$ | 0.00 |
| | meltwater route | all other nodes | $1.1\times10^{-5}$ | $1.3\times10^{-5}$ | $1.5\times10^{-5}$ | 0.00 |
| | meltwater route | murtoo route | $-2.05\times10^{-6}$ | $1.36\times10^{-6}$ | $4.77\times10^{-6}$ | 0.99 |
| March | all other nodes | murtoo route | $-1.26\times10^{-5}$ | $-9.49\times10^{-6}$ | $-6.37\times10^{-6}$ | 0.00 |
| | meltwater route | all other nodes | $8.86\times10^{-6}$ | $1.08\times10^{-5}$ | $1.28\times10^{-5}$ | 0.00 |
| | meltwater route | murtoo route | $-2.66\times10^{-6}$ | $7.6\times10^{-7}$ | $4.18\times10^{-6}$ | 0.99 |
| April | all other nodes | murtoo route | $-1.14\times10^{-5}$ | $-8.28\times10^{-6}$ | $-5.15\times10^{-6}$ | 0.00 |
| | meltwater route | all other nodes | $7.05\times10^{-6}$ | $9.04\times10^{-6}$ | $1.10\times10^{-5}$ | 0.00 |
| | meltwater route | murtoo route | $-2.92\times10^{-6}$ | $4.99\times10^{-7}$ | $3.92\times10^{-6}$ | 0.99 |
| May | all other nodes | murtoo route | $-7.11\times10^{-6}$ | $-3.98\times10^{-6}$ | $-8.53\times10^{-5}$ | 0.00 |
| | meltwater route | all other nodes | $2.49\times10^{-6}$ | $4.48\times10^{-6}$ | $6.47\times10^{-6}$ | 0.00 |
| | meltwater route | murtoo route | $9.57\times10^{-7}$ | $4.46\times10^{-6}$ | $7.97\times10^{-6}$ | 0.00 |
| June | all other nodes | murtoo route | $6.04\times10^{-6}$ | $9.25\times10^{-6}$ | $1.25\times10^{-5}$ | 0.00 |
| | meltwater route | all other nodes | $-6.83\times10^{-6}$ | $-4.79\times10^{-6}$ | $-2.75\times10^{-6}$ | 0.00 |
| | meltwater route | murtoo route | $1.36\times10^{-5}$ | $1.7\times10^{-5}$ | $2.05\times10^{-5}$ | 0.00 |
| July | all other nodes | murtoo route | $9.95\times10^{-6}$ | $1.31\times10^{-5}$ | $1.63\times10^{-5}$ | 0.00 |
| | meltwater route | all other nodes | $1.89\times10^{-6}$ | $3.9\times10^{-6}$ | $5.91\times10^{-6}$ | 0.00 |
| | meltwater route | murtoo route | $1.82\times10^{-5}$ | $2.17\times10^{-5}$ | $2.52\times10^{-5}$ | 0.00 |
| August | all other nodes | murtoo route | $-9.99\times10^{-6}$ | $-6.79\times10^{-6}$ | $-3.59\times10^{-6}$ | 0.00 |
| | meltwater route | all other nodes | $2.65\times10^{-5}$ | $2.85\times10^{-5}$ | $3.05\times10^{-5}$ | 0.00 |
| | meltwater route | murtoo route | $1.06\times10^{-5}$ | $1.41\times10^{-5}$ | $1.75\times10^{-5}$ | 0.00 |
| September | all other nodes | murtoo route | $-2.33\times10^{-5}$ | $-2.02\times10^{-5}$ | $1.7\times10^{-5}$ | 0.00 |
| | meltwater route | all other nodes | $3.22\times10^{-5}$ | $3.42\times10^{-5}$ | $3.62\times10^{-5}$ | 0.00 |
| | meltwater route | murtoo route | $5.67\times10^{-6}$ | $9.15\times10^{-6}$ | $1.26\times10^{-5}$ | 0.00 |
| October | all other nodes | murtoo route | $-2.44\times10^{-5}$ | $-2.12\times10^{-5}$ | $-1.8\times10^{-5}$ | 0.00 |
| | meltwater route | all other nodes | $2.84\times10^{-5}$ | $3.04\times10^{-5}$ | $3.24\times10^{-5}$ | 0.00 |
| | meltwater route | murtoo route | $2.52\times10^{-6}$ | $6\times10^{-6}$ | $9.48\times10^{-6}$ | 0.00 |
| November | all other nodes | murtoo route | $-2.14\times10^{-5}$ | $-1.82\times10^{-5}$ | $-1.50\times10^{-5}$ | 0.00 |
| | meltwater route | all other nodes | $2.22\times10^{-5}$ | $2.42\times10^{-5}$ | $2.62\times10^{-5}$ | 0.00 |
| | meltwater route | murtoo route | $7.23\times10^{-7}$ | $4.22\times10^{-6}$ | $7.71\times10^{-6}$ | 0.00 |
| December | all other nodes | murtoo route | $-1.87\times10^{-5}$ | $-1.55\times10^{-5}$ | $1.23\times10^{-5}$ | 0.00 |
| | meltwater route | all other nodes | $1.77\times10^{-5}$ | $1.97\times10^{-5}$ | $2.18\times10^{-5}$ | 0.00 |

**Table A3.** Tukey-Kramer HSD test of $Q_c$ in meltwater routes, murtoo routes, and non-meltwater routes between 40–60 km from the ice margin. The upper and lower limits describe the 95% confidence intervals for the true mean difference, A-B is the difference between group means.

| Month | Group A | Group B | Lower Limit | A-B | Upper limit | P-Value |
|-------|---------|---------|-------------|-----|-------------|---------|
| | meltwater route | murtoo route | $-7.57\times10^{-3}$ | $6.51\times10^{-4}$ | $8.87\times10^{-3}$ | 0.99 |
| January | all other nodes | murtoo route | $-6.19\times10^{-3}$ | $1.34\times10^{-3}$ | $8.86\times10^{-3}$ | 0.99 |
| | meltwater route | all other nodes | $-5.47\times10^{-3}$ | $-6.86\times10^{-4}$ | $4.1\times10^{-3}$ | 0.99 |
| | meltwater route | murtoo route | $-7.78\times10^{-3}$ | $5.08\times10^{-4}$ | $8.79\times10^{-3}$ | 0.99 |
| February | all other nodes | murtoo route | $-6.72\times10^{-3}$ | $5.08\times10^{-4}$ | $8.44\times10^{-3}$ | 0.99 |
| | meltwater route | all other nodes | $-5.18\times10^{-3}$ | $-3.53\times10^{-4}$ | $4.47\times10^{-3}$ | 0.99 |
| | meltwater route | murtoo route | $-7.86\times10^{-3}$ | $4.06\times10^{-4}$ | $8.677\times10^{-3}$ | 0.99 |
| March | all other nodes | murtoo route | $-6.93\times10^{-3}$ | $6.3\times10^{-4}$ | $8.19\times10^{-3}$ | 0.99 |
| | meltwater route | all other nodes | $-5.04\times10^{-3}$ | $-2.24\times10^{-4}$ | $4.59\times10^{-5}$ | 0.99 |
| | meltwater route | murtoo route | $-7.98\times10^{-3}$ | $3.07\times10^{-4}$ | $8.59\times10^{-3}$ | 0.99 |
| April | all other nodes | murtoo route | $-7.10\times10^{-3}$ | $4.83\times10^{-4}$ | $8.06\times10^{-3}$ | 0.99 |
| | meltwater route | all other nodes | $-5\times10^{-3}$ | $-1.76\times10^{-4}$ | $4.65\times10^{-3}$ | 0.99 |
| | meltwater route | murtoo route | $-8,02\times10^{-3}$ | $2,60\times10^{-4}$ | $8,54\times10^{-3}$ | 0.99 |
| May | all other nodes | murtoo route | $-5,58\times10^{-3}$ | $2,01\times10^{-3}$ | $9,59\times10^{-3}$ | 0.99 |
| | meltwater route | all other nodes | $-6,57\times10^{-3}$ | $-1,75\times10^{-3}$ | $3,08\times10^{-3}$ | 0.99 |
| | meltwater route | murtoo route | $-7,93\times10^{-3}$ | $5,65\times10^{-4}$ | $9,06\times10^{-3}$ | 0.99 |
| June | all other nodes | murtoo route | $7,28\times10^{-3}$ | $1,51\times10^{-2}$ | $2,28\times10^{-2}$ | 0.00 |
| | meltwater route | all other nodes | $-1,94\times10^{-2}$ | $-1,45\times10^{-2}$ | $-9,54\times10^{-3}$ | 0.00 |
| | meltwater route | murtoo route | $-6,22\times10^{-3}$ | $2,14\times10^{-3}$ | $1,05\times10^{-2}$ | 0.99 |
| July | all other nodes | murtoo route | $3,20\times10^{-2}$ | $3,97\times10^{-2}$ | $4,73\times10^{-2}$ | 0.00 |
| | meltwater route | all other nodes | $-4,24\times10^{-2}$ | $-3,75\times10^{-2}$ | $-3,27\times10^{-2}$ | 0.00 |
| | meltwater route | murtoo route | $-4,45\times10^{-3}$ | $4,02\times10^{-3}$ | $1,25\times10^{-2}$ | 0.99 |
| August | all other nodes | murtoo route | $3,97\times10^{-2}$ | $4,74\times10^{-2}$ | $5,52\times10^{-2}$ | 0.00 |
| | meltwater route | all other nodes | $-4,84\times10^{-2}$ | $-4,34\times10^{-2}$ | $-3,85\times10^{-2}$ | 0.00 |
| | meltwater route | murtoo route | $-4,64\times10^{-3}$ | $3,75\times10^{-3}$ | $1,21\times10^{-2}$ | 0.99 |
| September | all other nodes | murtoo route | $2,24\times10^{-2}$ | $3,01\times10^{-2}$ | $3,78\times10^{-2}$ | 0.00 |
| | meltwater route | all other nodes | $-3,12\times10^{-2}$ | $-2,63\times10^{-2}$ | $-2,15\times10^{-2}$ | 0.00 |
| | meltwater route | murtoo route | $-6,24\times10^{-3}$ | $2,19\times10^{-3}$ | $1,06\times10^{-2}$ | 0.99 |
| October | all other nodes | murtoo route | $2,22\times10^{-3}$ | $9,94\times10^{-3}$ | $1,77\times10^{-2}$ | 0.00 |
| | meltwater route | all other nodes | $-1,27\times10^{-2}$ | $-7,75\times10^{-3}$ | $-2,84\times10^{-3}$ | 0.00 |
| | meltwater route | murtoo route | $-7,16\times10^{-3}$ | $1,27\times10^{-3}$ | $9,70\times10^{-3}$ | 0.99 |
| November | all other nodes | murtoo route | $-4,08\times10^{-3}$ | $3,63\times10^{-3}$ | $1,13\times10^{-2}$ | 0.99 |
| | meltwater route | all other nodes | $-7,27\times10^{-3}$ | $-2,36\times10^{-3}$ | $2,55\times10^{-3}$ | 0.99 |
| | meltwater route | murtoo route | $-7,56\times10^{-3}$ | $9,10\times10^{-4}$ | $9,38\times10^{-3}$ | 0.99 |
| December | all other nodes | murtoo route | $-5,57\times10^{-3}$ | $2,18\times10^{-3}$ | $9,94\times10^{-3}$ | 0.99 |
| | meltwater route | all other nodes | $-6,21\times10^{-3}$ | $-1,27\times10^{-3}$ | $3,66\times10^{-3}$ | 0.99 |

**Table A4.** Tukey-Kramer HSD test of $V_W$ in meltwater routes, murtoo routes, and non-meltwater routes between 40–60 km from the ice margin. The upper and lower limits describe the 95% confidence intervals for the true mean difference, A-B is the difference between group means.

| Month | Group A | Group B | Lower Limit | A-B | Upper limit | P-Value |
|---|---|---|---|---|---|---|
| | meltwater route | murtoo route | $1.41\times10^{-6}$ | $7.42\times10^{-6}$ | $1.34\times10^{-5}$ | 0.00 |
| January | all other nodes | murtoo route | $-5.63\times10^{-5}$ | $-5.08\times10^{-5}$ | $-4.53\times10^{-5}$ | 0.00 |
| | meltwater route | all other nodes | $5.47\times10^{-5}$ | $5.82\times10^{-5}$ | $6.17\times10^{-5}$ | 0.00 |
| | meltwater route | murtoo route | $1.11\times10^{-6}$ | $7.17\times10^{-6}$ | $1.32\times10^{-5}$ | 0.00 |
| Febuary | all other nodes | murtoo route | $-4.63\times10^{-5}$ | $-4.08\times10^{-5}$ | $-3.53\times10^{-5}$ | 0.00 |
| | meltwater route | all other nodes | $4.44\times10^{-5}$ | $4.80\times10^{-5}$ | $5.15\times10^{-5}$ | 0.00 |
| | meltwater route | murtoo route | $7.90\times10^{-7}$ | $6.83\times10^{-6}$ | $1.29\times10^{-5}$ | 0.01 |
| March | all other nodes | murtoo route | $-3.88\times10^{-5}$ | $-3.32\times10^{-5}$ | $-2.77\times10^{-5}$ | 0.00 |
| | meltwater route | all other nodes | $3.65\times10^{-5}$ | $4.01\times10^{-5}$ | $4.36\times10^{-5}$ | 0.00 |
| | meltwater route | murtoo route | $1.06\times10^{-6}$ | $7.11\times10^{-6}$ | $1.32\times10^{-5}$ | 0.00 |
| April | all other nodes | murtoo route | $-3.26\times10^{-5}$ | $-2.71\times10^{-5}$ | $-2.16\times10^{-5}$ | 0.00 |
| | meltwater route | all other nodes | $3.07\times10^{-5}$ | $3.42\times10^{-5}$ | $3.77\times10^{-5}$ | 0.00 |
| | meltwater route | murtoo route | $1.73\times10^{-6}$ | $7.78\times10^{-6}$ | $1.38\times10^{-5}$ | 0.00 |
| May | all other nodes | murtoo route | $-6.47\times10^{-6}$ | $-9.33\times10^{-7}$ | $4.61\times10^{-6}$ | 1.00 |
| | meltwater route | all other nodes | $5.19\times10^{-6}$ | $8.72\times10^{-6}$ | $1.22\times10^{-5}$ | 0.00 |
| | meltwater route | murtoo route | $1.22\times10^{-5}$ | $1.84\times10^{-5}$ | $2.46\times10^{-5}$ | 0.00 |
| June | all other nodes | murtoo route | $4.12\times10^{-5}$ | $4.69\times10^{-5}$ | $5.26\times10^{-5}$ | 0.00 |
| | meltwater route | all other nodes | $-3.21\times10^{-5}$ | $-2.85\times10^{-5}$ | $-2.48\times10^{-5}$ | 0.00 |
| | meltwater route | murtoo route | $2.47\times10^{-5}$ | $3.09\times10^{-5}$ | $3.70\times10^{-5}$ | 0.00 |
| July | all other nodes | murtoo route | $7.34\times10^{-7}$ | $6.33\times10^{-6}$ | $1.19\times10^{-5}$ | 0.01 |
| | meltwater route | all other nodes | $2.10\times10^{-5}$ | $2.45\times10^{-5}$ | $2.81\times10^{-5}$ | 0.00 |
| | meltwater route | murtoo route | $1.46\times10^{-5}$ | $2.08\times10^{-5}$ | $2.70\times10^{-5}$ | 0.00 |
| August | all other nodes | murtoo route | $-1.05\times10^{-4}$ | $-9.90\times10^{-5}$ | $-9.34\times10^{-5}$ | 0.00 |
| | meltwater route | all other nodes | $1.16\times10^{-4}$ | $1.20\times10^{-4}$ | $1.23\times10^{-4}$ | 0.00 |
| | meltwater route | murtoo route | $2.52\times10^{-6}$ | $8.65\times10^{-6}$ | $1.48\times10^{-5}$ | 0.00 |
| September | all other nodes | murtoo route | $-1.19\times10^{-4}$ | $-1.14\times10^{-4}$ | $-1.08\times10^{-4}$ | 0.00 |
| | meltwater route | all other nodes | $1.19\times10^{-4}$ | $1.22\times10^{-4}$ | $1.26\times10^{-4}$ | 0.00 |
| | meltwater route | murtoo route | $8.15\times10^{-6}$ | $1.43\times10^{-5}$ | $2.05\times10^{-5}$ | 0.00 |
| October | all other nodes | murtoo route | $-9.90\times10^{-5}$ | $-9.34\times10^{-5}$ | $-8.77\times10^{-5}$ | 0.00 |
| | meltwater route | all other nodes | $1.04\times10^{-4}$ | $1.08\times10^{-4}$ | $1.11\times10^{-4}$ | 0.00 |
| | meltwater route | murtoo route | $6.68\times10^{-6}$ | $1.28\times10^{-5}$ | $1.90\times10^{-5}$ | 0.00 |
| November | all other nodes | murtoo route | $-8.03\times10^{-5}$ | $-7.46\times10^{-5}$ | $-6.90\times10^{-5}$ | 0.00 |
| | meltwater route | all other nodes | $8.39\times10^{-5}$ | $8.75\times10^{-5}$ | $9.10\times10^{-5}$ | 0.00 |
| | meltwater route | murtoo route | $3.95\times10^{-6}$ | $1.01\times10^{-5}$ | $1.63\times10^{-5}$ | 0.00 |
| December | all other nodes | murtoo route | $-6.79\times10^{-5}$ | $-6.22\times10^{-5}$ | $-5.65\times10^{-5}$ | 0.00 |
| | meltwater route | all other nodes | $6.87\times10^{-5}$ | $7.23\times10^{-5}$ | $7.59\times10^{-5}$ | 0.00 |