# Peer review of "Reorganisation of subglacial drainage processes during rapid melting of the Fennoscandian Ice Sheet"

_EGUsphere, 2023_

## Referee Comment (RC1)

**Review of "Reorganisation of subglacial drainage processes during rapid melting of the Fennoscandian Ice Sheet"**

**Letter**

Dear Editor,

This manuscript aims to validate or verify a sophisticated subglacial hydrology model with the presences and absence of landforms known as "murtoos" in a paleo-ice sheet setting. The basis of this is that murtoos form under specific subglacial conditions.

I personally found this paper inspiring in that it integrated numerical modeling with observational data of sedimentary structures, using glaciological and sedimentary knowledge together. Also, I found the experiments well designed, discussion detailed and paper well written, potentially making a substantial contribution.

Despite this positive assessment, there are several questions and matters that I believe that should be addressed before publication. These matters are presented below.

Hopefully the authors find the review useful.

**General comments**

- I found the title a bit misleading about the topic of the paper. I think the manuscript speaks to the past location of hydraulic conditions below the glacier, but I found little in the manuscript about "reorganisation of subgalacial drainage processes". Maybe "organisation of subglacial processes." Additionally, "rapid melting" does not seem like a big part of the paper, especially by reading the abstract.

- Throughout, but especially in the abstract and introduction, the authors make statements of "parameterizing and testing models of subglacial hydrology", "basal hydrology in models", or "basal hydraulic conditions." These conditions or parameterizations can cover a wide range of features describing subglacial processes, include channel size or shape, water pressure, sediment transport, water velocity, distributed or channelized drainage. I believe the authors must be more specific and deliberate in describing the specific subglacial hydraulic features they aim to examine and link these features to murtoo development and persistence.

- Related to the last point, it seems that the description of murtoo formation could be improved. At times, it seems that hydraulic processes associated with different stages of murtoo development are in contradiction. An examples are given below.

- Section 3.1.1: From my reading of this section, it seems that no diurnal forcings were used. While this makes sense in a paleo setting, I am concerned about the impact on results. For instance on the GrIS, hydraulic head can very over 150m and bed separation can be in excess of 25cm (Andrews et al., 2014). It seems like such short temporal changes in subglacial hydrology could impact the formation of murtoos and move from one murtoo sequence to another over a very short time period (stages mentioned in Introduction, Hovikoski et al., 2023). I realize that application of such variable water discharges to hydraulic models can be difficult and in many scenarios not necessary. However, it seems like it could be important in this application.

- To the best of my knowledge GlaDS uses a semicircular channel geometry that is fixed (i.e. shape of the channel does not evolve). However, it seems that a key feature of murtoo development is low

broad channels, potentially with changing channel shape. This seems to be discussed in Hovikoski et al., 2023 and in the manuscript at lines 511 to 523. Hooke et al., 1990 speaks to the effects of channel shape on subglacial hydraulics. I am aware that certain trade offs can be made between the friction factor and channel shape to end up with similar hydraulic characteristics. In some applications this may minimize the impact of the semicircular assumption. However, because sediment transport relationships are scaled to unit width of the channel, sediment deposition can be sensitive to the width of the channel floor, and thus the general shape of the channel. However, please comment on how this may impact the results. Is this such a consideration with the development of the drainage system? What are the impacts of the semi-circular and potentially fixed channel shape on the formation of murtoos?

- More out of curiosity, how do the murtoo fields persist given the retreat of the glacier and the presumptive movement of the channelized drainage area up the glacier? Might retreat have occurred too rapidly to "destroy" the murtoos?

**Specific comments**

- Ln 46: to the best of my knowledge Werder et al. (2013) examines hard bedded characteristics below glaciers, "subsurface material" needs clarifying. Would an alternative be sediment floored channels or canals.

- Ln 59-71: the modeling work of F. Beaud is likely relevant here, as is the manuscript Hewitt and Creyts (2018) about eskers. Consider adding.

- Ln 87: what does "more dynamic" mean also, I can imagine what "interlobate joints" are, but please clarify.

- Enumerated 1-4 in Intro: I found this useful, and closely linked to Figure 10 in Hovikoski et al. 2023. Would the authors consider applying the cartoon in this manuscript? Additionally, it was difficult for me extract in the enumerated section the model output that would be indicative of this process in murtoo development. Please clarify. Might a table with one column of subglacial hydrology model output help?

- About points 1-2, I am a little curious about the idea that there is sediment deposition at the onset of melt. It seems that the conduit could be small, thus increasing water follow could increase sediment transport capacity, rather than cause sediment deposition. Although available observations of sediment transport are from the terminus, there can often be an increase in sediment transport at this time of the season.

- Ln 120: "higher water velocity" and "development of an englacial pond." To me these processes should not happen at the same place and time. Also, please define "upper-flow-regime."

- Figure 1: Could estimated glacier flow lines be added?

- Ln 195: "modified digital elevation model"…can the section where this is described be referenced?

- Table 1: I am curious if "mean annual velocity" is really an "input" or a model result or output, given the coupling with ISSM.

- Ln 215: "Fixed cross section" or "to the bed at every node." Does this go well here? or is somehow part of experimental design?

- Ln 361: "At node 3,842": maybe make clear that these nodes are representative of their surrounding.

- Figure 3c: should "D" be written as distance? also it seems like this is the end of the melt season of one year. Would it make sense for an "average " to be represented? Also would it make sense to add A-E in the plot in C as to clarify which plots go with which points?

- Ln 505–506: why $10^0$ in one line and 1 in the next?

- Ln 515: "limited cavity expansion" might this be channel floor width?

- Ln 520: "The reason...sediment supply..." My initial reaction upon reading this is that I normally do not consider a distributed drainage system able to transport large amounts of sediment, thus I am unsure about how sediment supply up glacier could impact the results here.

- Ln 524: "More broadly" good pun after speaking of broad channels... "More generally"

- Ln 544-549: I might be missing something. However, melt water input location also seems like a control. For instance, Gagliardini and Werder 2018 may speak to this.

- Ln 558-567: From reading this paragraph, the authors seem to point out the differences between GrIS and the runs here. However, I seem to miss the analysis of the causes of this difference between the two systems.

- Ln 593: "sub-lobes they bound" something funny grammatically, also I am not sure what is meant.

- Ln 606: "1-2 day...walltime?" ]

- Ln 613: $\sim 0.75°C$ how much more water does this result in?

- Ln 627: "macro conditions" what are these precise conditions?

- Figure A2. what is a median discharge? Also, it seems like units are missing.

---

## Referee Comment (RC2)

**Review: Reorganisation of subglacial drainage processes during rapid melting of the Fennoscandian Ice Sheet – Hepburn et al.**
* * *
**General comments – see main review comment box.**

**Problematic text**

The title doesn't seem appropriate: neither "reorganisation" nor "rapid" melting are explicitly examined or discussed.

*Introduction* – there is useful and relevant content here, but its presentation is disjointed and doesn't build towards a specific research question. The murtoo section is important but reads like a fact dump, while the rest of the hydrology overview flags various unknowns without us really knowing which, if any, of these you hope to address. From your "In this paper" statement, I am not sure what you hope to achieve or what you will actually do: "exploring conditions for murtoo formation" and "evaluating models" are rather vague ideas. I suggest: (i) pull out the bulk of the murtoo background and move that to the Study Area section (e.g. "Study area and significance of murtoos for basal hydrology"), leaving just a brief intro and summary of the significance of murtoos for your study in the Introduction; and (ii) revise the remaining sections to build a more coherent narrative that poses a problem and sets out a specific goal of this work.

At the end of the *Study area* section, I am still not clear on what the purpose of murtoos is to this study. Are you testing the conceptual (sedimentology-based) models of how they form? Are you testing numerical model performance, if we accept the sedimentology conceptual model? What is the purpose and scope of this work? Given that scope and research question, what murtoo traits are specifically important to your study, and in what way? Why is e.g. distribution with respect to lineations important, since you don't return to this later in the Discussion? In presenting murtoos as a tool for understanding hydrology, make explicit what information they give and what's important to your study… and come back to this in the Discussion.

*Method section 3.2* – here, as above, I still don't follow what the actual research question or strategy is. What are you looking for when you "compare the GLaDS output to geomorphological evidence"? Are you trying to learn something about the landforms or the modelled hydrology? Why are you choosing a select zone in which to make model-landform comparisons? The Method should outline what you are trying to achieve and why, as well as how.

*Results* - most of the substance (that's developed in the discussion) comes from the baseline model. For greater clarity, consider splitting the baseline experiment results into two subsections, with the split at line 373: 4.1.1 – model behaviour; and 4.1.2 – hydrology in the hypothesised murtoo formation zone.

The sensitivity reporting in 4.2 reports trends, but in neither this section of the results nor in the Discussion is there any *evaluation* of the sensitivity tests. In the absence of any evaluation or any narrative connecting the results to a research question, it is difficult to see what the work contributes. (In this regard, it doesn't help that all the figures are supplementary.) Your Abstract and your Conclusions state that sensitivity testing leads you to a specific parameter space for murtoo formation, but you haven't demonstrated this through any evaluation of sensitivity test outcomes. Which parameter space do you find most plausible, and why? Which parameter space best produces features that fit the geomorphological record (channels/meltwater routes and murtoos), and which best matches the sedimentological interpretation for murtoos – and do these preferred parameter spaces align?

In the _Discussion_, there is some repetition (summary) of results but little "so what" exploration that explicitly follows, while other parts of the Discussion are framed around how well the model performs, rather than what the model finds and what insights that gives us. The discussion of biannual patterns seems like it could be an important finding, but is confusing: one paragraph argues this behaviour is a model artefact, but the next gives a glaciological explanation for it – what's your argument? Some key ideas are lost in heavy text.

The _Conclusions_ focus on what you've done rather than what you've found, and consequently fall flat. Could you revise to e.g. We assume murtoo formation near the headward onset of channelisation, where we find the following conditions: i, ii, iii, iv… Murtoos aren't universally present where those conditions exist, which we interpret in the following way… / which we interpret to mean they also need conditions a, b…

**Line by line technical comments**

17: unless I've missed something in the manuscript, I don't recall "water depths in terrain surrounding murtoos fields" being explored or discussed – what does this refer to?

28: delete one instance of "as"

34: wall melt and channelisation will lower the _water_ pressure, which _raises_ the effective pressure

43-58: the specific topic(s) of this passage and the problem areas or unknowns shift back and forth, making it hard to follow what the limitations are and what "however", "instead", "yet" actually refers to in each instance. E.g. basal hydrology – topography – hydraulic properties – hydraulic connectivity (same meaning?) – back to subglacial hydrology (does "instead" contrast directly with hydraulic properties of sediments, or all knowledge of basal hydrology?) – channelised drainage extent (relation to previous points?) – bed characteristics – "basal parameters". With this rather vague term ending the passage, I'm not sure what the key point was or what the "fundamental challenge" is.

50: "However" should start a new sentence in this construction. (Also line 241, 365)

59: change to glaciofluvial, as elsewhere in the manuscript

60: delete "during periods of rapid ice loss" – this requires a confident and explicit link between a glaciofluvial landform and specific (high) mass loss estimates, which aren't given.

62: define "meltwater routes" – listed here, they sound as if they are different from eskers and tunnel valleys – are these not also meltwater routes?

64-66: this sentence about distributed systems sits out of place with both preceding and subsequent passages dealing with channelised

65: with high _water_ pressures (give low effective pressure)

73: awkward wording, suggest "assumptions about the water pressure, prescribing…"

86: here you define meltwater routes (needed earlier) but this idea was formulated long before Dewald et al. 2022 – use a more appropriate reference (e.g. Utting, Peterson, Lewington, Ahokangas…)

93: delete "closely associated" – it's redundant here

98-99: several ideas need further explanation or definition here: transition to – or from?; what is "semi-distribution drainage"?; why does the spatial proximity of murtoos with eskers "therefore" indicate "repeated and brief pulses of meltwater"?

103 and other instances throughout: reference should be Peterson Becher & Johnson 2021

105-6: here do distal and proximal refer to across a single landform, or the field of murtoos?

106: for consistency with rest of the sentence structure – "proximally is comprised of glaciofluvial deposits with structures such as current ripples….".

116: Stage 2 – what about the diamict that is interbedded?

123: define "upper flow regime"

131: why does laminated mud indicate sudden cessation of murtoo formation? What's the environment for that mud settling from suspension? This sounds unlikely in a subglacial environment, which is presumably where the murtoo is forming. Why should the water in channels or linked cavities be sufficiently still for suspension settling?

132: sudden decay of what?

135: "comprising a main body"

139: "interbedding… is suggested to result from"

141-2: the "small size" of murtoos hasn't yet been presented, and the "onset of channelization" is an interpretation (rather than observation) that murtoos form synchronously with eskers, but upstream. I think this point needs to be rehearsed more fully since it is a fundamental assumption you draw on in interpreting your model output. What supports synchronous formation of murtoos and downstream eskers, rather than murtoo formation at a later stage (i.e. the whole landform assemblage is time-transgressive)?

147: you have a study area, not one specific site that you focus on – revise this header to Study Area (and murtoo significance…)

150: suggest insert "in south-central Finland" after "moraines"

155: "the lateral margins"

158: study area

160: you refer to the other margins (next sentence) as lateral, so for clarity I would here write "bound at its terminal margin" (or distal)

167: typo drumlins

173: "complex landform assemblage" is vague, and what does its complexity have to do with surface melting?

174: revise to "accompanied by calving into the Baltic Sea" – Greenwood et al. 2017 (also 2023, if you find this relevant) find plenty of iceberg scours indicative of an actively calving margin

176: I think it is relevant to note in this section that the ice margin was (shallow) sub-aquatic and not land-terminating.

177-188: this passage is hard to work through – it's very difficult to visualise spatially what's being described. Some information isn't necessary, several sentences give qualifying information before we even learn what's being qualified, and language such as "association between", "borders", variably described sectors/bands/routes is vague. E.g. in the upstream trunk murtoos occur with rm + hummocks in two (?) longitudinal bands, each bounded by a band of lineations. In the northeastern band, murtoos and eskers… (describe arrangement). Downstream, where the FLDIL splays out, murtoo distribution is

fragmented and terrain is more dominated by hummocky moraine. Murtoos are sparse within 40km of the Salpausselkä.

195: somewhere in the methods, note that GLaDS is used with only one-way coupling, i.e. there is no feedback of hydrology on the ice sheet.

203, Table 1 + other initial instances: since the manuscript deals with both an ice sheet and a water sheet, I suggest using the phrase "water sheet" where this is intended and the term first introduced, to avoid confusion

206: "cross-sectional area of which" is a bit confusing – what does "of which" refer back to? Grammatically, it refers to edges, but do you mean channels? Are these the same thing, conceptually? (In which case, how does an edge have a cross-section area?)

210: should be $\rho_i$ not $p_i$ (?) – also in Table 1.

213-14: is this threshold just a matter of classification or does a different equation apply (Eqn 2) above the threshold? Please clarify.

220: I stumbled over "surface elevation" on first reading, since you've just been referring to ice parameters. Suggest "We anticipate that the modern topography (bed elevation) is not representative… and we therefore subtract…"

229: can you finish off this comment on base level change with a suggestion of what effect omitting GIA is anticipated to have?

233: typo except

247: an adaptive

274: delete "around"

282: "nodes were pressured"? Could you add a few words to explain why this is necessary? And what does the velocity of 30 m/yr apply to? Table 1 lists ice velocity as 100-200 m/yr.

291: do you mean the final configuration? "End-member" implies one alternative, at the end of a spectrum of possibilities towards another, opposing, alternative.

298 (and Table 1): what is the basal bump height? If the basal topography is taken from the modified DEM, what is this additional basal elevation variable?

303: note that a shallow water body better replicates the palaeo setting

311: "masked" is ambiguous – did you select these nodes for analysis or exclude them?

322: (overburden$_\%$);

327: does the median include or exclude the 10 years of adjustment time?

327: suggest deleting "pressure expressed as a percentage of overburden" – you've already defined overburden$_\%$

328: should be $Q_c$ for channel discharge

329: since observed murtoos are shown on the plots, I would clarify that the black solid lines are *modelled* channels, not observed/mapped ones.

331-3: the comments about in/efficient drainage here veer towards interpretation, rather than straight reporting of results. It would make it easier to read and digest if you remove these comments and just report overburden$_\%$. I also suggest starting a new paragraph after the overburden sentences – there is a lot of info in this paragraph and splitting it up would make it easier to digest.

333-6: "Towards the ice margin" and "60km from the ice margin" in the same sentence is confusing. I can't follow the description, without looking at the figure. The next two sentences also confused me, since the difference in channels appears to suggest a contrast between summer and winter, but spatially I'm not sure whether you're trying to express a contrast or not. Does the following (paraphrased) work? $q_s$ is orders of magnitude higher in the zone within 60km of the ice margin than further upstream, in both summer and winter. In summer, high $q_s$ is found in patches within the channelised zone. In winter, $q_s$ is much lower, and its peak is shifted to a zone headward of the uppermost channel reach, 40-60km from the margin. (?)

337: as earlier, comments about channels confuse the issue, which is about velocity in sheet water. What does "reflects the concentration of drainage" mean, in this context? It is an interpretation of a mechanism that would account for a particular trait of $V_w$, but would help if it were explicitly presented this way. Focus on presenting the patterns in the main variable, and being clear in expressing spatial relationships – I'm not sure if you are reporting a contrast between summer and winter or not, in either spatial terms or velocity magnitude. I suggest also using the same units/terms as used in the figure, i.e. $15 \times 10^{-4}$ rather than $1.5 \times 10^{-3}$.

341-2: superscript $s^{-1}$. Also lines 348, 352, 369 – check elsewhere.

342: Can you be more specific about the channels that persist over winter – how many? Or what proportion of the summer total?

343/4: suggest move the two sentences about Fig 3A to here, and open with it. By closing the reporting of fig 3 with a "nothing to see" case, all the interest and power of the findings about biannual signals is lost – much better to end with these interesting cases.

351: sharp increase in overburden$_\%$ - not overburden pressure. There are numerous cases in the next paragraph(s) also where "overburden" or "overburden pressure" ought to be "overburden$_\%$", or alternatively "water pressure".

351: "trends towards" – can you say if this is an increase or decrease or values hover around one value?

353: with only a small increase in overburden$_\%$ … and little taken up by sheet flow?

356: this is confusing, suggest "with no overwinter channels evident in the central area while lateral margin channels persist over winter".

357 + throughout: I strongly suggest sticking to expressing pressure either as water pressure or as overburden$_\%$. It is enough with two alternative ways of expressing the same thing. There is no particular need to express the same thing in terms of effective pressure – it's confusing at best and erroneous at worst. Replace here with e.g. "a persistent area of high basal water pressure (overburden$_\%$ approaches and exceeds 100%)…"

360: increase in *water* pressure to 120% of overburden

360: "associated with" is vague, it's not clear what you actually mean in either case you refer to. Suggest delete this phrase here and be more specific later, where necessary. E.g. node 3842, located on a channel onset site. E.g. node 16402, located in between channel onset areas.

371: "remaining at 80% of overburden" – no need for another way of expressing pressure!

373: "To explore behaviours potentially associated with murtoo formation…" ??

375: "…of a mapped meltwater route that also hosts…". Can you also give n=? in each case with the definition of the group (i, ii, iii)? You should also note again here that these meltwater routes are taken from Ahokangas et al., and note how you treat eskers (part of a meltwater route, or separate?).

377: does "throughout the year" mean "mean annual"?

382: is the statistical difference between groups significant for each of the 4 variables, considered separately? Or in combination?

388, 393, 396: could you be more specific where you state a variable is "significantly higher" or lower? In this context, it is easy to misinterpret significantly higher for substantially higher; a difference can be small, but statistically significant. What is meant here? Generally, the blue/orange variables in Fig 4, 5, A32 look pretty similar to each other, so any differences seem like they would be small, but statistically significant – can you express this more accurately and specifically in the written text?

400: what do you mean, "to best apply the model to a palaeoglacial setting"?

404: "water is efficiently evacuated" (or, water pressure is dissipated??)

406 + throughout: as above, express pressure in terms of either water pressure or $overburden_{\%}$

408: "At a minimum sheet conductivity of $10^{-5}$…" – symbol not needed since you wrote out in words

411: $K_C$ should be $k_c$ (small letters)

411 + throughout Section 4.2: Table 1 gives units for $k_s$ as $m^{7/4} kg^{-1/2}$ and for $k_c$ as $m^{3/2} kg^{-1/2}$. These are muddled and inconsistent (7/4 or 3/2) through the Section.

412: "low pressure" – which variable? (Also line 429)

420: "the location of which" – what does "of which" refer back to?

448-455: superscript $yr^{-1}$. Note that elsewhere in the text the unit for years is given as a.

452: what do you mean by "more tightly constrains the observed summer water pressure…" ? I don't follow.

Section 4.2 overall: see comment above (problematic text) – this sections reports the end member behaviour and selected middle option of the range of sensitivity tests, but doesn't anywhere *evaluate* these outputs. How stable/robust are the results presented for the baseline case? What do we learn from these sensitivity tests?

5.1 header: seasonal drainage is implicit in some parts of your discussion in which you compare winter and summer conditions, but seasonality isn't a trait you explicitly dwell on and explore. This doesn't seem a particularly appropriate header (or could be made more so by extracting aspects of seasonality for dedicated discussion).

460: "…demonstrated by Kirkham et al., who evaluated hypotheses…" – in its present formulation, I moved onto the next phrase + reference as a new item, not connected to Kirkham.

472: do papers by Hewitt, Schoof etc not include transitions between channelised + distributed systems (and compare to landform-based predictions of geometry)?

474: does "as" mean because, where, when, … in this context?

475: what does "define a parameter space for basal hydrology models" mean, specifically, in this context?

479: I'm having trouble visualising "vertically arcuate along a comparable path to the surface" – can you rephrase?

483: delete "Instead" – doesn't seem like there's a contrast

485: "…gradients limit the growth of channels…" (?)

486: permit (delete s)

490: "Across the full domain <70km from the margin" sounds like you've got two mutually exclusive areas. I think you perhaps mean within the width of the domain and within 70km of the margin ?

494: suggest deleting "when plotted as a summer average" and change previous line to "summer water pressure"

494: Elsewhere *in the downstream zone*, water pressure … ?

496: insert ii) since you started a list with item i) – and ii) the limited… dropping below 75% of overburden

502 and next two paragraphs: these read more like a summary of the results than a discussion. What's new, what key insights have you gained into murtoo formation and/or murtoo vs channel hydrology? What about all the murtoos upstream of the 40-60km band?

506: if "channel discharge" is only ever close to but not exceeding the threshold for being designated a channel, how does that entity have a "channel discharge" and be measured as $Q_c$? In fact, I should have asked this earlier with the presentation of results and figures e.g. Fig 3, 4, 5 – if a channel isn't a channel until it has a discharge of 1 $m^3$/s, then how can a channel have discharge $Q_c$ below that?

515: what is it that suggests that cavity expansion necessary for murtoos is captured in your model? Expand on "which suggest". I think you also need to expand on the argument that murtoo formation requires the existence of small channels, as a pre-requisite. Is that the case, or have I extracted the wrong idea from what I think you're alluding to? (This is sort of stated in the introduction to murtoo sedimentology, and on the previous page you refer to cavity enlargement, but does a cavity or conduit mean – in model terms – a small channel?)

526: "such landforms relating to meltwater drainage have been mapped…"

531: there is quite a lengthy history of data-model comparison with regards to eskers, and some of that work is likely relevant here (also in the opening of the Discussion, where you also address data-model comparison) e.g. papers by Boulton 2007-2009, by Hewitt 2011, Hewitt & Creyts 2019; work by Flavien Beaud 2016, 2018 among others

535-6: repetitive – suggest "…grid-based models, but the exact location…". "The spacing of channels, however, remains robust…". Delete the early part of the second sentence.

539: does this contradict the earlier result that I flagged in my general comments, that channel discharge is higher outside mapped meltwater routes?

541: presumably the baseline run was selected in order to match the Greenland geometry, though – it's not really a result that this is the case, is it?

542: consistent language with earlier: "In the baseline model run, small channels…"

558: "between our model results and the observations beneath Greenland" ? – "those" implies model results beneath Greenland…

561: "in these systems" – does "these" refer back to the distributed system, channels, or both? Be specific.

571-2: this sentence repeats the opening sentence of the paragraph, but more specifically and concisely – suggest delete the opener or replace it.

572: overburden$_\%$

576: as above, if the "channel" has discharge orders of magnitude below the threshold to be a channel, how can these be considered channels?

583-4, + preceding + following paragraph: you argue biannual behaviour in the model output is a consequence of internal model dynamics (i.e. an artefact) yet then offer an explanation for it in glaciological terms. What is your argument? And what is the relation to a patchy murtoo distribution?

623: study area

631: while you have performed some sensitivity runs, neither your Results nor Discussion section explicitly analyses or evaluates those outputs, so it is false to say that "By sensitivity testing… we demonstrate…". You have selected the baseline run as your favoured answer.

635: that murtoos arise because of "consecutive years of elevated meltwater volumes" has not been clearly demonstrated or argued in the Discussion. Your Discussion has presented biannual behaviour linked to winter persistence of some channels, and implied some related cavity/channel growth behaviour (although see above, 583-604), but your Discussion does not explicitly tie this to murtoo formation. "Elevated meltwater volumes" is also an ambiguous phrase: volume is not considered, as a model variable – do you mean pressure, discharge or velocity?

**Figures & Tables**

*Table 1*
- channel conductivity: the main text refers to experiments with values 0.001, 0.05, 0.5
- ice velocity: given the inference of ice streaming, a tested velocity range of 100-200 m/yr seems rather limited, and low – why?

*Fig 1*
I don't think that slope visualisation of terrain models is the most intuitive way to illustrate and distinguish murtoo morphology, especially for those not accustomed to working with terrain models or unfamiliar with these landform types. I would suggest a conventional hillshade, with illumination best suited to each panel.
Also consider rotating the triangle symbols in the direction of murtoos?
Caption: change to study *area*.
A) "Murtoo fields identified by Ahokangas…".
D) If murtoo fields are adjacent to the esker, then murtoo fields have not undergone an "abrupt downstream transition" to an esker – please revise, the description of a transition is misleading.
E) "within the centre" – delete "at"

*Fig 2*

- give units for $q_s$, $V_w$ and $Q_c$
- In the caption, move the sentence about channels as black lines to after (C), and say that this is the case in panels A-C. (Not just A)
- murtoo triangles aren't visible at this scale (appear as dots), nor is channel scaling for discharge (appear as black lines). I'd remove the murtoo/channel legend, and simply add to the last line of the caption that black dots are murtoo fields, from (which ref?).

*Fig 3*

- label A, B, D, E next to coloured dots on panel C.
- consider flipping the inset graph so that distance = 0 to the right, for consistency with the map plot?
- caption: opening sentence, suggest "at four nodes over model years 15-25 in the baseline model run". And panel C) … "in model year 19 (arbitrarily selected)."
- caption final sentence, should be $q_s$

*Fig 4*

- give units for $q_s$, $V_w$ and $Q_c$
- colours: the red and orange are rather close, and the pink too – could you choose more distinct colours? Also, the legend shows solid pink while the graphs show dashed, and the legend describes as dashed purple. Please revise one or an other to be consistent.
- "Meltwater routes that do not contain murtoo fields (murtoo hosting)"… "routes which do host murtoos (murtoo free)" – both of these phrases in the caption are internally inconsistent. They are also (possibly?) inconsistent with the legend. Please revise.

*Fig 5*

- give units for $q_s$, $V_w$ and $Q_c$
- same errors for murtoo hosting / murtoo free – the caption is internally inconsistent, and please check consistency with legend.
    - I note that I have not checked legends/captions for all appendix figures, but at least A32 has the same error – please revise, and check other figures thoroughly.

*Fig 6*

- caption line 2: delete "against"

---

## Author Comment (AC1)

**Response to reviews: "Reorganisation of subglacial drainage processes during rapid melting of the Fennoscandian Ice Sheet"**

**Reviewer 2**

We thank reviewer 2 for their detailed and helpful comments. Below, we list each comment, our reply and the changes we have made in response. We have made significant changes throughout to our writing and we thank reviewer 2 for their diligence.

**General comments**
* * *
**Comment 2.1:** In principle, this paper puts forward an exciting approach to compare conceptual ideas (ie geologically-driven) of landform (murtoo) formation with model predictions of hydrological settings. I see a lot of potential in such data-model exploration, particularly in relation to murtoos, since they are hypothesised to represent a transition in drainage style. However, I found this manuscript hard work due to "heavy" text, to the extent it is difficult to evaluate the significance of what has been found. The manuscript is suitably constructed in its overall structure, but I struggled with a lack of direction and clarity in the writing. I wasn't sure by the end of the Introduction, or the Methods, what the study was designed to explore, specifically, or what question was being asked; I had to read the Results to try and work out what the authors were actually trying to achieve (beyond a generality). Parts of the Discussion are more a summary of results than an exploration of their significance, and important insights are hidden or implied rather than explicitly stated. The same is true of the Conclusions, which ultimately don't say anything concrete about what has been learned.

**Reply**: We have significantly rewritten the introduction of our paper, including adding specific aims which we hope clarifies what our work was designed to explore, which specifically is a comparison of the specific hydrological conditions proposed for murtoo genesis against model outputs from GlaDS, a model capable of resolving the transition in drainage styles. We have tried to clarify heavy text, particularly in the introduction, placing more focus on why the transition between distributed and channelised drainage is important for glacial hydrology and ice dynamics, and of the potential for palaeo beds to shed light on the transition. We have made major changes to the writing in the discussion. We have also heavily rewritten the conclusions
* * *
**Comment 2.2:** The study area for modelling – and the distribution of murtoos – encompasses the whole Lake District lobe but the entire focus of the results is in the distal few 10s km. I realise the duration of the model effectively only considers one ice time-slice in the development of the whole lobe's landform system, i.e. when the ice margin sits at Salpausselkä II. However, I think this needs to be stated explicitly, and I think the paper needs to discuss what the modelling findings imply for the formation of the other (most of the) murtoos in the domain. Do your results imply (or reject) that the upstream murtoos could form at the same time as those you consider in the near- marginal zone? Do your results imply that they must form time-transgressively (headward) during margin retreat?

**Reply**: We have stated in the discussion that because the area of high $overburden_\%$ is restricted to 40–60 km in most of our model outcomes, murtoos are implied to be time-transgressive, the new lines (586–588) read:

" If we accept the hypothesis that murtoos form where $overburden_\% \approx 100\%$ our modelling supports the idea that the murtoos mapped >70 km from the ice margin postdate 12 cal. ka and that murtoo formation is time-transgressive (Ahokangas et al., 2021)"

We also now note throughout that our model corresponds to the 12 kyr time slice. As this model isn't run as a transient hydrology/ice dynamics coupled configuration we are only able to comment on the conditions at the time slice we are examining and therefore we can't make any further arguments about formation of murtoos from other time periods. Future experiments, involving a more complex model setup could address this interesting question.

**Comment 2.3:** Related, considering the distribution of murtoos over your model domain and the 40-60km band you focus your analysis on, one might wonder why (if) this particular time-slice is well-suited to the investigation – murtoo fields are actually rather few/sparse in this band, compared to elsewhere up the trunk of the Lake District lobe. Could you offer some justification for your approach?

**Reply**: Markers of ice margin position are relatively sparse within the Lake District lobe, so rather than arbitrarily picking a margin position based on murtoo density we chose to bound our model with the clearest ice margin marker at the second Salpausselkä. We have changed the study site section significantly in response to additional comments below, and we do elaborate on the fact that the FLDIL is a particularly well constrained ice lobe, and that there is no ice margin markers beyond Salpausselkä 2.
* * *
**Comment 2.4:** Some recent work is suggesting that the YD may have experienced extreme seasonality, with relatively warm summers but extreme winters (e.g. Schenk et al. 2018 Nat Comms, 2020 QSR; Amon et al. 2022 Clim. Past). You replicate a Younger Dryas climate by lowering present monthly temperatures uniformly by 15°C. I wonder how extreme (or simply, different) YD seasonality would impact your results? How important is this choice of climate forcing for your hydrology conclusions?

**Reply**: We recognise that by simply depressing the climate by 15 degrees we are heavily simplifying the complex climate seasonality during the YD as indicated by these papers. Anecdotally, shorter summers and more extreme winters would likely reduce the length of channels and their discharge, restricting the duration and extent of the murtoo forming zone. It is difficult to say decisively what influence this may have had on our results because neither suggested paper gives a prescriptive climate record on a annual resolution, however, we do acknowledge that the repetitive annual signal here is likely in part responsible for enabling the biannual signal to appear. Formally including this seasonality (in a statistical sense) is potentially a very interesting avenue for future work. However, we note that in fixing our margin to the second Salpausselkä, our domain is representative of the end of the YD during which this seasonality gave way to a markedly warmer climate (e.g., Mangerud et al., 2023).

To acknowledge this, and the important work mentioned above, we further elaborate on our choice of forcing (Lines 310–314), which now reads:

"In simply depressing the climate we are neglecting to include the complex seasonality (short, warm summers with extreme winters) that characterised the Younger Dryas cold reversal in Fennoscandia (Schenk et al., 2018; Amon et al., 2022). However in fixing our domain to the second Salpausselkä our model is representative of the end of the Younger Dryas at which time this extreme seasonality rapidly gave way to a markedly warmer climate with similar seasonality to the present day (Mangerud et al., 2023)"

**Reference:** Jan Mangerud, Anna L.C. Hughes, Mark D. Johnson, Juha Pekka Lunkka, Chapter 46 - The Fennoscandian Ice Sheet during the Younger Dryas Stadial, Editor(s): David Palacios, Philip D. Hughes, José M. García-Ruiz, Nuria Andrés, European Glacial Landscapes, Elsevier, 2023, Pages 437-452, ISBN 9780323918992, https://doi.org/10.1016/B978-0-323-91899-2.00060-7.
* * *
**Comment 2.5:** Section 3.2 implies that only the zone 40-60km upstream of the ice margin is analysed, on the basis that this zone is favourable to murtoo formation. This seems a little circular to me. By focusing only here, do you not exclude the possibility of identifying conditions that would suit murtoo formation elsewhere? And exclude possible comparable hydrological/glaciological conditions elsewhere that may or may not support murtoos?

**Reply**: We have modified Section 3.2 to make clear that we did indeed analyse the full domain, but that given the time-integrated nature of the landform record, we did not expect to (for example) be able to compare modelled channels within our domain (representative of ~12 cal. ka) against eskers formed across the domain and likely formed long after 12 cal. ka. Regarding the murtoos specifically, we isolated the zone

40–60 km from the ice margin because that is the hypothesised area of murtoo formation, and it provides a readily testable set of conditions with which we can begin to explore our model results.

It is true that in doing so, we ignore the potential for murtoo-suitable conditions elsewhere. However, across the suite of our model sensitivity tests we note that the area of $overburden_\%  \approx 100\%$, a key characteristic of the hypothesised murtoo formation environment, is largely confined to this 40–60 km area $\pm 10$ km. In tests where this is not true, for example when sheet or channel conductivity are at the tested limits, modelled channels are extremely short and densely spaced, or not present at all, which provides an independent query of those specific parameters. The modified text to this effect can be found in Section 3.2.
* * *
**Comment 2.6:** One of your results is rather surprising, which calls into question the appropriateness of comparing model output with specific mapped geomorphology: if channel discharge in late summer is significantly higher outside mapped meltwater routes (line 395) than where meltwater routes have been recorded, then this suggests the spatial distribution of channels predicted by the model is offset from where channels are known to have existed. This mismatch therefore also questions the validity of comparing hydrological parameters with where murtoos have been mapped or not mapped.

**Reply**: Thank for noting this, which is a major error on our part. Figures 4 & 5 show that channel discharge is not higher outside of mapped meltwater routes than within mapped meltwater routes, and in fact the opposite is true. We have fixed this text which now reads:

"There is no significant difference between any group in terms of $Q_c$ with the exception of between June–October (Table A3), during which $Q_c$ is significantly higher within murtoo and meltwater routes than beyond"
* * *
**Comment 2.7:** A further uncertainty in this regard concerns what is meant by a meltwater route? Section 3.2 suggests these are based on Ahokangas et al. As far as I understand, those authors keep eskers separate from their meltwater route classes – how did you treat eskers in your classification of "meltwater routes"?

At the very least, I think the presentation of results relating to geomorphological classifications needs to be preceded by a justification that the model performance is adequate in terms of what we know of the geomorphology: does the model do a good job of replicating channels? If it does, then it's a valid tool to explore other landforms, and if not, then it's not. This partly is presented in the Discussion (524- 540) (though the earlier result I've queried here is not addressed) and I think would better serve the Results section if it were brought earlier. I also wonder, if the exact location of channels is sensitive to mesh geometry (536) then does this also not suggest that specific site-to-site data-model comparisons may not be appropriate?

**Reply**: In response to the first point about meltwater routes, Ahokangas et al., do treat meltwater routes separately from esker deposits (including them as "channelised routes"), however in many places they are coincident and cross or follow other meltwater routes, suggesting they are later features. Without an age-control on each of these, in this classification, we do not make a distinction between eskers and meltwater routes, reasoning that all are geomorphological indicators of meltwater flow. We have added two sentences to the end of this section (3.2) which now read:

"Ahokangas et al., (2021) mapped eskers separately to meltwater routes, including these as "channelised routes" in their datase, however, they note that many of these routes fall within meltwater routes and likely correspond to a later time of formation. Accordingly, without age-control, we do not make a distinction between meltwater routes and channelised routes here. "

To the second point, we have added a sentence to the results saying that the spacing and appearance of model channels compares well to mapped esker deposits:

"...with channels arranged perpendicular to, and extending up to $\sim 50$ km inland of, the ice margin and comparable in structure and spacing to the location of esker deposits in the FLDIL..."

We have also moved what was figure 6 (which shows the comparison of eskers to modelled channels) into the results. Regarding their specific location, and what it means for the model to be "doing a good job", though

the exact location of channels varies according to mesh geometry, within reasonable bounds (i.e., not when edge lengths exceed tens of km) the spacing and length of channels remains robust, and channels largely fall in similar locations. Any modelling work has to make a decision on what level of fidelity is an reasonable approximation of reality, balancing this against the greater computational cost of highly refined meshes. As such, where the mesh is fixed in space and time, the exact location of channels will always depend on mesh geometry to some extent. Note recent work by Felden et al., 2023 which uses an adaptive mesh in order to attempt to overcome this, however, they do so using a grid structure which imposes directional biases.

We have added a sentence in the discussion to explain why mesh dependency arises is the case and we also do note that the spacing and alternating pattern of larger and smaller eskers is similarly reproduced within our model. The text (Lines 626–629) in question now reads:

"Modelled channels in our baseline model (Figure 2) and many of the sensitivity tests have similar locations as eskers mapped by Palmu et al (2021), particularly in terms of their lateral spacing, length, and the observation that smaller esker deposits are alternately found between large features (Figure 2)"

and on Line 632–637:

"In the baseline model specifically, at several locations, modelled channel outputs closely track the location of several particularly large esker deposits (Figure 2B–C). We caveat this by noting that because our model operates on a mesh, the resolution of which is a balance of suitable fidelity against the increased computational cost of resolving finer details, the exact location of these modelled channels is sensitive to mesh geometry. Channels cannot form where no element edge exists. Differences in the exact channel location also arise because of moulin density and location, bed topography, velocity, and basal bump height. Nonetheless, the spacing and length of channels remains robust against the parameters tested here"

As elaborated upon on in our limitations section (Section 5.4), many of our decisions and restrictions mean our representation of the FLDIL is idealised, as to some extent all modelling is. Accordingly, we do not argue in our manuscript that our modelling is capable of resolving specific murtoo fields or esker deposits, rather in our idealised representation of the FLDIL we hope to be able to represent the broad patterns of drainage.

**Reference:** Felden, A. M., Martin, D. F., and Ng, E. G.: SUHMO: an adaptive mesh refinement SUbglacial Hydrology MOdel v1.0, Geosci. Model Dev., 16, 407–425, https://doi.org/10.5194/gmd-16-407-2023, 2023.
* * *
**Comment 2.8:** Overall, the manuscript would have benefitted from a thorough proof-read – there are numerous typos, left-over words from earlier constructions, unit errors, muddled variables.

I note these in a rather lengthy list of line-by-line technical comments in the attached pdf, where I also identify issues with writing clarity, section by section, and suggest how the direction and framing of the work could be improved.

**Reply**: Below, we address each of these technical comments.

**Specific comments**

**Comment 2.9:** The title doesn't seem appropriate: neither "reorganisation" nor "rapid" melting are explicitly examined or discussed.

**Reply**: Please see our response to Comment 1.1:, the new title is: "The organisation of subglacial drainage during the demise of the Finnish Lake District Ice-Lobe" .

**Comment 2.10:** Introduction – there is useful and relevant content here, but its presentation is disjointed and doesn't build towards a specific research question. The murtoo section is important but reads like a fact dump, while the rest of the hydrology overview flags various unknowns without us really knowing which, if any, of these you hope to address. From your "In this paper" statement, I am not sure what you hope to achieve or what you will actually do: "exploring conditions for murtoo formation" and "evaluating models" are rather vague ideas. I suggest: (i) pull out the bulk of the murtoo background and move that to the Study Area section (e.g. "Study area and significance of murtoos for basal hydrology"), leaving just a brief intro and summary of the significance of murtoos for your study in the Introduction; and (ii) revise the remaining sections to build a more coherent narrative that poses a problem and sets out a specific goal of this work.

**Reply**: As suggested, we have removed the bulk of the murtoo detail and folded it into the study area section (Section 2). We have also heavily modified the introduction, which we hope now builds towards our specific research aims, listed at the end of the shortened introduction and which now read:
"

- Compare the subglacial hydrological conditions proposed for murtoo genesis and their associated landforms against model outputs from GlaDS.

- Sensitivity test GlaDS across a range of possible parameter values to explore the influence of these parameters on our outcomes in order to evaluate the potential of such models to be used to interrogate palaeo-hydrological systems more broadly and in turn motivate future work in this area.

"

**Comment 2.11:** At the end of the Study area section, I am still not clear on what the purpose of murtoos is to this study. Are you testing the conceptual (sedimentology-based) models of how they form? Are you testing numerical model performance, if we accept the sedimentology conceptual model? What is the purpose and scope of this work? Given that scope and research question, what murtoo traits are specifically important to your study, and in what way? Why is e.g. distribution with respect to lineations important, since you don't return to this later in the Discussion? In presenting murtoos as a tool for understanding hydrology, make explicit what information they give and what's important to your study... and come back to this in the Discussion.

**Reply**: Following the previous comment, we reworked the introduction and study area section as suggested. Our aims to compare predictions of murtoo genesis to modelling, and sensitivity test the model, are laid out in the introduction and we have moved the murtoos to a separate section (Section 2). In Section 2, we detail the key developmental characteristics other murtoo studies have suggested are associated with murtoo formation. Following a suggestion from Reviewer 1, we have added a table (Table 1) which lists the murtoo developmental stages, the sedimentological evidence, and the expected model outcomes. We refer to this throughout the discussion, using the table as a signpost to do so.

**Comment 2.12:** Method section 3.2 – here, as above, I still don't follow what the actual research question or strategy is. What are you looking for when you "compare the GLaDS output to geomorphological evidence"? Are you trying to learn something about the landforms or the modelled hydrology? Why are you choosing a

select zone in which to make model-landform comparisons? The Method should outline what you are trying to achieve and why, as well as how.

**Reply**: We have rewritten this section, and changed the opening sentence to echo the language of our specific aims given in the introduction so that Line 380 now reads:

"Finally, we compared the GlaDS output to the subglacial hydrological conditions proposed for murtoo genesis".

Regarding the specific zone of 40–60 km we have added text (Lines 387–390) explaining why we did so, which now read:

"Similarly, and assuming that the mapped murtoo distribution is also representative of a time-transgressive mode of origin, we examine the performance of our model within the hypothesised zone of murtoo formation (e.g., Ojala et al., 2019) by specifically isolating model nodes falling within 40–60 km of our ice margin representative of the FLDIL extent ∼12 cal. ka"
* * *
**Comment 2.13:** Results - most of the substance (that's developed in the discussion) comes from the baseline model. For greater clarity, consider splitting the baseline experiment results into two subsections, with the split at line 373: 4.1.1 – model behaviour; and 4.1.2 – hydrology in the hypothesised murtoo formation zone.

**Reply**: As suggested, we have added two subsections to our description of the baseline model, one on model behaviour and the other on hydrology in the hypothesised zone of murtoo formation.
* * *
**Comment 2.14:** The sensitivity reporting in 4.2 reports trends, but in neither this section of the results nor in the Discussion is there any evaluation of the sensitivity tests. In the absence of any evaluation or any narrative connecting the results to a research question, it is difficult to see what the work contributes. (In this regard, it doesn't help that all the figures are supplementary.) Your Abstract and your Conclusions state that sensitivity testing leads you to a specific parameter space for murtoo formation, but you haven't demonstrated this through any evaluation of sensitivity test outcomes. Which parameter space do you find most plausible, and why? Which parameter space best produces features that fit the geomorphological record (channels/meltwater routes and murtoos), and which best matches the sedimentological interpretation for murtoos – and do these preferred parameter spaces align?

**Reply**: The purpose of sensitivity testing in GlaDS is to determine a reasonable range of outputs and we have changed our abstract and conclusions. We have added discussion to this effect in our methods, results, and we also further discuss this in the discussion. GlaDS has already been extensively sensitivity tested by a number of previous authors, so we did not repeat this effort by carrying out and reporting extensive fresh sensitivity tests here. Instead, we used knowledge from contemporary ice sheets and hydrology model outputs to determine what the 'most likely', or plausible, model outcome is. For example, maximum channel lengths of less than 1 km long are not realistic, and water pressures consistently below 75% of overburden 60 km from the ice margin are not realistic. Our baseline model (particularly in terms of channel and sheet conductivity terms, which are known to be the most important in GlaDS) therefore presents the most likely output given this range of sensitivity tests that we have conducted. The most plausible parameter space is therefore the baseline run. We intended to report the dependence of our findings on these parameters in Section 4.2. Our finding that without reasonable ranges, i.e., both those found by other modelling studies and runs which remain numerically stable, our conclusions (about channel spacing, length, and the pattern of overburden) are largely insensitive to specific modelling choices, accordingly we did not spend a great deal of text explaining this. However, we acknowledge that we clearly dedicated too little text to this, and have added throughout, in the methods (Lines 340–345) we explicitly state how we arrived at our baseline model parameters:

" We set the parameters in our baseline model (default values listed in Table 2) following the default values in these studies which provide a reasonable approximation of contemporary ice sheet subglacial conditions. We

then explored the sensitivity of our specific model outcomes to the available parameters (e.g., conductivity terms)"
* * *
**Comment 2.15:** In the Discussion, there is some repetition (summary) of results but little "so what" exploration that explicitly follows, while other parts of the Discussion are framed around how well the model performs, rather than what the model finds and what insights that gives us. The discussion of biannual patterns seems like it could be an important finding, but is confusing: one paragraph argues this behaviour is a model artefact, but the next gives a glaciological explanation for it – what's your argument? Some key ideas are lost in heavy text.

**Reply**: We have rewritten the discussion extensively to more clearly discuss our findings, which as per our aims is primarily focused on seeing whether GlaDS produces the hypothesised conditions of murtoo formation. We also have reworked our discussion about the biannual signal to argue that the biannual pattern arises from spatial variability in our forcing combined with flow direction divergence within the ice lobe.
* * *
**Comment 2.16:** The Conclusions focus on what you've done rather than what you've found, and consequently fall flat. Could you revise to e.g. We assume murtoo formation near the headward onset of channelisation, where we find the following conditions: i, ii, iii, iv... Murtoos aren't universally present where those conditions exist, which we interpret in the following way... / which we interpret to mean they also need conditions a, b...

**Reply**: We have rewritten the conclusions, which we do agree were not sufficiently clear before. We hope we have addressed this concern by adopting the suggested structure. Although we note that we do not assume murtoo formation near the headward onset of channelisation, within a range of our sensitivity tests our modelling results *support* this hypothesis.
* * *
**Comment 2.17:** 17: unless I've missed something in the manuscript, I don't recall "water depths in terrain surrounding murtoos fields" being explored or discussed – what does this refer to?

**Reply**: We have deleted this text in our modifications of the abstract
* * *
**Comment 2.18:** 28: delete one instance of "as"

**Reply**: We have deleted the first as.
* * *
**Comment 2.19:** 34: wall melt and channelisation will lower the water pressure, which raises the effective pressure

**Reply**: We have corrected this as suggested.
* * *
**Comment 2.20:** 43-58: the specific topic(s) of this passage and the problem areas or unknowns shift back and forth, making it hard to follow what the limitations are and what "however", "instead", "yet" actually refers to in each instance. E.g. basal hydrology – topography – hydraulic properties – hydraulic connectivity (same meaning?) – back to subglacial hydrology (does "instead" contrast directly with hydraulic properties of sediments, or all knowledge of basal hydrology?) – channelised drainage extent (relation to previous points?) – bed characteristics – "basal parameters". With this rather vague term ending the passage, I'm not sure what the key point was or what the "fundamental challenge" is.

**Reply**: As part of the rewrite we have heavily reworked this, removing the troublesome last sentence entirely. We have rewritten the introduction with a view to 1) emphasising the importance of basal hydrology in

modelling ice sheet mass loss, 2) stating such models are rarely included in full ice sheet models because of significant uncertainties as a result of overlying ice cover, 3) putting forward the idea that in order to correctly model basal hydrology, we must use all available sources of data to understand the parameter space of these models.
* * *
**Comment 2.21:** 50: "However" should start a new sentence in this construction. (Also line 241, 365)

**Reply**: The 'however' in this construction is a parenthetical aside providing additional information not essential to the main clause. It therefore does not need to be a new sentence. Nonetheless, as part of the rewrite line 50 was removed, but we have left the however on line 365 (now 451) as is. We have changed added a new sentence before the 'however' on what was line 241 (now 295) to break up the otherwise long sentence.
* * *
**Comment 2.22:** 59: change to glaciofluvial, as elsewhere in the manuscript

**Reply**: This is the only instance of 'fluvioglacial' in the manuscript we could find. We have changed it to glaciofluvial.
* * *
**Comment 2.23:** 60: delete "during periods of rapid ice loss" – this requires a confident and explicit link between a glaciofluvial landform and specific (high) mass loss estimates, which aren't given.

**Reply**: We have deleted this as suggested.
* * *
**Comment 2.24:** 62: define "meltwater routes" – listed here, they sound as if they are different from eskers and tunnel valleys – are these not also meltwater routes?

**Reply**: There is an unfortunately hazy terminology in this respect between different studies, and to avoid this we delete the reference here to meltwater routes. We introduce the concept later on and define it there (see the comment below).
* * *
**Comment 2.25:** 64-66: this sentence about distributed systems sits out of place with both preceding and subsequent passages dealing with channelised

**Reply**: In changing the intro, we have positioned this sentence about the distributed system more as a direct comparison to the abundance of channelised forms.
* * *
**Comment 2.26:** 65: with high water pressures (give low effective pressure)

**Reply**: We have made the suggested change
* * *
**Comment 2.27:** 73: awkward wording, suggest "assumptions about the water pressure, prescribing..."

**Reply**: As suggested, we have simplified this wording
* * *
**Comment 2.28:** 86: here you define meltwater routes (needed earlier) but this idea was formulated long before Dewald et al. 2022 – use a more appropriate reference (e.g. Utting, Peterson, Lewington, Ahokangas...)

**Reply**: For the specific terminology meltwater route, we have added references to Lewington et al., 2020, and Ahokangas et al., 2021 as suggested. This is now the first reference to meltwater routes, as so we leave the definition here.
* * *
**Comment 2.29:** 93: delete "closely associated" – it's redundant here

**Reply**: We have deleted this as suggested
* * *
**Comment 2.30:** 98-99: several ideas need further explanation or definition here: transition to – or from?; what is "semi- distribution drainage"?; why does the spatial proximity of murtoos with eskers "therefore" indicate "repeated and brief pulses of meltwater"?

**Reply**: We have clarified that semi-distributed represents a transition between distributed and channelised. We have also removed the reference to repeated and brief pulses of meltwater, which does have no relevance in this particular sentence.
* * *
**Comment 2.31:** 103 and other instances throughout: reference should be Peterson Becher & Johnson 2021

**Reply**: We have corrected this throughout.
* * *
**Comment 2.32:** 105-6: here do distal and proximal refer to across a single landform, or the field of murtoos?

**Reply**: We have added the word individual, although we did use murtoo (singular) throughout.
* * *
**Comment 2.33:** 106: for consistency with rest of the sentence structure – "proximally is comprised of glaciofluvial deposits with structures such as current ripples....".

**Reply**: We have reworked this sentence as suggested.
* * *
**Comment 2.34:** 116: Stage 2 – what about the diamict that is interbedded?

**Reply**: Interbedded diamicton (often trough-shaped structures in Unit 2b by Mäkinen et al 2023) primarily corresponds to late summer melt and so we have added this to Stage 4.
* * *
**Comment 2.35:** 123: define "upper flow regime"

**Reply**: Please see our changes made in response to reviewer 1's similar comment (1.12)
* * *
**Comment 2.36:** 131: why does laminated mud indicate sudden cessation of murtoo formation? What's the environment for that mud settling from suspension? This sounds unlikely in a subglacial environment, which is presumably where the murtoo is forming. Why should the water in channels or linked cavities be sufficiently still for suspension settling?

**Reply**: Laminated muds are widely documented in the marginal channels of murtoos (see Ojala et al., 2022) and are interpreted as a change in the local hydraulic connectivity of that channel. We know beneath contemporary ice that adjacent area of the bed can be hydraulically isolated and respond differently (or not at all) to localised meltwater inputs (e.g., Rada and Schoof 2018). Whether it by localised sediment deposition associated with the murtoo shape or otherwise (e.g., at the end of the melt season) the appearance of laminated muds suggests that what was a connected channel becomes isolated and the remaining water

sufficiently still for suspension settling and any additional water is rerouted elsewhere. We have modified the text to suggest this which now reads:

"Finally, murtoo deposition is abruptly terminated and marginal channels are abandoned. The final sedimentation within these marginal channels is characterised by suspension settling and laminated muds, indicating that the depositional space (0.6–0.8 m) remained open and water filled but no longer hydraulically connected to the wider meltwater system (Ojala et al., 2022, Hovioski et al., 2023)."

We also note in the discussion that GlaDS is unable to represent changes in hydraulic connectivity, and that we do not observe such changes in our model outputs.

**References:** Rada, C. and Schoof, C.: Channelized, distributed, and disconnected: subglacial drainage under a valley glacier in the Yukon, *The Cryosphere*, 12, 2609–2636.

Ojala, A.E.K., Mäkinen, J., Kajuutti, K., Ahokangas, E., Palmu, J.-P. (2022) Subglacial evolution from distributed to channelized drainage: Evidence from the Lake Murtoo area in SW Finland. *Earth Surface Processes and Landforms*, 47(12), 2877–2896.
* * *
**Comment 2.37:** 132: sudden decay of what?

**Reply**: We have modified this sentence to now read:

"The sedimentological architecture within murtoos suggest an overall increase and abrupt reduction in meltwater discharge which indicates that murtoo development occurs within a single melt season"
* * *
**Comment 2.38:** 135: "comprising a main body"

**Reply**: We have adopted this suggestion
* * *
**Comment 2.39:** 139: "interbedding... is suggested to result from"

**Reply**: We have adopted this suggestion
* * *
**Comment 2.40:** 141-2: the "small size" of murtoos hasn't yet been presented...

**Reply**: We have added "small" to the introduction where we had already described murtoos as low-relief.
* * *
**Comment 2.41:** Cont:... and the "onset of channelization" is an interpretation (rather than observation) that murtoos form synchronously with eskers, but upstream. I think this point needs to be rehearsed more fully since it is a fundamental assumption you draw on in interpreting your model output. What supports synchronous formation of murtoos and downstream eskers, rather than murtoo formation at a later stage (i.e. the whole landform assemblage is time- transgressive)?

**Reply**: Considered in isolation, the link to eskers does not solely imply murtoo/esker formation was synchronous, however the sedimentological evidence, murtoo distance from the Salpausselkä ice marginal features, and their geomorphology all support the idea that they do not post-date eskers (e.g., formed at or nearer to the ice margin than the limit of channelisation). We have modified this section (Lines 119–126) to clarify this, which now reads:

"Murtoo morphometry (Mäkinen et al., 2017; Ojala et al., 2021), their sedimentological architecture (Peterson Becher and Johnson, 2021; Hovikoski et al., 2023; Mäkinen et al., 2023), and close spatial association with eskers, ribbed tracts, and putative subglacial lakes (Ojala et al., 2021; Ahokangas et al., 2021; Vérité et al., 2022; Mäkinen et al., 2023) is suggestive of rapid murtoo formation within broad and low conduits, at

effective pressures close to zero, characterised by short sediment transport distances, and subject to repeated short pulses of meltwater, such as might be found at the spatial onset of channelisation in a 'semi-distributed' transitional drainage system (Hovikoski et al., 2023).

Murtoos are therefore unique glaciofluvial landforms, and their short formation time, small size, and apparent location at the spatial onset of channelisation make murtoos potentially important components of the subglacial system."

We go on to more fully establish these ideas in the next section.
* * *
**Comment 2.42:** 147: you have a study area, not one specific site that you focus on – revise this header to Study Area (and murtoo significance...)

**Reply**: We have revised as suggested.
* * *
**Comment 2.43:** 150: suggest insert "in south-central Finland" after "moraines"

**Reply**: We have revised this as suggested.
* * *
**Comment 2.44:** 155: "the lateral margins"

**Reply**: We have revised this as suggested.
* * *
**Comment 2.45:** 158: study area

**Reply**: In response to the previous comment (Comment 2.42:) we have revised this throughout.
* * *
**Comment 2.46:** 160: you refer to the other margins (next sentence) as lateral, so for clarity I would here write "bound at its terminal margin" (or distal)

**Reply**: We have revised this as suggested.
* * *
**Comment 2.47:** 167: typo drumlins

**Reply**: We have revised this.
* * *
**Comment 2.48:** 173: "complex landform assemblage" is vague, and what does its complexity have to do with surface melting?

**Reply**: We have clarified that the landform assemblage are glaciofluvial which now reads:

"...together with the complex assemblage of glaciofluvial landforms..."
* * *
**Comment 2.49:** 174: revise to "accompanied by calving into the Baltic Sea" – Greenwood et al. 2017 (also 2023, if you find this relevant) find plenty of iceberg scours indicative of an actively calving margin

**Reply**: We have revised this as suggested.
* * *
**Comment 2.50:** 176: I think it is relevant to note in this section that the ice margin was (shallow) sub-aquatic and not land-terminating.

**Reply**: We have added a line to this effect in our study area section (Section 2, Line 149–150) which reads:

"Shoreline data indicates that the second Salpausselkä terminated in a shallow water body ranging in depth from <5 m to ∼50 m (Lunkka and Erikkilä, 2012)"
* * *
**Comment 2.51:** 177-188: this passage is hard to work through – it's very difficult to visualise spatially what's being described. Some information isn't necessary, several sentences give qualifying information before we even learn what's being qualified, and language such as "association between", "borders", variably described sectors/bands/routes is vague. E.g. in the upstream trunk murtoos occur with rm + hummocks in two (?) longitudinal bands, each bounded by a band of lineations. In the northeastern band, murtoos and eskers... (describe arrangement). Downstream, where the FLDIL splays out, murtoo distribution is fragmented and terrain is more dominated by hummocky moraine. Murtoos are sparse within 40km of the Salpausselkä.

**Reply**: We have reworked much of this text in our attempt to trim down the 'fact-dump' in response to a previous comment. The edits, following the suggested structure, can be found in Section 2.
* * *
**Comment 2.52:** 195: somewhere in the methods, note that GLaDS is used with only one-way coupling, i.e. there is no feedback of hydrology on the ice sheet.

**Reply**: We have added a line to this effect at the end of Section 3.1 which reads:

"Finally, in the iteration used here, GlaDS is not coupled two-ways to a model of ice dynamics, and instead we prescribe an ice velocity and geometry that is not variable in response to hydrological forcing."
* * *
**Comment 2.53:** 203, Table 1 + other initial instances: since the manuscript deals with both an ice sheet and a water sheet, I suggest using the phrase "water sheet" where this is intended and the term first introduced, to avoid confusion

**Reply**: We have not adopted this suggested wording. Sheet is an accepted term within the hydrology modelling community and beyond, and we are dealing with a fixed 'ice sheet' geometry here and do not discuss the ice sheet itself in any substantial detai. Where this [ice] sheet is discussed it is always prefixed by 'ice'. Where the '[water] sheet' is discussed, it is always in reference to the distributed system However, we have added a note to the table caption (now Table 2) and to the first use of the term in the methods in order to address the potential for any confusion.
* * *
**Comment 2.54:** 206: "cross-sectional area of which" is a bit confusing – what does "of which" refer back to? Grammatically, it refers to edges, but do you mean channels? Are these the same thing, conceptually? (In which case, how does an edge have a cross-section area?)

**Reply**: In GlaDS, every edge is effectively a channel and always has a non-zero water discharge (which is why it is necessary to set a minimum threshold for classification as a 'meaningful' channel). Though each edge is fixed spatially, they do have a parameter which describes their effective cross-sectional area in the same way that a 2D ice sheet model can have an ice thickness value despite being dimensionless in the $z$-direction. We have modified the text to address this confusion, so that it now reads:

"Sheet elements exchange water with channels and the cross sectional area of these channels $S$, evolves through time due to the dissipation of potential energy" to fix the ambiguous "which"

Above, this we have also (in response to a separate comment from reviewer 1) modified the description of the mesh to clarify that the sheet is represented by elements, and channels are represented by edges. This now reads:

"The GlaDS model operates on an unstructured mesh and includes a model of distributed flow through linked cavities represented by a continuous sheet of variable thickness at mesh elements, and channelised flow—describing uniform, semi-circular Röthlisberger channels (R-channels) that are allowed to change diameter—along element edges"
* * *
**Comment 2.55:** 210: should be $\rho_i$i not p$_i$ (?) – also in Table 1.

**Reply**: We have corrected this to $\rho$ throughout
* * *
**Comment 2.56:** 213-14: is this threshold just a matter of classification or does a different equation apply (Eqn 2) above the threshold? Please clarify.

**Reply**: We have clarified this line to now read:

"In GlaDS, water discharge is non-zero along all edges and so following Werder er al (2013), we set a threshold discharge of $Q_c = 1\,\mathrm{m^3\,s^{-1}}$ above which an element edge is classified as a channel for our subsequent analysis"
* * *
**Comment 2.57:** 220: I stumbled over "surface elevation" on first reading, since you've just been referring to ice parameters. Suggest "We anticipate that the modern topography (bed elevation) is not representative... and we therefore subtract..."

**Reply**: We have modified this line to now read:

"We anticipate that the modern topography is not representative of bed elevation $\sim$12 cal. ka. Therefore, as the baseline boundary condition, $z_b$, we account for changes, particularly..."
* * *
**Comment 2.58:** 229: can you finish off this comment on base level change with a suggestion of what effect omitting GIA is anticipated to have?

**Reply**: We have added that this would increase the mean annual air temperature by $\sim$0.75 C in Section 3.1.1. We go on to imagine what effect this may have on our model in the limitations and future work section.
* * *
**Comment 2.59:** 233: typo except

**Reply**: We have rectified this.
* * *
**Comment 2.60:** 247: an adaptive

**Reply**: We have rectified this.
* * *
**Comment 2.61:** 274: delete "around"

**Reply**: We have rectified this.
* * *
**Comment 2.62:** 282: "nodes were pressured"? Could you add a few words to explain why this is necessary? And what does the velocity of 30 m/yr apply to? Table 1 lists ice velocity as 100-200 m/yr.

**Reply**: We have added three more equations and a few lines to our description of how the sheet thickness evolves through time in the method. This has allowed us to more clearly state that the basal sliding velocity controls cavity opening rates in the distributed system, and a lower velocity, $U_b$ forces cavity opening rates to remain low, raising the pressure, and best approximating the winter state of the system (a highly pressurised distributed system). The preceding two sentences explain more clearly why we did this (Lines 347–350). They now read:

"For all model runs, to avoid overwhelming an unpressurised initial system with sudden meltwater inputs and to approximate a wintertime hydrology configuration characterised by a high pressure distributed system, we first ran GlaDS to steady state with no surface melt and fixed basal meltwater input. To guarantee the majority of elements were pressurised at the end of our steady state run, we used a low, fixed velocity of $30 \, \mathrm{m \, yr^{-1}}$ which limited cavity expansion (see Equation 3)"
* * *
**Comment 2.63:** 291: do you mean the final configuration? "End-member" implies one alternative, at the end of a spectrum of possibilities towards another, opposing, alternative.

**Reply**: We have adopted the suggested wording.
* * *
**Comment 2.64:** 298 (and Table 1): what is the basal bump height? If the basal topography is taken from the modified DEM, what is this additional basal elevation variable?

**Reply**: Basal bump height is a term to account for sub-grid variability in the local elevation of the surface, beyond the resolution of the DEM. It is a term important in allowing small cavities to nucleate and allow the distributed sheet thickness to evolve through time (See the new equation 3).
* * *
**Comment 2.65:** 303: note that a shallow water body better replicates the palaeo setting

**Reply**: We have modified the text to include this, but we also note that our sensitivity testing indicates accounting for the shallow water body has limited influence on the model outcomes.
* * *
**Comment 2.66:** 311: "masked" is ambiguous – did you select these nodes for analysis or exclude them?

**Reply**: Masked has been replaced with "isolated"
* * *
**Comment 2.67:** 322: (overburden$_\%$

**Reply**: we have updated this to *overburden*$_\%$ throughout the manuscript.
* * *
**Comment 2.68:** 327: does the median include or exclude the 10 years of adjustment time?

**Reply**: It does not include the adjustment time, but the adjustment time should be 5 years (as stated in the caption for Figure 2) we have corrected this.
* * *
**Comment 2.69:** 327: suggest deleting "pressure expressed as a percentage of overburden" – you've already defined overburden%

**Reply**: We have removed the suggested text.
* * *
**Comment 2.70:** 328: should be $Q_c$ for channel discharge

**Reply**: We have modified this.
* * *
**Comment 2.71:** 329: since observed murtoos are shown on the plots, I would clarify that the black solid lines are modelled channels, not observed/mapped ones.

**Reply**: We have made the suggested change.
* * *
**Comment 2.72:** 331-3: the comments about in/efficient drainage here veer towards interpretation, rather than straight reporting of results. It would make it easier to read and digest if you remove these comments and just report overburden%. I also suggest starting a new paragraph after the overburden sentences – there is a lot of info in this paragraph and splitting it up would make it easier to digest.

**Reply**: We have removed the references to the efficiency of the drainage system and split the paragraph after our reporting of $overburden_\%$.
* * *
**Comment 2.73:** 333-6: "Towards the ice margin" and "60km from the ice margin" in the same sentence is confusing. I can't follow the description, without looking at the figure. The next two sentences also confused me, since the difference in channels appears to suggest a contrast between summer and winter, but spatially I'm not sure whether you're trying to express a contrast or not. Does the following (paraphrased) work? qs is orders of magnitude higher in the zone within 60km of the ice margin than further upstream, in both summer and winter. In summer, high qs is found in patches within the channelised zone. In winter, qs is much lower, and its peak is shifted to a zone headward of the uppermost channel reach, 40-60km from the margin. (?)

**Reply**: Addressing the first point of this, we have changed line 415 to now read:

"Throughout the year, $q_s$ sharply decreases $60\,\mathrm{km}$ from the margin"

To the second point, the following two sentences now read:

"In summer, areas of high $q_s$ (approaching $10^{-1}\,\mathrm{m^2\,s^{-1}}$) are found between channels 30–40 km from the ice margin which we interpret as arising due to channels draw down water from surrounding areas. In winter, $q_c$ is lower throughout the domain, and the highest sheet discharge ($\sim 1 \times 10^{-3}\,\mathrm{m^2\,s^{-1}}$ Figure 2B) is found in patchy areas within $60\,\mathrm{km}$ of the ice margin"

which we hope addresses the confusion.
* * *
**Comment 2.74:** 337: as earlier, comments about channels confuse the issue, which is about velocity in sheet water. What does "reflects the concentration of drainage" mean, in this context? It is an interpretation of a mechanism that would account for a particular trait of Vw, but would help if it were explicitly presented this way. Focus on presenting the patterns in the main variable, and being clear in expressing spatial relationships – I'm not sure if you are reporting a contrast between summer and winter or not, in either spatial terms or velocity magnitude. I suggest also using the same units/terms as used in the figure, i.e. 15 x 10-4 rather than 1.5 x 10-3.

**Reply**: As described above, discussing $q_s$ we have added the phrase "...which we interpret as arising due to channels draw down water from surrounding areas" and we have deleted references to channelisation entirely from our discussion of $V_W$ and fixed the units as suggested.
* * *
**Comment 2.75:** 341-2: superscript s-1. Also lines 348, 352, 369 – check elsewhere.

**Reply**: We have corrected all instances where this error occurs

**Comment 2.76:** 342: Can you be more specific about the channels that persist over winter – how many? Or what proportion of the summer total?

**Reply**: We have added this as suggested to Line 423.
* * *
**Comment 2.77:** 343/4: suggest move the two sentences about Fig 3A to here, and open with it. By closing the reporting of fig 3 with a "nothing to see" case, all the interest and power of the findings about biannual signals is lost – much better to end with these interesting cases.

**Reply**: We have moved the two suggested sentences, making minor text changes in order to smoothly accommodate them.
* * *
**Comment 2.78:** 351: sharp increase in overburden% - not overburden pressure. There are numerous cases in the next paragraph(s) also where "overburden" or "overburden pressure" ought to be "overburden%", or alternatively "water pressure".

**Reply**: We have modified this throughout the manuscript in order to be more internally consistent.
* * *
**Comment 2.79:** 351: "trends towards" – can you say if this is an increase or decrease or values hover around one value?

**Reply**: We have replaced "trends" with "reduces" in order to make our meaning clearer.
* * *
**Comment 2.80:** 353: with only a small increase in overburden% ... and little taken up by sheet flow?

**Reply**: We have added this as suggested.
* * *
**Comment 2.81:** 356: this is confusing, suggest "with no overwinter channels evident in the central area while lateral margin channels persist over winter".

**Reply**: We have replaced the confusing text with:

" In any given year, channels will persist through winter in either the central third of the lobe or in the remaining two thirds of the lobe"
* * *
**Comment 2.82:** 357 + throughout: I strongly suggest sticking to expressing pressure either as water pressure or as overburden%. It is enough with two alternative ways of expressing the same thing. There is no particular need to express the same thing in terms of effective pressure – it's confusing at best and erroneous at worst. Replace here with e.g. "a persistent area of high basal water pressure (overburden% approaches and exceeds 100%)..."

**Reply**: In response to point 2.78 we have changed this throughout.
* * *
**Comment 2.83:** 360: increase in water pressure to 120% of overburden

**Reply**: The symbol was intended to convey nuance in the pressure change, we have replaced this with "...pressure up to a maximum of approximately 120 %..."
* * *
**Comment 2.84:** 360: "associated with" is vague, it's not clear what you actually mean in either case you refer to. Suggest delete this phrase here and be more specific later, where necessary. E.g. node 3842, located on a channel onset site. E.g. node 16402, located in between channel onset areas.

**Reply**: To address this point, we have made several changes. The modified now reads:

"demonstrates the seasonal evolution of two nodes in this area, each nearby to channel systems"

The phrasing about node 3,842 (already modified in relation to comment 1.17) now reads:

"At node 3,842, chosen to as representative of surrounding nodes at the onset of a channel"

and finally, the line about node 16402 now reads:

"which is located ∼0.7 km from a murtoo field between the onset of channels."
* * *
**Comment 2.85:** 371: "remaining at 80% of overburden" – no need for another way of expressing pressure!

**Reply**: We have modified this line to read: "Here, the system is effectively inert, with $overburden_\%$ remaining $\approx 80\%$"
* * *
**Comment 2.86:** 373: "To explore behaviours potentially associated with murtoo formation..." ??

**Reply**: We have adopted a slightly modified version of this suggestion. The lines (458–459) in question now read:

"We explored behaviours potentially associated with murtoo formation by focusing on nodes 40–60 km from the ice margin, within the zone thought to be associated with murtoo formation ∼12 cal. ka..."
* * *
**Comment 2.87:** 375: "...of a mapped meltwater route that also hosts...". Can you also give n=? in each case with the definition of the group (i, ii, iii)? You should also note again here that these meltwater routes are taken from Ahokangas et al., and note how you treat eskers (part of a meltwater route, or separate?).

**Reply**: The $n =$ values are reported in the methods already, and on the line after but we have repeated them here. We have also added the suggested clarification so that it now reads:

"As noted in Section 3.2, group ii may also include eskers ('channelised routes' in Ahokangas et al., 2021) as these are often coincident with meltwater routes."
* * *
**Comment 2.88:** 377: does "throughout the year" mean "mean annual"?

**Reply**: We have added "...and *at every point* throughout the year". We hope this clarifies that throughout the year means at every point during a year. It is true that "mean annual [$overburden_\%$]" is also true in our data, however, that is not what Figure 4 (now Figure 5) explicitly shows.
* * *
**Comment 2.89:** 382: is the statistical difference between groups significant for each of the 4 variables, considered separately? Or in combination?

**Reply**: We have added "for each of the four parameters discussed here" to clarify that it is the former case.
* * *
**Comment 2.90:** 388, 393, 396: could you be more specific where you state a variable is "significantly higher" or lower? In this context, it is easy to misinterpret significantly higher for substantially higher; a difference can be small, but statistically significant. What is meant here? Generally, the blue/orange variables in Fig

4, 5, A32 look pretty similar to each other, so any differences seem like they would be small, but statistically significant – can you express this more accurately and specifically in the written text?

**Reply**: We have rewritten this paragraph to clarify that the significance is exclusively statistical and is not a comment on the value of the difference. To address that point, which we agree is a useful one to make, we also have added an approximate value of this difference. This updated paragraph can be found between Lines 472 and 488.
* * *
**Comment 2.91:** 400: what do you mean, "to best apply the model to a palaeoglacial setting"?

**Reply**: We have removed this text as part of our discussion overhaul
* * *
**Comment 2.92:** 404: "water is efficiently evacuated" (or, water pressure is dissipated??)

**Reply**: We have changed this line (Line 495) to now read: "no channels longer than one km are formed and water is instead more readily transmitted through the distributed system at relatively low pressures ($overburden_\% < 60\%$)"
* * *
**Comment 2.93:** 06 + throughout: as above, express pressure in terms of either water pressure or overburden%

**Reply**: We have changed this specific case to "high $overburden_\%$" and have adopted the same throughout
* * *
**Comment 2.94:** 408: "At a minimum sheet conductivity of 10-5..." – symbol not needed since you wrote out in words

**Reply**: We have deleted the errant "of" but left the mathematical symbol in the brackets for consistency with how we report all sensitivity tests.
* * *
**Comment 2.95:** 411: KC should be kc (small letters)

**Reply**: We have made this change.
* * *
**Comment 2.96:** 411 + throughout Section 4.2: Table 1 gives units for ks as m7/4 kg-1/2 and for kc as m3/2 kg-1/2. These are muddled and inconsistent (7/4 or 3/2) through the Section.

**Reply**: We have fixed the two incorrect units for $k_c$ to be $m^{3/2} \, kg^{-1/2}$
* * *
**Comment 2.97:** 412: "low pressure" – which variable? (Also line 429)

**Reply**: We have addressed each of these by fixing the reference to $overburden_\%$ only.
* * *
**Comment 2.98:** 420: "the location of which" – what does "of which" refer back to?

**Reply**: Line 510 now reads: "...approximately 25 channels extend up to 50 km from the ice margin. The location of *these channels* closely follow the position of high discharge moulins...".
* * *
**Comment 2.99:** 448-455: superscript yr-1. Note that elsewhere in the text the unit for years is given as a.

**Reply**: We have fixed this to be superscript, and also fixed the one erroneous year notation we could find.
* * *
**Comment 2.100:** 452: what do you mean by "more tightly constrains the observed summer water pressure..." ? I don't follow.

**Reply**: We have modified this (Line 543) to now read: "In addition, the transient velocity results in a spatial distribution of $overburden_\%\approx100\%$ in stronger agreement with the contours of 40–60 km from the ice margin compared to the baseline scenario"

We hope this addresses the uncertainty.
* * *
**Comment 2.101:** Section 4.2 overall: see comment above (problematic text) – this sections reports the end member behaviour and selected middle option of the range of sensitivity tests, but doesn't anywhere evaluate these outputs. How stable/robust are the results presented for the baseline case? What do we learn from these sensitivity tests?

**Reply**: As described above we have now added description into Section 4.2 demonstrating which outputs are less realistic and therefore pointing towards the baseline model being the most appropriate choice. Sensitivity tests are an important parts of exploring the responses of the model and therefore we argue these outputs are a useful addition
* * *
**Comment 2.102:** 5.1 header: seasonal drainage is implicit in some parts of your discussion in which you compare winter and summer conditions, but seasonality isn't a trait you explicitly dwell on and explore. This doesn't seem a particularly appropriate header (or could be made more so by extracting aspects of seasonality for dedicated discussion).

**Reply**: In rewriting the discussion we now have several new section names, which are:

**"5.1 Catchment-scale hydrological configuration"**

**"5.2 Comparison with glaciofluvial landforms"**

**"5.3 Comparison between model outputs and mapped murtoo locations"**

and

**"5.4 Limitations and future work"**
* * *
**Comment 2.103:** 460: "...demonstrated by Kirkham et al., who evaluated hypotheses..." – in its present formulation, I moved onto the next phrase + reference as a new item, not connected to Kirkham.

**Reply**: We have removed this text, which on reflection read more like an introduction. The reference to the work of Kirkham et al., remains in the intro but this specific text no longer features in the discussion.
* * *
**Comment 2.104:** 472: do papers by Hewitt, Schoof etc not include transitions between channelised + distributed systems (and compare to landform-based predictions of geometry)?

**Reply**: As above, we have removed this section, but in this sentence we were explicitly referring to the inherently channelised area-routing algorithms and their application in the palaeo setting. Hewitt & Creyts (2019) (A model for the formation of eskers; *GRL*) does describe a model esker formation, however it concerns a single channel, and does not include a sheet component to the model. Hewitt et al., 2011 does include a continuum description of the distributed system coupled to a single channel (and indeed is what GlaDS is based on, together with the Schoof 2010 description of channelised drainage). They do make a comparison

to esker spacing based on scaling relationships. We have added this to the introduction, to which we have also added references to the work of Boulton, Hewitt, and Beaud.
* * *
**Comment 2.105:** 474: does "as" mean because, where, when, ... in this context?

**Reply**: We have modified "as" to "where" to clarify we are referring to a spatial transition
* * *
**Comment 2.106:** 475: what does "define a parameter space for basal hydrology models" mean, specifically, in this context?

**Reply**: We have significantly rewritten this section and as such this line no longer exists.
* * *
**Comment 2.107:** 479: I'm having trouble visualising "vertically arcuate along a comparable path to the surface" – can you rephrase?

**Reply**: We have changed this on Line 562 to:

"...are arcuate at a similar curvature to the surface slope..."
* * *
**Comment 2.108:** 483: delete "Instead" – doesn't seem like there's a contrast

**Reply**: We have changed "instead" to "accordingly" and also tweaked the following sentences (Lines 570–571) so that they now more clearly communicate the idea that 40–60 km from the ice margin is the likely area of murtoo formation
* * *
**Comment 2.109:** 485: "...gradients limit the growth of channels..." (?)

**Reply**: We have added "hydraulic potential gradients" to clarify the effect low surface gradients have on the basal hydrological system.
* * *
**Comment 2.110:** 486: permit (delete s)

**Reply**: We have adopted this change
* * *
**Comment 2.111:** 490: "Across the full domain <70km from the margin" sounds like you've got two mutually exclusive areas. I think you perhaps mean within the width of the domain and within 70km of the margin ?

**Reply**: We have removed this specific text as part of the discussion rewrite and have been more precise in our description of processes within this area.
* * *
**Comment 2.112:** 494: suggest deleting "when plotted as a summer average" and change previous line to "summer water pressure"

**Reply**: We have removed this specific text as part of the discussion rewrite, but anywhere we discuss summer water pressure we have been sure to adopt the suggested wording.
* * *
**Comment 2.113:** 494: Elsewhere in the downstream zone, water pressure ... ?

**Reply**: We have removed this specific text as part of the discussion rewrite.
* * *
**Comment 2.114:** 496: insert ii) since you started a list with item i) – and ii) the limited... dropping below 75% of overburden

**Reply**: We have added the missing "ii)" on line 590.
* * *
**Comment 2.115:** 502 and next two paragraphs: these read more like a summary of the results than a discussion. What's new, what key insights have you gained into murtoo formation and/or murtoo vs channel hydrology? What about all the murtoos upstream of the 40-60km band?

**Reply**: In rewriting the discussion extensively we hope to have addressed the suggestion that this is just a summary of the results, instead we focus on exploring GlaDS ability to reproduce the conditions associated with murtoo formation. We have also added a line (586–588) about the implication for Murtoos beyond the 40–60 km band—we conclude these formed as the ice margin retreated, please see our response to the previous comment on this subject above.
* * *
**Comment 2.116:** 506: if "channel discharge" is only ever close to but not exceeding the threshold for being designated a channel, how does that entity have a "channel discharge" and be measured as Qc? In fact, I should have asked this earlier with the presentation of results and figures e.g. Fig 3, 4, 5 – if a channel isn't a channel until it has a discharge of 1 m3/s, then how can a channel have discharge Qc below that?

**Reply**: We have addressed this query in response to Comment 2.56 concerning whether or not different equations apply when channel discharge exceeds $1\,\mathrm{m^3\,s^{-1}}$. But to address it again specifically here, every edge of the mesh in GlaDS is allowed to have a non-zero discharge at all times, so is in effect *always* a 'channel' of sorts. However only a few edges will ever reach an appreciable discharge and so it is necessary to set a threshold above which an edge (with non-zero discharge) is said to be a meaningful channel. This threshold is arbitrary, but in setting it we follow the work of every other paper which has used GlaDS to explore drainage dynamics.
* * *
**Comment 2.117:** 515: what is it that suggests that cavity expansion necessary for murtoos is captured in your model? Expand on "which suggest". I think you also need to expand on the argument that murtoo formation requires the existence of small channels, as a pre-requisite. Is that the case, or have I extracted the wrong idea from what I think you're alluding to? (This is sort of stated in the introduction to murtoo sedimentology, and on the previous page you refer to cavity enlargement, but does a cavity or conduit mean – in model terms – a small channel?)

**Reply**: To the first point, we have added an explicit link to the maximum cavity height of $1\,\mathrm{m}$ as suggested by Hovikoski et al., 2023 and Mäkinen et al., 2023. The text in question (Lines 580—581) now includes:

"close to the maximum cavity height of $1\,\mathrm{m}$ inferred boulder distributions in the upper slope of murtoos".

We hope to have addressed the second portion relating to the hypothesis of limited cavity expansion throughout the text through our other modifications.
* * *
**Comment 2.118:** 526: "such landforms relating to meltwater drainage have been mapped..."

**Reply**: This specific wording was removed in the discussion rewrite
* * *
**Comment 2.119:** 531: there is quite a lengthy history of data-model comparison with regards to eskers, and some of that work is likely relevant here (also in the opening of the Discussion, where you also address

data-model comparison) e.g. papers by Boulton 2007-2009, by Hewitt 2011, Hewitt & Creyts 2019; work by Flavien Beaud 2016, 2018 among others

**Reply**: We have added references to several of these papers in the introduction, and the specific text referenced here has been removed as part of the discussion rewrite. However, we do continue to make the point that mapped glaciofluvial make a useful means of comparison for our model outputs. The new text can be found on Line 630 to which we add references to the work of Boulton et al., 2009
* * *
**Comment 2.120:** 535-6: repetitive – suggest "...grid-based models, but the exact location...". "The spacing of channels, however, remains robust...". Delete the early part of the second sentence.

**Reply**: We have removed this specific text within the discussion rewrite. We continue to include text about this subject (the mesh dependency of channels) at the end of section 5.2.
* * *
**Comment 2.121:** 539: does this contradict the earlier result that I flagged in my general comments, that channel discharge is higher outside mapped meltwater routes?

**Reply**: We have corrected the earlier comment in response to Comment 2.6. There is no longer a contradiction.
* * *
**Comment 2.122:** 541: presumably the baseline run was selected in order to match the Greenland geometry, though – it's not really a result that this is the case, is it?

**Reply**: The baseline run was selected because it provides the most reasonable intermediate hydrology outputs e.g., channels of a length comparable to contemporary ice sheets, water pressures not far below overburden in winter. These parameters are the same as baseline models applied to contemporary ice sheets that have been tested against *in situ* data and therefore appear to provide a reasonable approximation of ice sheet subglacial conditions. The ice sheet geometry itself is not tested in the sensitivity tests, only the hydrology parameters that have been shown in other studies to be most important for drainage development (i.e. sheet and channel conductivity). We have changed this text to address this (Line 605–608):

"However, without extant ice in the FLDIL against which to test our models, we are unable to fully determine the correct parameters for our FLDIL domain. As a result, the baseline model was parameterised following existing work on contemporary ice sheets (see Section 3.1.2). As expected, the baseline model provides a range of seasonal water pressure and channel lengths that are similar to models of contemporary ice sheets validated with geophysical methods (e.g., Dow et al., 2020)"
* * *
**Comment 2.123:** 542: consistent language with earlier: "In the baseline model run, small channels..."

**Reply**: We have removed this specific text within the discussion rewrite. We have attempted to be consistent with this in the new discussion
* * *
**Comment 2.124:** 558: "between our model results and the observations beneath Greenland" ? – "those" implies model results beneath Greenland...

**Reply**: We have made the suggested change and also added a specific reference to observations on Line 612, which now includes Van de Wal et al., 2015.

**References:** Van de Wal, R.S.W., Smeets, C.J.P.P., Boot, W., Stoffelen, M., Van Kampen, R., Doyle, S.H., Wilhelms, F., van den Broeke, M.R., Reijmer, C.H., Oerlemans, J. and Hubbard, A., 2015. Self-regulation of ice flow varies across the ablation area in south-west Greenland. The Cryosphere, 9(2), pp.603-611.
* * *
**Comment 2.125:** 561: "in these systems" – does "these" refer back to the distributed system, channels, or both? Be specific.

**Reply**: We have replaced "these systems" with "in the distributed system" on Line 614
* * *
**Comment 2.126:** 571-2: this sentence repeats the opening sentence of the paragraph, but more specifically and concisely – suggest delete the opener or replace it.

**Reply**: We have deleted the opening sentence as suggested.
* * *
**Comment 2.127:** 572: overburden%

**Reply**: We have corrected this.
* * *
**Comment 2.128:** 576: as above, if the "channel" has discharge orders of magnitude below the threshold to be a channel, how can these be considered channels?

**Reply**: The order of magnitude was an error, and we have updated the text to clarify that the channels seen to persist through winter are very close to the arbitrary threshold we apply to channel discharge, and that a slightly higher threshold would have excluded winter channels. Lines 701-702 now read:

"Channels persisting through winter months tend to operate at very low discharges of $1$–$3\,\mathrm{m}^3\,\mathrm{s}^{-1}$, and would not be categorised as channels with a higher threshold"
* * *
**Comment 2.129:** 583-4, + preceding + following paragraph: you argue biannual behaviour in the model output is a consequence of internal model dynamics (i.e. an artefact) yet then offer an explanation for it in glaciological terms. What is your argument? And what is the relation to a patchy murtoo distribution?

**Reply**: We have attempted to clarify the relevant paragraphs to say that the biannual forcing arises through a combination of our fixed climate (giving rise to the repetitive biannual signal), but that the spatial distribution of these channels arises because of divergences in the ice flow direction, which is also important in the generation of glaciofluvial features in the FLDIL.

The relation to the patchy distribution of murtoos is as follows: meltwater routes are distributed evenly across the FLDIL, but murtoos are confined to the outer two thirds of the lobe and absent from the central third. Every winter, channels persist either in the central lobe or the outer two lobes, alternating each year. These winter channels affect the water pressure into the following summer. In winters where channels in the central third of the lobe persist, they influence the overburden of meltwater routes upstream of these channels in the following summer, but they have no affect on murtoo routes because there are none present in the centre of the lobe. In the following winter, winter channels affect summer water pressure in both murtoo routes and meltwater routes, giving rise to the significant differences between murtoo routes and meltwater routes considered over the whole of our model run.

We have updated the paragraphs in the discussion (Section 5.3) to address this observation, which is likely routed in both model setup and glaciolgical reality.
* * *
**Comment 2.130:** 623: study area

**Reply**: We have made this change, now found on Line 781
* * *
**Comment 2.131:** 631: while you have performed some sensitivity runs, neither your Results nor Discussion section explicitly analyses or evaluates those outputs, so it is false to say that "By sensitivity testing... we demonstrate...". You have selected the baseline run as your favoured answer.

**Reply**: When rewriting the conclusion we removed this specific phrase, but to reiterate the other comments regarding sensitivity testing, our baseline was chosen because it provides the most reasonable/plausible hydrology outputs including winter pressures and channel lengths when compared to contemporary ice sheets which have been tested against *in-situ* data.
* * *
**Comment 2.132:** 635: that murtoos arise because of "consecutive years of elevated meltwater volumes" has not been clearly demonstrated or argued in the Discussion. Your Discussion has presented biannual behaviour linked to winter persistence of some channels, and implied some related cavity/channel growth behaviour (although see above, 583-604), but your Discussion does not explicitly tie this to murtoo formation. "Elevated meltwater volumes" is also an ambiguous phrase: volume is not considered, as a model variable – do you mean pressure, discharge or velocity?

**Reply**: We have significantly changed this section and we hope to have addressed this comment in doing so. The logic regarding elevated meltwater volumes was incomplete and speculative. We have removed the text in question entirely.
* * *
**Comment 2.133:** Table 1 Channel conductivity: the main text refers to experiments with values 0.001, 0.05, 0.5. Ice velocity: given the inference of ice streaming, a tested velocity range of 100-200 m/yr seems rather limited, and low – why?

**Reply**: We have changed the table values to the $\times 10^x$ notation in the table throughout, and corrected the testing range. We do not (intentionally) infer ice streaming within the FLDIL. Instead, velocity was chosen to be comparable to melt-season surface velocity in land-terminating sectors of the Greenland Ice Sheet (e.g., Tedstone et al., 2015). We have added a sentence in the methods (Lines 367–368) clarifying the rationale for our chosen range of velocity values. This reads: "For basal velocity, we tested values between $100$–$200\,m\,yr^{-1}$ chosen to be comparable to GPS measurements of surface velocity across land-terminating sectors of the Greenland Ice Sheet (e.g., Tedstone et al., 2015)."

**Reference:** Tedstone, A.J., Nienow, P.W., Gourmelen, N., Dehecq, A., Goldberg, D. and Hanna, E., 2015. Decadal slowdown of a land-terminating sector of the Greenland Ice Sheet despite warming. Nature, 526(7575), pp.692-695.
* * *
**Comment 2.134:** Fig 1 I don't think that slope visualisation of terrain models is the most intuitive way to illustrate and distinguish murtoo morphology, especially for those not accustomed to working with terrain models or unfamiliar with these landform types. I would suggest a conventional hillshade, with illumination best suited to each panel. Also consider rotating the triangle symbols in the direction of murtoos? Caption: change to study area. A) "Murtoo fields identified by Ahokangas...". D) If murtoo fields are adjacent to the esker, then murtoo fields have not undergone an "abrupt downstream transition" to an esker – please revise, the description of a transition is misleading. E) "within the centre" – delete "at"

**Reply**: Figure 1 does not show a slope model. Each panel shows a multi-directional hillshade optimised for the mapping of glaciofluvial landforms in the region as described in Ahokangas et al., 2021. We have clarified this in the caption and adopted the other suggested caption modifications. The end of the caption now reads:

"All panels show a multi-directional oblique weighted hillshade based on 2 m LiDAR data (see Ahokangas et al., 2021, for details)."

We have added rotated triangles and also added ice flow directions as suggested by reviewer 1.

**Comment 2.135:** Fig 2 give units for qs, Vw and Qc. In the caption, move the sentence about channels as black lines to after (C), and say that this is the case in panels A-C. (Not just A). murtoo triangles aren't visible at this scale (appear as dots), nor is channel scaling for discharge (appear as black lines). I'd remove the murtoo/channel legend, and simply add to the last line of the caption that black dots are murtoo fields, from (which ref?).

**Reply**: We have made all of the suggested changes to what is now Figure 3.
* * *
**Comment 2.136:** Fig 3 label A, B, D, E next to coloured dots on panel C. consider flipping the inset graph so that distance = 0 to the right, for consistency with the map plot? caption: opening sentence, suggest "at four nodes over model years 15-25 in the baseline model run". And panel C) ... "in model year 19 (arbitrarily selected)." caption final sentence, should be qs

**Reply**: We have made most of the suggested changes to what is now Figure 4, we have not changed the orientation of the inset graph.
* * *
**Comment 2.137:** Fig 4 give units for qs, Vw and Qc colours: the red and orange are rather close, and the pink too – could you choose more distinct colours? Also, the legend shows solid pink while the graphs show dashed, and the legend describes as dashed purple. Please revise one or an other to be consistent. "Meltwater routes that do not contain murtoo fields (murtoo hosting)"... "routes which do host murtoos (murtoo free)" – both of these phrases in the caption are internally inconsistent. They are also (possibly?) inconsistent with the legend. Please revise.

**Reply**: We have changed the colours, and updated the caption in what is now figure 5, making sure to fix the colours for all other figures using this scheme. We have also revised the murtoo/meltwater route naming convention. We also have corrected the residual difference component of panel D, which had previously been a duplicate of the residual difference graph from panel B.
* * *
**Comment 2.138:** Fig 5. give units for qs, Vw and Qc same errors for murtoo hosting / murtoo free – the caption is internally inconsistent, and please check consistency with legend. I note that I have not checked legends/captions for all appendix figures, but at least A32 has the same error – please revise, and check other figures thoroughly.

**Reply**: We have made the suggested changes, and changed Figure A32.
* * *
**Comment 2.139:** Fig 6 caption line 2: delete "against"

**Reply**: We have deleted this against.

---

## Author Comment (AC2)

**Response to reviews: "Reorganisation of subglacial drainage processes during rapid melting of the Fennoscandian Ice Sheet"**

**Reviewer 1**

We thank reviewer 1 for their encouraging and helpful comments. Below we detail our response to each and in turn the changes we have made to our manuscript. We believe that these changes have improved the manuscript.

**General comments**

**Comment 1.1:** I found the title a bit misleading about the topic of the paper. I think the manuscript speaks to the past location of hydraulic conditions below the glacier, but I found little in the manuscript about "reorganisation of subgalacial drainage processes". Maybe "organisation of subglacial processes." Additionally, "rapid melting" does not seem like a big part of the paper, especially by reading the abstract.

**Reply**: Both reviewers queried the appropriateness of the title, we have modified the title to now be:

"The organisation of subglacial drainage during the demise of the Finnish Lake District Ice-Lobe".

We hope this better reflects the contents of the paper.
* * *
**Comment 1.2:** Throughout, but especially in the abstract and introduction, the authors make statements of "parame- terizing and testing models of subglacial hydrology", "basal hydrology in models", or "basal hydraulic conditions." These conditions or parameterizations can cover a wide range of features describing sub- glacial processes, include channel size or shape, water pressure, sediment transport, water velocity, distributed or channelized drainage. I believe the authors must be more specific and deliberate in describing the specific subglacial hydraulic features they aim to examine and link these features to murtoo development and persistence.

**Reply**: We have altered the abstract and introduction as suggested to be more explicit about our approach and aims. . Particularly with regard to our aims. The aims now state:
"

- Compare the subglacial hydrological conditions proposed for murtoo genesis and their associated landforms against model outputs from GlaDS.

- Sensitivity test GlaDS across a range of possible parameter values to explore the influence of these parameters on our outcomes in order to evaluate the potential of such models to be used to interrogate palaeo-hydrological systems more broadly and in turn motivate future work in this area.

"
* * *
**Comment 1.3:** Related to the last point, it seems that the description of murtoo formation could be improved. At times, it seems that hydraulic processes associated with different stages of murtoo development are in contradiction. An examples are given below.

**Reply**: We have addressed this by responding to the specific comments below.
* * *
**Comment 1.4:** Section 3.1.1: From my reading of this section, it seems that no diurnal forcings were used. While this makes sense in a paleo setting, I am concerned about the impact on results. For instance on the GrIS, hydraulic head can very over 150m and bed separation can be in excess of 25cm (Andrews et

al., 2014). It seems like such short temporal changes in subglacial hydrology could impact the formation of murtoos and move from one murtoo sequence to another over a very short time period (stages mentioned in Introduction, Hovikoski et al., 2023). I realize that application of such variable water discharges to hydraulic models can be difficult and in many scenarios not necessary. However, it seems like it could be important in this application.

**Reply**: Yes, we did not include any diurnal forcing. In reality, diurnal forcing may well be extremely important in murtoo formation right at the very onset of channelisation and is absolutely a target for future work on this subject. We have added it to our limiations. However, for the work presented here which aims primarily to describe the catchment scale processes in hydrological development, we note that in Werder et al., 2013, their diurnal experiment results in channels that largely follow the same spatial expression as in other runs. They do experience large fluctuations in discharge and pressure throughout the 24H cycle, however, at distances greater than $2\,\mathrm{km}$ from channels (and moulins), pressure fluctuations are minimised by the englacial storage term, which reduces the spatial influence of changing meltwater inputs at short timescales. Within the murtoo forming zone then, we expect that the pattern of pressure across the full-width of the domain (especially when averaged seasonally) would likely be comparable to our forcing runs without diurnal fluctuations, with changes in the pattern restricted near to channel heads and specific moulin inputs.
* * *
**Comment 1.5:** To the best of my knowledge GlaDS uses a semicircular channel geometry that is fixed (i.e. shape of the channel does not evolve). However, it seems that a key feature of murtoo development is low broad channels, potentially with changing channel shape. This seems to be discussed in Hovikoski et al., 2023 and in the manuscript at lines 511 to 523. Hooke et al., 1990 speaks to the effects of channel shape on subglacial hydraulics. I am aware that certain trade offs can be made between the friction factor and channel shape to end up with similar hydraulic characteristics. In some applications this may minimize the impact of the semicircular assumption. However, because sediment transport relationships are scaled to unit width of the channel, sediment deposition can be sensitive to the width of the channel floor, and thus the general shape of the channel. However, please comment on how this may impact the results. Is this such a consideration with the development of the drainage system? What are the impacts of the semi-circular and potentially fixed channel shape on the formation of murtoos?

**Reply**: GlaDS does indeed model channels as semi-circles, and we have added a line to this effect in the methods (Line 245) which reads:

"channelised flow—describing uniform, semi-circular Röthlisberger channels (R-channels) that are allowed to change diameter—along element edges".

Because GlaDS does not include any explicit treatment of sediment dynamics we are not truly modelling murtoo formation. Instead, we are attempting to reproduce the conditions associated with murtoo formation, particularly water pressure throughout the melt season. It is therefore difficult to evaluate what effects changing the channel geometry may have on murtoo formation. Further, without a detailed understanding of exactly how channel geometry varies in both space and time it is difficult to imagine how we might robustly explore this. Nonetheless, we have also more explicitly raised this as a limitation of our work. This can be found in Section 5.4.
* * *
**Comment 1.6:** More out of curiosity, how do the murtoo fields persist given the retreat of the glacier and the presumptive movement of the channelized drainage area up the glacier? Might retreat have occurred too rapidly to "destroy" the murtoos?

**Reply**: That is an interesting question, and one subject to ongoing investigation, but it is beyond the scope of this manuscript.

**Specific comments**

**Comment 1.7:** Ln 46: to the best of my knowledge Werder et al. (2013) examines hard bedded characteristics below glaciers, "subsurface material" needs clarifying. Would an alternative be sediment floored channels or canals.

**Reply**: We have replaced this reference with one more appropriate, specifically Chu, V.W., 2014. Greenland ice sheet hydrology: A review. *Progress in Physical Geography*, 38(1), pp.19-54.
* * *
**Comment 1.8:** Ln 59-71: the modeling work of F. Beaud is likely relevant here, as is the manuscript Hewitt and Creyts (2018) about eskers. Consider adding.

**Reply**: We have added the references to both in an expanded portion detailing previous modelling work in the palaeo setting, as well as work by Boulton et al. This can be found on Lines 88–100.
* * *
**Comment 1.9:** Ln 87: what does "more dynamic" mean also, I can imagine what "interlobate joints" are, but please clarify.

**Reply**: We have changed this sentence to clarify that 'more dynamic' means faster, and removed the reference to interlobate joints in response to a comment by Reviewer 2, the sentence now (Line 163) reads:

"...which are in turn concentrated in faster flowing, *warm-based* sectors of the FIS including the FLDIL..."
* * *
**Comment 1.10:** Enumerated 1-4 in Intro: I found this useful, and closely linked to Figure 10 in Hovikoski et al. 2023. Would the authors consider applying the cartoon in this manuscript? Additionally, it was difficult for me extract in the enumerated section the model output that would be indicative of this process in murtoo development. Please clarify. Might a table with one column of subglacial hydrology model output help?

**Reply**: Thank you for this suggestion. We have added a table (Table 1) linking the murtoo developmental stages to expected model outputs, and added callbacks to this table in our results and discussion. We have referenced the Hoviloski et al., paper more explicitly, but have not included their specific figure so as to avoid any copyright reproduction issues.
* * *
**Comment 1.11:** About points 1-2, I am a little curious about the idea that there is sediment deposition at the onset of melt. It seems that the conduit could be small, thus increasing water follow could increase sediment transport capacity, rather than cause sediment deposition. Although available observations of sediment transport are from the terminus, there can often be an increase in sediment transport at this time of the season.

**Reply**: We envisage that the subglacial water flow is in pulses against an overall backdrop of increasing discharge through the melt season. When flow fluctuates parts of broadening/low conduits become rapidly clogged by sediment-rich flows in shallow flow space and sediments are periodically slightly deformed by ice. During conduit widening, the marginal channels of murtoos seem to have the highest transport capacity. We have clarified our stages 1–2 to reflect this. Stages 1–2 are now:

"

1. With the onset of spring melt, pulses of water deposit the murtoo body within an increasingly large conduit. As each pulse increases in discharge and then wanes they promote the deposition of sand lenses, sinusoidally stratified sand, and poorly-sorted gravel, with silt commonly draping ripple-scale features. In this phase of formation, cobbles are the largest clast size, which places an upper limit on water depth of $\sim 25\,\text{cm}$ (Hovikoski et al., 2023).

2. As the melt season continues through summer, an increasingly enlarged pond forms in response to higher discharge. In turn, the increasing grain size indicates higher water velocity and sediments on the upper slope appear consistent with high velocity, upper-flow-regime deposits and the boulder size-distribution suggest a maximum flow space of 1 m (Hovikoski et al., 2023).

"
* * *
**Comment 1.12:** Ln 120: "higher water velocity" and "development of an englacial pond." To me these processes should not happen at the same place and time. Also, please define "upper-flow-regime."

**Reply**: Our original phrasing was a little muddled, though we note our original manuscript contained "development of an *enlarged* pond…" and not an "englacial pond". The expectation is that the pond already exists before higher velocity flows reach and form upper-flow regime structures. We have corrected the text to reflect this:

"3. As the melt season continues through summer, an increasingly enlarged pond forms in response to higher discharge. In turn, the increasing grain size indicates higher water velocity and sediments on the upper slope appear consistent with high velocity, upper-flow-regime deposits and the boulder size-distribution suggest a maximum flow space of 1 m"

Regarding the definition of sedimentological terms, in this section we briefly describe the sedimentological architecture of murtoos as background, referencing several papers with this as their main focus. Accordingly, we have not defined upper- (or lower-) flow regimes, both of which are relatively common concepts in sedimentology but whose specific meaning is adjacent to the main focus of our study. However, we have adjusted the section such that the deposits we reference are more specifically linked to the regime in which they form so that the line (189–192) now reads:

"proximally is comprised of alternating sequences of glaciofluvial deposits, with current ripples (formed in low discharge, lower flow regimes) giving way to transitional cross-bedding (transitional flow regimes), and antidunal sinusoidal lamination (formed in higher discharge, upper flow regimes;…"
* * *
**Comment 1.13:** Figure 1: Could estimated glacier flow lines be added?

**Reply**: Yes, we have added arrows indicating the approximate ice flow direction to each panel in Figure 1 and updated the caption accordingly.
* * *
**Comment 1.14:** Ln 195: "modified digital elevation model"… can the section where this is described be referenced?

**Reply**: We have added the section reference so that the line (Line 235) now reads:

"Then, using GlaDS parameterised by this input ice geometry and a modified digital elevation model (DEM) of the region (see Section 3.1.1)"
* * *
**Comment 1.15:** Table 1: I am curious if "mean annual velocity" is really an "input" or a model result or output, given the coupling with ISSM.

**Reply**: In this work, we are not using GlaDS coupled to ice dynamics. Instead, ice velocity (and other ISSM parameters, such as the ice rheological properties) are effectively model inputs. We have clarified this in response to one of Reviewer 2's comments (Comment 2.52:). The line to this effect (Line 270–271) now reads:

"Finally, in the iteration used here, GlaDS is not coupled two-ways to a model of ice dynamics, and instead we prescribe an ice velocity and geometry that is not variable in response to hydrological forcing"

**Comment 1.16:** "Fixed cross section" or "to the bed at every node." Does this go well here? or is somehow part of experimental design?

**Reply**: We have left these in place, as we are describing the GlaDS model design and believe this to be the most relevant section for this.
* * *
**Comment 1.17:** Ln 361: "At node 3,842": maybe make clear that these nodes are representative of their surrounding.

**Reply**: We have added a clarification to this effect. The sentence now reads:

"At node 3,842, chosen to be representative of surrounding nodes,..."
* * *
**Comment 1.18:** Figure 3c: should "D" be written as distance? also it seems like this is the end of the melt season of one year. Would it make sense for an "average" to be represented? Also would it make sense to add A-E in the plot in C as to clarify which plots go with which points?

**Reply**: We have changed the inset plot in what is now Figure 4C to read 'Distance from the ice margin (km)'. We have also added A–E labels into the plot as was also suggested by reviewer 2. We chose to illustrate the end of the melt season to give an idea of what channels looked like spatially at the end of the melt season and to give context to the time-series plots from four points. We also note that the summer average model condition is shown in what is now Figure 3.
* * *
**Comment 1.19:** Ln 505–506: why $10^0$ in one line and 1 in the next?

**Reply**: Thank you for noting this, we have changed the former to keep consistency.
* * *
**Comment 1.20:** Ln 515: "limited cavity expansion" might this be channel floor width?

**Reply**: We have changed the line (now 584–586) to now read: "however the agreement in dimension suggests that the limited cavity expansion or restricted channel floor width within which murtoo form is captured within our model"
* * *
**Comment 1.21:** Ln 520: "The reason... sediment supply..." My initial reaction upon reading this is that I normally do not consider a distributed drainage system able to transport large amounts of sediment, thus I am unsure about how sediment supply up glacier could impact the results here.

**Reply**: Murtoos occupy a semi-distributed environment where movement of subglacial sediment is an important factor as indicated by the sedimentological studies of murtoos. We have expanded on this idea within our rewritten discussion, which can be found on Line 690–696:

"Murtoos appear to form within a semi-distributed drainage environment, and sedimentological studies indicate the movement of sediment is important in murtoo formation (Peterson Becher and Johnson, 2021; Mäkinen et al., 2023; Hovikoski et al., 2023). The reason that murtoos are not present in an area of the FLDIL where our modelling suggests they should form may be a preservation issue or due to limited sediment supply. Sediment cover in this area is very thin, and the large areas of exposed bedrock likely limited the supply of sediment from which murtoos could form (Figure A1B), an interaction not yet accounted for in our modelling. Modern lakes are also abundant in the centre of the FLDIL and may also act to mask murtoo routes."
* * *
**Comment 1.22:** Ln 524: "More broadly" good pun after speaking of broad channels... "More generally"

**Reply**: The pun was unintended and we have changed the text as suggested.
* * *
**Comment 1.23:** Ln 544-549: I might be missing something. However, melt water input location also seems like a control. For instance, Gagliardini and Werder 2018 may speak to this.

**Reply**: We have added this reference and a pointer to lower crevasse density at higher elevation also inhibiting meltwater input. The line (570) now reads:

"shallow surface gradients engender low hydraulic potential gradients, while low crevasse density limits meltwater input to the bed (Gagliardini and Werder 2018)"
* * *
**Comment 1.24:** Ln 558-567: From reading this paragraph, the authors seem to point out the differences between GrIS and the runs here. However, I seem to miss the analysis of the causes of this difference between the two systems.

**Reply**: We expect that the difference arises because of the low-relief of our domain. We have added a sentence to this effect on Line 621–623 which reads:

"The FLDIL is relatively low-relief compared to the steep margins of Greenland (e.g., Wright et al., 2016), and the shallow topography may act to reduce the hydraulic gradient between distributed and channelised drainage."
* * *
**Comment 1.25:** Ln 593: "sub-lobes they bound" something funny grammatically, also I am not sure what is meant.

**Reply**: We have modified the sentence (now on Line 720) to read:

" As a result, landforms within the FLDIL have previously been divided into three sub-lobes. The boundaries between these three sub-lobes are demarcated by particularly large esker deposits suggesting a concentration..."
* * *
**Comment 1.26:** Ln 606: "1-2 day. . . walltime?"

**Reply**: Yes, thank you for noting this. We have clarified to this effect which now reads: "could run to completion *with a walltime* of 1–2 days while"
* * *
**Comment 1.27:** Ln 613: $\sim 0.75°$C how much more water does this result in?

**Reply**: Raising the model MAAT by a uniform $\sim 0.75°$C changes the mean moulin discharge (at every non-zero moulin through the full model run) by $\sim 10\%$ from 2.544 to $2.774 \, \mathrm{m^3 s^{-1}}$ (an upper limit based on running our PDD scheme 5 times). However, this represents an upper limit because uplift and tilting was non-uniform in our domain and the actual change in domain-wide MAAT is likely to be lower. We have not reported this in the manuscript, but we have changed the wording to make clear that the $\sim 0.75°$C is confined to the areas of highest uplift. These changes can be found in both the methods (Line 280–285) and the limitations (Line 730–734) sections.
* * *
**Comment 1.28:** Ln 627: "macro conditions" what are these precise conditions?

**Reply**: In response to several comments from Reviewer 2 we have significantly changed our Conclusions section, in doing so removing this term and defining the conditions to which we are referring.

**Comment 1.29:** Figure A2. what is a median discharge? Also, it seems like units are missing.

**Reply**: Thank you for noting this, we have rectified both the missing units and clarified what is meant by median discharge.

---

## Referee Report (RR1)

**Review: The organisation of subglacial drainage during the demise of the Finnish Lake District Ice Lobe – Hepburn et al.**

**General comments**

I think this is a much improved manuscript from the version I saw last (first version). The logic of the paper is much better presented, clear to follow, and much of the heavy language and confusing constructions have been addressed and resolved. I think the work is a good illustration of how the geomorphological record of former glaciation can be used to test ice sheet/hydrology models and give confidence in their wider utility, as well as shed light on patterns observed in the landform record.

I see that you've given some attention to refining how you formulate your overall aim, and I think that it now works ("explore the ability of GLaDS, a process-based subglacial hydrology model, to explain murtoo formation in both space and time"). I read this, and your manuscript, as testing whether GLaDS can generate the right conditions in the right place for murtoo formation, and then learn something about how these conditions compare to other meltwater landforming settings. I think you could be a bit clearer about what your comparisons can provide, and what they can't. The comparison of distributions is somewhat circular because of the widespread and time-integrated distribution of murtoos in Finland: murtoos are hypothesised to from under high pressure in an area near the onset of channelisation, and the model produces a high overburden zone upstream of channelisation. We could conclude the model is good. Yet, if the conceptual model is wrong and they can form under different settings, then we could look to a different zone in the model output, still find murtoos, and therefore also conclude that the model is good. We can't confirm the hypothesis. In this context, the comparison of model seasonal behaviour against sedimentology is really important, because there are multiple predictions in a sequence that the model must achieve. A good match for sedimentology makes this a genuine test of the physical and conceptual models, and I think some words to this effect in section 5.2 and/or in the conclusions would strengthen the paper.

I think Section 5.3 needs a bit more work – there are passages I still find confusing, and passages where I think important findings are lost that should be brought forward more clearly. Some of these findings draw from section 5.2 also. In particular, I'm concerned that a key argument that is made, that there is a significant biannual difference between murtoo routes and meltwater routes, mixes two sets of findings and hasn't been explicitly demonstrated itself. I expand on this further below with some suggestions for how to improve the clarity of the arguments made in these sections. I think an additional figure that illustrates the spatial distribution of contrasting modelled drainage behaviours would really help the narrative (also explained further below).

**Specific and technical comments, by line**

14-17: "Our model outputs match many of the predictions" would be a stronger (and appropriate) statement than the rather vague "represent". I think you can say more here about what you've actually found and discussed, and importantly, include that the model matches what we would expect from murtoo sedimentology (as above). For example: "Our model outputs match the general distribution of channelised drainage landforms such as esker and meltwater routes. Many of the predictions for murtoo formation are produced by the model, including the location… and,

importantly, the seasonal sequence of drainage conditions inferred from murtoo sedimentology. These conclusions are largely robust to a range of parameter decisions, and we explore seasonal and inter-annual drainage behaviour associated with murtoo zones and meltwater pathways."

32: "analyses… have been applied"

39-40: to better fit the development of ideas in this paragraph, I would move the phrase beginning "potentially including" to line 45, which would then flow "…ideal targets against which to evaluate subglacial hydrology models, potentially including processes variable at sub-annual scales and across the distributed-channelised transition". (Note, wherever it is place, suggest replace "those" with "processes", for clarity.)

46: landform genesis uncertainty arises from both fundamentally different concepts of how a landform is formed, and also spatial and temporal scales of formation.

59: odd punctuation: write out "length and spacing scaling relationships"

65-67: suggest deleting "modern" from "modern subglacial hydrology models" – unless there is a particular reason for this word, to compare with others?

73: I suggest changing the wording to "and likely represent time-transgressive formation over decades-millennia" – the current wording suggests that there is esker building taking place along the full length of a >10km esker over millennia, which I don't think is what is meant.

74: typo – glaciofluvial

84: published as Peterson et al. 2017

91: "closer than 40-60km to the…"

93-116: in this paragraph, could you add references to the numbered developmental stages in Table 1, at the relevant point in the text? You begin this way (line 97: represents the first stage…), but it would be helpful throughout. E.g. change text in parentheses on line 99 to "developmental stage 2: Table 1", and thereafter at the relevant point just (Stage 3), (Stage 4)…

107: does "disappearance" mean erosion or non-deposition?

111-113: there are 3 phrases here beginning "final" or "finally" – suggest rephrasing so that only one event is "final".

132: "however" in this construction is not being used as a parenthetical aside. Suggest change to "…in the centre of the ice lobe where thin sediment cover may have limited…"

158-162: I'm not sure these sentences accurately describe what you present in the results and discussion. In my view, you: examine catchment-scale hydrology parameters and compare to murtoo formation predictions and distribution of channelised landforms (eskers); you specifically explore seasonal and inter-annual drainage parameters in the zone where murtoos are hypothesised to form; you investigate differences in modelled hydrology between observed murtoo and meltwater routes, and where no glaciofluvial landforms exist; and you test the sensitivity of your results to a range of parameters.

173: suggest delete "uniform" – it seems to contradict being allowed to change diameter

221: spell out MAT (mean annual or monthly air temp?)

242-3: "Monthly melt was kept fixed annually" – sounds a bit confusing, do you mean that it was kept fixed year on year, i.e. no inter-annual variability, each month's temp was the same each year?

245: again, "total monthly melt was converted to yearly melt rates" – this is also confusing, I'm not sure here what you've done or why

260-1: while the variables subject to sensitivity testing are clearly listed in the Appendix Table 1, and nicely explained through the following paragraphs, I would have found it helpful to have a clear summary list of those variables in the text. Consider adding a summary sentence here, or modifying line 206-1, along the lines of "Sensitivity testing was performed on: basal melt rate, moulin density and distribution, conductivity of sheet and channelised water, englacial storage, basal ice velocity, land or water-terminating ice, basal bump height, bed topography surface, and mesh geometry."

306: delete "between"

324-5: "and each peaks… and remains…"  Here, I also wonder why the peak in sheet discharge and water velocity occurs *adjacent* to a channel?

325-326: typo – $V_W$ not $W_v$ (three instances)

326: delete one "to"

327: you do have constraints (= landforms) on your model output, so I suggest amending to "Without independent constraint against which to compare our model output"

330: for more emphasis, I would suggest deleting "and", and breaking the sentence. Catchment hydrology "remains consistent across most of the sensitivity tests. Furthermore, sensitivity test results remain consistent with predictions for murtoo genesis."

331-3: I would delete the additional descriptors for each parameter (e.g. variable, modified, differences in…). It's redundant in the context of sensitivity testing, and risks confusing between an experiment in which a parameter varies spatially/temporally, vs one that has a uniform distribution of the parameter but it differs from the baseline experiment.

334: I got a little confused here and the next paragraph, since you've just stated that almost all parameters showed relative insensitivity but then go on to discuss how several parameters affect channel location. It would help to start a new paragraph on line 334, and open with a clear statement differentiating catchment-scale results with channel-scale results. E.g. "While catchment-scale trends are robust, the exact location of channels, and their length and local overburden%, vary between sensitivity tests."

336: "differences in channel location"

341: again to help the narrative, suggest open with "Besides channel location, channel length and overburden% vary considerably". And I would also be specific here about which sensitivity tests, not just "six of". E.g. "…vary considerably in our testing of sensitivity to conductivity".

347: again to help the narrative, suggest open with "For the other end member"

347: does the "and" in this line mean that both conditions are met (minimum sheet k at the same time as maximum channel k), or that this result arises when either one of those conditions is met? I'm guessing the latter since each has a separate figure reference. I wonder if there's a way to clarify this. (Same for line 351.)

357-60: I think a clearer way to argue this is that excessively long or short channels are considered 'invalid', on the basis of modern Greenland observations, and that an anomalous overburden% distribution is considered invalid on the basis of the conceptual model for murtoo distribution, and therefore the baseline terms are considered most plausible.

366: "the pressure conditions… are notably different"

367: suggest breaking the sentence to make it clear which channels (modelled or observed) you refer to in the final clause. "In Greenland, channels exist at lower pressure…, and the resultant hydraulic potential gradient…"

376: typo – FLDIL

Section 5.2
From your description of sites D&E, I struggle to see the difference in landform/hydrological context. You start with E, and describe it as being representative of channel onset nearby, and conclude that the model drainage behaviour is consistent with murtoo development phases (although murtoos themselves aren't evident here). Then you describe D as being located at the head of two channels, and near murtoos, and interpret that 1 of the 2 modes of drainage here is consistent with murtoo formation. I see that the drainage behaviour differs at sites D&E, but both can allow for murtoo formation and both have some relationship with channel onset, so I think this part of section 5.2 (e.g. lines ~389-406) would be better packaged if you instead: introduce D&E as both being in the zone hypothesised to favour murtoo formation, and that both display drainage behaviour that could accommodate murtoo formation, but they show different interannual behaviour; and note also that one node is close to an observed murtoo while the other is not.

I think this would better frame the later discussion of absence of murtoos where they're predicted, and the discussion of the biannual behaviour and its relationship to landform distribution.

Here and below: a figure, perhaps plotted in map view, of nodes that display biannual behaviour vs annual behaviour would really help this discussion and that in section 5.3.

392: wording is a bit awkward – I suggest "representative of nodes surrounding the onset of a channel" ?

402-5: there's a few awkward phrases here. Consider: "before quickly dropping to an overburden% that is elevated relative to the previous winter." "Years with an elevated overburden% are associated with lower Qc…". Put commas round "approaching 1 m3 s-1". Replace odd-numbered years with "We consider that the latter case is more consistent with…"

410: double "maximum"

411: "the channel remains active over winter".

418-9: suggest you add to the end of the sentence "that have been invoked to explain murtoo sedimentology".

422: since GLaDS is pervasively connected and therefore can't produce the rapid changes in flow that you describe, is this limitation more widely a problem for other modelling questions? Of course the model is a necessary compromise on certain aspects in order to be manageable, but does your work flag that this is actually important to implement more fully?

Section 5.3

Is the heading appropriate? I think something like "Comparing murtoo and meltwater route hydrology" is a closer description of what this Section considers.

Paragraph 1 is overcomplicated, and I think it is mis-framed as an evaluation of GLaDS ability to represent meltwater pathways – you do something more specific than that and it raises an interesting discussion about different types of behaviour. It could be much more succinct and clear if you frame the opening to this Section as: You further explore drainage behaviour in the zone hypothesised to be relevant for murtoo formation. You group all the nodes in this zone according to whether they are located among murtoos, meltwater routes, or neither. Murtoos and meltwater route distribution are based on Ahokangas et al, and you include eskers with meltwater routes, though you acknowledge there is no age control to say if they all form simultaneously (and in the relevant time-slice).

Line 448: "winter minima"

449: "do not intersect mapped glaciofluvial geomorphology"

450: "… are lower…"

P3 could be more impactful, and it's missing a clear discussion of all elements in Fig 6 – in the second half of the paragraph, you overlook Fig 6 and only talk about what's in the Appendix. I suggest, for all parameters, first comment on what is shown in Fig 6, then comment on what differences are apparent during specific months.

I also think there's some interesting behaviour in Fig 6 that you haven't commented on (but here would be the place for it), and I wonder if it would contribute to your discussion of biannual behaviour and/or differences in drainage mode within the same 'murtoo-favouring zone'. For all parameters, the murtoo data has a bimodal distribution, while the channels either have a single peak or are also bimodal – is this something you can comment on?

P4-P6 – I still find it hard to work through the arguments relating to the biannual drainage signal. It's an interesting behaviour, but I'm still not really sure what you are saying about it in relation to either space or landforms (or whether it's a model artefact that wouldn't be present in reality). I think there's a missing link in your arguments that relates to the statements on lines 472-3 and 481-2: that there is a spatial component to the biannual signal in murtoo route outputs, and a significant biannual difference between murtoo routes and meltwater routes.

I don't believe you've demonstrated a significant biannual difference between murtoo routes and meltwater routes. Section 5.2/Fig 3 show that two nodes in the hypothesised murtoo forming zone have different signals – one annual, and one biannual. You've concluded that both behaviours are suited to murtoo formation (albeit only one of the two modes in the biannual case). Section 5.3/Fig 6 show that murtoo and melt routes differ in behaviour within a season/year, but you don't demonstrate here that one or the other is characterised by a biannual signal. Therefore, your statements on lines 472-3 and 481-2 combine these two results, without demonstrating that the biannual signal is landform-specific.

I think paragraphs 4-6 (i.e. remainder of Section 5.3, from line 467), ought to be revised to sharpen the discussion about the biannual signal, its spatial distribution, and how it might relate to a specific landform type. As above, I think that a figure that demonstrates the "spatial component to the biannual signal" would really help – whether or not this spatial biannual pattern also has a connection to a specific landform type. This could be a map – nodes with an annual/biannual signal.

And/or an equivalent to Fig 6, but plotted for the two lateral or central parts of the lobe instead of by landform type?

Some further specific writing amendments…

In P4, there are far too many ideas all stuffed into one paragraph:
- I would move the opening to conclude P2 and open P3: murtoo & meltwater routes are both very different to where there are no landforms, and in this way GLaDS faithfully represents FLDIL drainage patterns. // This is further evident when plotted as a PDF. Here, however, you also see subtle differences between murtoos and meltwater routes.
- Start P4 by posing the problem that murtoos are absent where the model (both conceptual and physical) suggests they ought to form. Offer your geological/data reasons for this. Then…
- Alternatively, murtoo distribution could be related to biannual channel discharge behaviour (reported in section 5.2, Fig 3). Biannual signal is interpreted as due to channels persisting through winter; they likely influence the nearby system the following summer, when the initial melt input would be discharged by an already established efficient pathway. There is also a spatial component to the biannual signal in our model output. When channels in the central third of the FLDIL persist over winter, those in the outer two thirds do not – and vice versa. Murtoo distribution – absent in the central third – could be a reflection of this spatial control on winter channel operation.

This sets up the whole of the final part of your discussion in the context of murtoo formation (presence/absence), and there is a clear purpose to the biannual discussion, rather than discussing a quirk that seems to have emerged from your modelling whose connection to your research question isn't really apparent.

P5 – I also think this is unnecessarily wordy and buries the point of the argument. Consider re-ordering (and trimming) the ideas:
The appearance of winter channels isn't surprising: they are evident on Greenland. Yours operate at very low discharges (below an arbitrary threshold for classifying a "channel") but nonetheless exhibit this behaviour. However, the spatial pattern of winter persistence is unexpected. There is spatial variability in your climate forcing, which translates into spatial variability in meltwater input, though your input has no interannual variability that would explain a biannual signal in channel Q and overburden.

This leads directly into the next paragraph, in which you offer a solution.

Line 503 – if this difference between murtoo and meltwater routes that you refer to here is the presence of a biannual signal, then as above, I don't think you have demonstrated this. (If you only mean that they differ, as in Fig 6, then this is ok.)

504: "divergence… appears to…"

506: be specific – which portions of the model have channels that resist closure during winter? A figure would really help this spatial discussion. Where/how do these portions relate to flow divergence?

508-9: but even if the very regular biannual signal isn't evident any more, do certain sectors have more of a tendency for over-winter channel persistence than others? I'm not sure you intended to conclude this discussion with 'it's a model artefact', but that's how it reads.

509-514: this is a really interesting idea, that the sub-lobes approximately match the pattern of winter channel persistence. Again, could a figure that illustrates this be combined with one as suggested above, or would the same figure serve both aspects of this discussion? And can you offer a concluding sentence that might explain why large eskers bound zones with different winter persistence of channels? In terms of the writing structure, I would break these lines off into a new, final paragraph.

Section 5.4 to end

517: I think you've actually explored drainage in relation to more than one glaciofluvial landform – suggest pluralise

520: the absence of topography is an awkward concept. Suggest amend to "including the absence of any relief"

523: the wording here is a bit confusing about the direction/trend of change (up/down or forward/backward in time) – suggest something like "Assuming this area has been uplifted by a maximum of ~100m, the volume of melt… would have been higher during the YD due to higher temperatures at lower altitude."

537-539: this seems very specific and a poor final sentence – is it necessary or can you move it earlier in the paragraph? It seems more natural to end with the future work sentence.

In this section you focus on all the things that you haven't included. But you have done some fairly extensive sensitivity testing. I think it would be entirely appropriate to note that you haven't included all these factors and this introduces some uncertainty, but that you have tested many parameters and your findings are largely robust.

544: "The alternating sedimentological sequence"

557: delete "extending", you've already said the channels extend

Appendix:

Make sure the title of the article matches the final title.

Fig A15. I think the caption ought to read 'comparison of basal melt 7x 10^-3… against baseline 1x 10^-3 ?

---

## Author Response (AR2)

We would like to thank The Cryosphere Editor for and opportunity to revise our manuscript, and the reviewer for their constructive comments on our writing. We carefully considered all comments and suggestions, consistently adding them into the revised version of the manuscript. Below we list those comments and detail our changes in response to those comments.

**Response to reviews: "Reorganisation of subglacial drainage processes during rapid melting of the Fennoscandian Ice Sheet [The organisation of subglacial drainage during the demise of the Finnish Lake District Ice-Lobe]"**

**Editor**

We thank the editor for noting an error in our experimental reporting and for communicating a detailed and helpful series of suggestions for improving our manuscript. Below, we describe our changes in response to the experimental error, and in the response to reviewer 1 go in to more detail regarding our changes to the overall manuscript.
* * *
**Comment 0.1:** I have also found an error in your experimental design: The Wake and Marshall 2015 (WM15) paper used mean hourly data to determine their sigmaPDD not mean daily as you implicitly assume in your reasoning above. As such, there is no justification to use -5°C as your reference (0 point) value for computing PDDs. Your justification of your PDD formula is also misleading. WM15 found significant variation in skewness and kurtosis and their derived distribution ("M2") for PDD accounts for this and is therefore not Gaussian, but you make no mention of this when you cite them for your choice of sigma(Tmonth) that you apply to a Gaussian.

**Reply**: We have corrected the error in our experimental design, adjusting Line 239 to make clear that we are following van den Broeke et al., (2010) in setting our threshold to be -5°C for the PDD. We have also added a line on 235–238 to explain why we did not take into account the changes in kurtosis and skewness with the M2 distribution from the Wake and Marshall 2015 paper, reasoning they become more significant where $T_M <$-20°C. The line in question reads:

"We used $\sigma_M$ from Wake and Marshall (2015), but did not take into account variations in kurtosis and skewness with temperature, as these become significant where $T_M < -20$°C (see Wake and Marshall, 2015), temperatures below those derived from our depressed MAT. Instead we used the calculated $\sigma_m$ to add Gaussian noise to a daily temperature record estimated by linearly interpolating our depressed MAT record."

**Response to reviews: "Reorganisation of subglacial drainage processes during rapid melting of the Fennoscandian Ice Sheet"**

**Reviewer 1**

We thank reviewer 1 for the detailed second round of reviews, and we have changed our manuscript significantly in response.

**General comments**
* * *
**Comment 1.1:** While the authors have worked to address the comments, in my opinion, some work remains in making the manuscript more readable and addressing the underlying scientific questions presented in the manuscript's introduction.

The authors did not seem to discuss the role of greater seasonal variability or diurnal variability in water discharge in the revised manuscript, despite in being mentioned by both reviewers. I believe that this could be important in the sediment dynamics controlling the formation of murtoos below the glacier.

Despite the uncertainties of the variability of the hydraulic forcing, I believe that model application is generally rigorous and that method is appropriate to address the research objectives. However, the writing remains hard to follow and does not fully address the research objectives. Reviewer 2 spoke to this in the last review, and with the benefit of hindsight, I should have pushed this matter further as well in my original review.

In the first section of the results, for example, it seems that much of the presentation is about the sensitivities of the model and description of the base case as opposed to addressing the questions posed in the introduction. Likewise, much of the first part of the paper summarizes the state of the research, as opposed to creating an imperative to address the research questions. For example, the link between the murtoo formation stages in section 2 and the model output that could represent these processes is not clear to me.

In my opinion, for this to be an impactful and unique work the authors must show how their results fit into the two research objectives at the end of the introduction. I think that both aims, particularly the second, can be accomplished. However, the paper will need some substantial rewriting and streamlining.

**Reply**: We have extensively rewritten the manuscript with a view to streamlining and improving the readability of the manuscript as well as making it clearer what questions we are posing, how we have sought to answer them, and what our answers suggest. Changes have been made in every section, but guided by these comments and additional communication with the editor, we have removed substantial superfluous portions of the text, rewritten our introduction to better set up the research imperative, placed information of limited interest into the appendix, merged our results and discussion section, and tried to streamline the text throughout. In response to a specific suggestion from the editor, we have changed our aim to

"to explore the ability of GlaDS, a process based subglacial hydrology model, to explain murtoo formation in both space and time"

and have restructured our text around that single aim. In merging the results and discussion, we have paid particular attention to ensuring we describe our results in reference to this aim, rather than describing the model outputs in abstract terms and then moving on to the discussion. We hope the writing is easier to follow throughout.

Addressing the more specific comments about seasonal and diurnal variability, we have expanded upon why we did not use diurnal forcing in both our methods and discussion. In our methods, (Section 4.1.1, Lines 249–253), the new text reads:

"Without a detailed record of daily melt variability we neglect to include daily and diurnal changes in melt, which are known to drive rapid changes in hydraulic head on the Greenland Ice Sheet (Andrews et al., 2014).

Smoothing melt variability reduced model size and improved the stability of GlaDS over the ~27 year model runs, and we note that the inclusion of an englacial storage term in GlaDS acts to restrict the influence of diurnal variability to within 2 km of moulins with a limited influence on the overall pattern of channelised drainage (see Werder et al., 2013)."

and in the discussion (Section 5.2, Lines 419–426), in elaborating on why we do not see the periodic isolation of murtoo cavities, we have added:

"We did not include diurnal variability in our modelling on the grounds of model stability and the limited influence diurnal forcing has on catchment scale drainage in GlaDS (Section 4.1.1 & Werder et al., 2013). Diurnal forcing would be critical in order to represent rapid changes in the flow regime within murtoo-forming cavities. However, GlaDS is also a model in which the subglacial system is assumed to be pervasively hydraulically connected, and there is no mechanism which can lead to the hydraulic isolation of specific areas of the bed (e.g., Rada and Schoof, 2018; Hoffman et al., 2016). As a result, even if diurnal forcing were to be included, we do not expect to be able to reproduce the rapid changes in meltwater discharge necessary to form upper and lower flow regime deposits (see Section 2, & Hovikoski et al., 2023) or laminated muds in marginal murtoo channels (e.g., Ojala et al., 2022)."

We did not account for seasonal variability compared to the present day (a point raised by Reviewer 2) because our model domain represents the end of the Younger Dryas, when such seasonal variability gave way to a markedly warmer climate with similar seasonality to present. This text was present in the previous version, but has now moved to Lines 221–225.

**Specific comments**
* * *
**Comment 1.2:** Velocity. Throughout the manuscript please clarify if velocity refers to water velocity or sliding velocity

**Reply**: We have made this clearer throughout, specifying either water or basal at every mention of velocity.
* * *
**Comment 1.3:** Ln 1. glacier thinning $->$ increased glacier melt? Melt, in addition to thinning, impacts basal characteristics.

**Reply**: We have changed thinning to melting, which more accurately represents the point we are trying to make.
* * *
**Comment 1.4:** Ln 9-10. This sentence is strangely worded. I had the thought: if it is "hypothesised" then how can it be "ignored"

**Reply**: We removed this sentence whilst tweaking the abstract
* * *
**Comment 1.5:** Introduction- Put e.g. in front of some citations. van den Broeke et al., 2023 were not the first to establish widespread melt in Greenland and Antarctica.

**Reply**: we have changed this as suggested, now found on Line 31.
* * *
**Comment 1.6:** Ln 55. detailed treatment-> This challenge is also in part due to the computational resources, number of parameters, etc.

**Reply**: The wider text amongst which this was part has been removed, and this point with it.
* * *
**Comment 1.7:** model ability -> model ability of what? to represent water pressure? Location of channelized vs pressurized flow?

**Reply**: We have changed this specific text, but the point remains and the new text (line 65–66) now reads:

"we are not aware of previous work which has evaluated their ability to reproduce the subglacial conditions (e.g., water pressure, channel location) associated with glaciofluvial landform formation."
* * *
**Comment 1.8:** Table 1. Reference labels. i.e. qs, Vw...

**Reply**: We are not sure how to interpret this comment, but we have made sure to check that all of the symbols are correctly formatted.
* * *
**Comment 1.9:** Section 2. Title, I do not see how these two matters are connected, thus deserving their section together. The murtoo formation descriptions are pretty dense and I did not follow them. Maybe just include citations about the formation? Also, I was looking for a concise description of the model conditions that would be indicative of murtoo development (this is in Table1, which I appreciate and wish was addressed more directly in the results.)

**Reply**: We merged murtoo description with the study site in direct response to a comment on the previous round, but we have since separated them into section 2 (The glaciofluvial significance of murtoos) and section 3 (Study area). We have also significantly reduced bulk in this text, reducing it down to three paragraphs on murtoo location/geomorphology and murtoo sedimentology/formation sequence. In the results and discussion section, we now are more explicit about how certain model conditions are or are not indicative of murtoo development.
* * *
**Comment 1.10:** Ln 277. The GIA discussion could fit in the discussion at the end if it is important. To me, it distracts from other messages here.

*and the related*

Ln 703. GIA is discussed here. Streamline the manuscript by removing the above.

**Reply**: We removed the GIA discussion from the methods and left it in Section 5.4 (Line 523–527)
* * *
**Comment 1.11:** Ln 316. I do not follow what was done with the PDD model here. Were Braitwaite and Olsen used or not? Please streamline and clarify.

**Reply**: In responding to the Editors comment above, we have reworked this section to clarify that we did not use Braithwaite and Olsen.
* * *
**Comment 1.12:** Section 3.1.2- Maybe a sentence or two about the purpose of the sensitivity tests and how this will interact with the murtoo observations and paleo conditions.

**Reply**: We have changed the last sentence of the opening paragraph in what is now Section 4.1.2 (line 261–262), adding detail about assigning confidence to findings where present across multiple tests which reads:

"We can assign higher confidence to our baseline model when similar model outputs (e.g., similar channel lengths or patterns of water pressure) are evident across multiple sensitivity tests".

We have also added text in our restructured results and discussions section about why we have more confidence in some parameter runs and less in others.

**Comment 1.13:** Ln 364- Basal velocities -> Are these prescribed or output of ISSM? Please adjust the sentence to reflect this.

**Reply**: The text in question (now Line 283) now includes 'prescribed'
* * *
**Comment 1.14:** Section 3.2 - Much of this section could fit in the discussion if it impacts the results. Some matters, such as the model representing a single time slice, might fit in other sections as well.

**Reply**: We have largely removed the text in this section, and as suggested moved much of it to the discussion (e.g., see Lines 428–440).
* * *
**Comment 1.15:** Section 4.1.1 - This section should be reworked to link back to the research objectives at the beginning of the paper. While these results somehow seem reasonable, it is uncertain to me what the aim is. Given the content in Figure 2, it seems that channel distributions would be a reasonable place to begin.

**Reply**: We have heavily reworked this as suggested, doing so by folding the results into the discussion section and more explicitly linking our results to the research objective.
* * *
**Comment 1.16:** Section 4.1.2 - Consider moving to the first section- Could this be used to show the viability of the model?

**Reply**: As per above, we have reworked this section and it now falls into Section 5.3 in which we compare modelling to the mapped meltwater routes in the FLDIL. Our primary finding from this comparison, that GlaDS reproduces the concentration of meltwater in mapped murtoo/meltwater routes compared to areas of the bed without evidence of meltwater, forms the first part of this new section and we do believe this shows the model is viable. We then move on to possible explanations for the statistically significant differences between murtoo routes and meltwater routes.
* * *
**Comment 1.17:** Figure 3. Can these variables be linked back to conditions in Table 1 and discussed? This is what I was hoping for when I looked at Table 1.

**Reply**: We cannot easily link the contents of Figure 3 (now Figure 2) to the murtoo developmental stages in Table 1. Figure 2 is instead intended to show our modelling reproduces the spatial pattern of murtoo formation; namely that they form where water pressure is equal to or exceeds ice overburden 40–60 km from the ice margin, at the onset of channelisation. Table 1 describes the developmental stages of an individual murtoo, and we link this directly to Figure 3, which shows the evolution of specific nodes through time. We make direct comparison between the two in Section 5.2 (Comparison to the murtoo developmental phases).
* * *
**Comment 1.18:** Paragraph 464-470 - This is a hard paragraph to follow.

**Reply**: The content of this paragraph can now be found on Lines 452-466, which we have reworked with the goal of improving its readability.
* * *
**Comment 1.19:** Ln. 490 - This sentence is almost exactly like the one at the top of 3.1.1.

**Reply**: This text was removed as part of our rewrite.
* * *
**Comment 1.20:** Paragraph 553-572 - The findings of the model results are not discussed here. I am curious about the interaction of the result with this existing knowledge.

**Reply**: We have removed this paragraph, instead more explicitly linking the murtoo features to model output throughout Section 5.
* * *
**Comment 1.21:** Ln 580. For reasons regarding channel shape discussed in my last review, I believe that these findings about the channel radius are highly speculative.

**Reply**: Because GlaDS represents channels as a fixed semi-circle, this reference to channel radius was intended to communicate that channel radius compared well to the maximum cavity height inferred from supra-murtoo boulders. However, in response to this comment we have moved references to channel radius and our findings therein.
* * *
**Comment 1.22:** Section 5.3 Isn't this topic introduced in the paragraph starting at 573?

**Reply**: In Section 5.3 we were communicating direct comparison between murtoo routes and modelling outputs, whereas in the paragraph starting at 573 we were comparing predictions about murtoo formation (in terms of distance from the ice margin and proximity to channelisation). We have changed our results and discussion heavily, but the section which most closely matches this is now called 'comparing model output to meltwater routing beneath the Finnish Lake District Ice-Lobe' in an effort to clarify this comment.
* * *
**Comment 1.23:** Ln 690. Thin sediment, low sediment supply -> this needs a citation.

**Reply**: We have added a citation on Line 486 to Bradwell, T, 2013: Identifying palaeo-ice-stream tributaries on hard beds. Mapping glacial bedforms and erosion zones in NW Scotland, Geomorphology.

---

## Author Response (AR3)

**Response to reviews: "Reorganisation of subglacial drainage processes during rapid melting of the Fennoscandian Ice Sheet"**

**Reviewer 2**

We thank reviewer 2 for their detailed second round of reviews, which have helped to further improve our manuscript. Below we address the general comments and then list our response to each of the specific and technical comments in turn.

**General comments**
* * *
**Comment 2.1** I think this is a much improved manuscript from the version I saw last (first version). The logic of the paper is much better presented, clear to follow, and much of the heavy language and confusing constructions have been addressed and resolved. I think the work is a good illustration of how the geomorphological record of former glaciation can be used to test ice sheet/hydrology models and give confidence in their wider utility, as well as shed light on patterns observed in the landform record.

I see that you've given some attention to refining how you formulate your overall aim, and I think that it now works ("explore the ability of GLaDS, a process-based subglacial hydrology model, to explain murtoo formation in both space and time"). I read this, and your manuscript, as testing whether GLaDS can generate the right conditions in the right place for murtoo formation, and then learn something about how these conditions compare to other meltwater landforming settings. I think you could be a bit clearer about what your comparisons can provide, and what they can't. The comparison of distributions is somewhat circular because of the widespread and time-integrated distribution of murtoos in Finland: murtoos are hypothesised to from under high pressure in an area near the onset of channelisation, and the model produces a high overburden zone upstream of channelisation. We could conclude the model is good. Yet, if the conceptual model is wrong and they can form under different settings, then we could look to a different zone in the model output, still find murtoos, and therefore also conclude that the model is good. We can't confirm the hypothesis. In this context, the comparison of model seasonal behaviour against sedimentology is really important, because there are multiple predictions in a sequence that the model must achieve. A good match for sedimentology makes this a genuine test of the physical and conceptual models, and I think some words to this effect in section 5.2 and/or in the conclusions would strengthen the paper.

I think Section 5.3 needs a bit more work – there are passages I still find confusing, and passages where I think important findings are lost that should be brought forward more clearly. Some of these findings draw from section 5.2 also. In particular, I'm concerned that a key argument that is made, that there is a significant biannual difference between murtoo routes and meltwater routes, mixes two sets of findings and hasn't been explicitly demonstrated itself. I expand on this further below with some suggestions for how to improve the clarity of the arguments made in these sections. I think an additional figure that illustrates the spatial distribution of contrasting modelled drainage behaviours would really help the narrative (also explained further below).

**Reply** We thank reviewer 2 for their positive appraisal of our heavily reworked manuscript. We are particularly glad that our overall writing is now clearer.

We appreciate the suggestion for Section 5.2 and the circularity raised by comparing murtoo distribution to model outputs alone which sedimentology addresses. We have added the following text to the opening of Section 5.2 (Line 389–394):

"The widespread and time-integrated distribution of murtoos throughout our model domain complicates model validation as murtoo formation conditions remain uncertain. The ability of GlaDS to reproduce the hypothesised spatial pattern of murtoo formation (i.e., summer $overburden_\%$ $\approx$100% 40-60 km from the ice margin) alone cannot definitively confirm or refute the hypothesized formation process because murtoos are distributed across our model domain. In this context, comparing seasonal model evolution to murtoo

sedimentology (e.g., Hovikoski et al., 2023; Mäkinen et al., 2023) becomes particularly important as there are multiple predictions in sequence that the model must achieve (Table 1)."

We respond to the specific suggested changes in section 5.3 below (Comment 2.45–Comment 2.57), but in short we have adopted a more cautious approach to the overwinter channels and their biannual signal because we are unable to rule them out as a model artefact. We have also added a supplementary figure in order to illustrate the spatial distribution of winter channels.

However, we emphasise that in removing some of the text about spatial overlaps, this does not alter our conclusions. GlaDS continues to reproduce the overall pattern of drainage beneath the FLDIL and also closely matches the murtoo developmental phases from murtoo sedimentology. Instead, Section 5.3 ends by suggesting some future research directions so that GlaDS may better reproduce the spatial pattern of individual murtoo routes.

**Specific comments**
* * *
**Comment 2.2** 14-17: "Our model outputs match many of the predictions" would be a stronger (and appropriate) statement than the rather vague "represent". I think you can say more here about what you've actually found and discussed, and importantly, include that the model matches what we would expect from murtoo sedimentology (as above). For example: "Our model outputs match the general distribution of channelised drainage landforms such as esker and meltwater routes. Many of the predictions for murtoo formation are produced by the model, including the location... and, importantly, the seasonal sequence of drainage conditions inferred from murtoo sedimentology. These conclusions are largely robust to a range of parameter decisions, and we explore seasonal and inter-annual drainage behaviour associated with murtoo zones and meltwater pathways"

**Reply** We thank the reviewer for this suggestion to strengthen our writing in the abstract. We have adopted the spirit of this suggested change, changing the suggested text slightly so that Lines 14–18 now read:

*"Our model outputs closely match the general spacing, direction and complexity of eskers and mapped assemblages of features related to subglacial drainage in 'meltwater routes'. Many of the predictions for murtoo formation are produced by the model, including the location of water pressure equal to ice overburden, the onset of channelised drainage, the transition in drainage modes, and importantly, the seasonal sequence of drainage conditions inferred from murtoo sedimentology. "*
* * *
**Comment 2.3** 32: "analyses... have been applied"

**Reply** We have fixed this as suggested on Line 33.
* * *
**Comment 2.4** 39-40: to better fit the development of ideas in this paragraph, I would move the phrase beginning "potentially including" to line 45, which would then flow "...ideal targets against which to evaluate subglacial hydrology models, potentially including processes variable at sub-annual scales and across the distributed-channelised transition". (Note, wherever it is place, suggest replace "those" with "processes", for clarity.)

**Reply** We have adopted this suggested change, including changing those to processes. This change can be found on Lines 39–46.
* * *
**Comment 2.5** 46: landform genesis uncertainty arises from both fundamentally different concepts of how a landform is formed, and also spatial and temporal scales of formation.

**Reply** We have changed Lines 47–48 as per this suggestion.
* * *
**Comment 2.6** 59: odd punctuation: write out "length and spacing scaling relationships"

**Reply** This was a character padding quirk breaking length/spacing where it should not have. We have spelled out "length and scaling relationships" on Line 59–60 as suggested to correct this.
* * *
**Comment 2.7** 65-67: suggest deleting "modern" from "modern subglacial hydrology models" – unless there is a particular reason for this word, to compare with others?

**Reply** Modern was intended to communicate the difference between process based hydrology models and more simple routing algorithms. We have reworked Lines 61–67 to add a parenthetical definition of 'modern' in this context. The text in question now reads:

"In contrast, modern subglacial hydrology models (i.e., those capable of resolving transitions between distributed and channelised drainage in both space and time) are widely applied to contemporary ice sheets (e.g., Flowers, 2018; Indrigo et al., 2021; Dow et al., 2022; Sommers et al., 2022; Ehrenfeucht et al., 2023). However, despite the critical need to evaluate and improve modern subglacial hydrology models using all available sources of data (Dow, 2023), we are not aware of previous work which has evaluated the ability of such models to reproduce the subglacial conditions (e.g., water pressure, channel location) associated with glaciofluvial landform formation."
* * *
**Comment 2.8** 73: I suggest changing the wording to "and likely represent time-transgressive formation over decades-millennia" – the current wording suggests that there is esker building taking place along the full length of a >10km esker over millennia, which I don't think is what is meant.

**Reply** We have adopted this suggest change, which can now be found on Line 74.
* * *
**Comment 2.9** 74: typo – glaciofluvial

**Reply** Thank you for identifying this typo, which we have corrected.
* * *
**Comment 2.10** 84: published as Peterson et al. 2017

**Reply** We have changed the bibliography reference throughout
* * *
**Comment 2.11** 91: "closer than 40-60km to the..."

**Reply** We have adopted this suggest change, which can be found on Line 91.
* * *
**Comment 2.12** 93-116: in this paragraph, could you add references to the numbered developmental stages in Table 1, at the relevant point in the text? You begin this way (line 97: represents the first stage...), but it would be helpful throughout. E.g. change text in parentheses on line 99 to "developmental stage 2: Table 1", and thereafter at the relevant point just (Stage 3), (Stage 4)...

**Reply** Thank you for this suggestion, which we have adopted throughout the paragraph starting on Line 94.
* * *
**Comment 2.13** 107: does "disappearance" mean erosion or non-deposition?

**Reply** We have added a parenthetical explanation so that Line 109 now reads:

"evidenced by a disappearance of sorted sediment (interpreted as non-deposition rather than erosion)"
* * *
**Comment 2.14** 111-113: there are 3 phrases here beginning "final" or "finally" – suggest rephrasing so that only one event is "final".

**Reply** We have corrected this as suggested
* * *
**Comment 2.15** 132: "however" in this construction is not being used as a parenthetical aside. Suggest change to "...in the centre of the ice lobe where thin sediment cover may have limited..."

**Reply** We have corrected this as suggested, which may now be found on Line 134.
* * *
**Comment 2.16** 158-162: I'm not sure these sentences accurately describe what you present in the results and discussion. In my view, you: examine catchment-scale hydrology parameters and compare to murtoo formation predictions and distribution of channelised landforms (eskers); you specifically explore seasonal and inter-annual drainage parameters in the zone where murtoos are hypothesised to form; you investigate differences in modelled hydrology between observed murtoo and meltwater routes, and where no glaciofluvial landforms exist; and you test the sensitivity of your results to a range of parameters.

**Reply** We have adopted a modified version of this, so that Lines 159–166 now read:

"We examined catchment-scale hydrology outputs and compare these to murtoo formation predictions as well as the distribution of channelised landforms (eskers) in the FLDIL. At individual nodes, we compared the evolution of nodes across our domain against the developmental phases recorded within murtoo sediment excavations (see Table 1, Hovikoski et al., 2023). We then explored seasonal and inter-annual model outputs in the area 40–60 km from the ice margin, at the upglacier limit of channelisation, where murtoos are hypothesised to form (Ahokangas et al., 2021; Ojala et al., 2021). Finally, we go on to investigate differences in model outputs between observed murtoo and meltwater routes, and where no glaciofluvial landforms. We sensitivity tested the robustness of all these findings to a range of parameters"
* * *
**Comment 2.17** 173: suggest delete "uniform" – it seems to contradict being allowed to change diameter

**Reply** We have deleted uniform as suggested
* * *
**Comment 2.18** 221: spell out MAT (mean annual or monthly air temp?)

**Reply** We have corrected this (on Line 225–226) to now read:

"the same depressed monthly average temperature and precipitation record as with the ice sheet model"
* * *
**Comment 2.19** 242-3: "Monthly melt was kept fixed annually" – sounds a bit confusing, do you mean that it was kept fixed year on year, i.e. no inter-annual variability, each month's temp was the same each year?

**Reply** Thank you for identifying this potential source of confusion, and we have corrected this as suggested. Line 246–247 now reads:

"We did not prescribe any inter-annual variability in average monthly temperature."
* * *
**Comment 2.20** 245: again, "total monthly melt was converted to yearly melt rates" – this is also confusing, I'm not sure here what you've done or why

**Reply** We have deleted this turn of phrase. ISSM takes melt input in units of $m.a^{-1}$ and so total monthly melt needs to be converted to a per annum rate value. However, this was also explained on Line 252 already.
* * *
**Comment 2.21** 260-1: while the variables subject to sensitivity testing are clearly listed in the Appendix Table 1, and nicely explained through the following paragraphs, I would have found it helpful to have a clear summary list of those variables in the text. Consider adding a summary sentence here, or modifying line 206-1, along the lines of "Sensitivity testing was performed on: basal melt rate, moulin density and distribution, conductivity of sheet and channelised water, englacial storage, basal ice velocity, land or water-terminating ice, basal bump height, bed topography surface, and mesh geometry."

**Reply** As suggested, we have added the following text to Line 265–266:

"We sensitivity tested for basal melt rate; moulin density and distribution; sheet and channel conductivity terms; basal bump height; the englacial void ratio; basal ice velocity; terminus boundary conditions; bed topography; and mesh geometry. "
* * *
**Comment 2.22** 306: delete "between"

**Reply** We have deleted this as suggested
* * *
**Comment 2.23** 324-5: "and each peaks... and remains..." Here, I also wonder why the peak in sheet discharge and water velocity occurs adjacent to a channel?

**Reply** We have changed peaks to "are highest" on Line 330. These values peak adjacent to channels because water is routed towards channels along hydropotential gradients and because channels are represented along element edges, while water velocity and sheet discharge are represented on element faces. We have not made text changes to clarify this because we feel it is tangential to our work.
* * *
**Comment 2.24** 325-326: typo – $V_W$ not $W_V$ (three instances)

**Reply** Thank you for identifying these typos, which are corrected throughout.
* * *
**Comment 2.25** 326: delete one "to"

**Reply** We have deleted the errant to
* * *
**Comment 2.26** 327: you do have constraints (= landforms) on your model output, so I suggest amending to "Without independent constraint against which to compare our model output"

**Reply** We have adopted this suggestion on Line 333.
* * *
**Comment 2.27** 330: for more emphasis, I would suggest deleting "and", and breaking the sentence. Catchment hydrology "remains consistent across most of the sensitivity tests. Furthermore, sensitivity test results remain consistent with predictions for murtoo genesis."

**Reply** We have adopted this suggestion on Line 335–336.

**Comment 2.28** 331-3: I would delete the additional descriptors for each parameter (e.g. variable, modified, differences in...). It's redundant in the context of sensitivity testing, and risks confusing between an experiment in which a parameter varies spatially/temporally, vs one that has a uniform distribution of the parameter but it differs from the baseline experiment.

**Reply** As suggested we have removed the additional descriptors.
* * *
**Comment 2.29** 334: I got a little confused here and the next paragraph, since you've just stated that almost all parameters showed relative insensitivity but then go on to discuss how several parameters affect channel location. It would help to start a new paragraph on line 334, and open with a clear statement differentiating catchment-scale results with channel-scale results. E.g. "While catchment- scale trends are robust, the exact location of channels, and their length and local $overburden_\%$, vary between sensitivity tests."

**Reply** We have adopted this suggestion, which can now be found on Lines 340–341.
* * *
**Comment 2.30** 336: "differences in channel location"

**Reply** We have adopted this suggestion on Line 343
* * *
**Comment 2.31** 341: again to help the narrative, suggest open with "Besides channel location, channel length and $overburden_\%$ vary considerably". And I would also be specific here about which sensitivity tests, not just "six of". E.g. "...vary considerably in our testing of sensitivity to conductivity".

**Reply** In the spirit of this suggestion, the opening of this paragraph (Line 349–350) now reads:

"Although consistent across the majority of tests, channel length and $overburden_\%$ does vary considerably at the tested limits of $k_s$ and $k_c$ parameters, describing the sheet and channel conductivity respectively"
* * *
**Comment 2.32** 347: again to help the narrative, suggest open with "For the other end member"

**Reply** We have not adopted this suggestion, as we feel end member is unnecessarily vague when here we are discussing both a minimum tested parameter (sheet conductivity) and a maximum tested parameter (channel conductivity).
* * *
**Comment 2.33** 347: does the "and" in this line mean that both conditions are met (minimum sheet k at the same time as maximum channel k), or that this result arises when either one of those conditions is met? I'm guessing the latter since each has a separate figure reference. I wonder if there's a way to clarify this. (Same for line 351.)

**Reply** It is the latter, as we only tested one individual parameters at a time. We have changed Line 355 to now read:

"For *both* the minimum sheet conductivity...and the maximum channel conductivity"

We have also made the same change above, on Line 351.

.
* * *
**Comment 2.34** 357-60: I think a clearer way to argue this is that excessively long or short channels are considered 'invalid', on the basis of modern Greenland observations, and that an anomalous overburdendistribution is considered invalid on the basis of the conceptual model for murtoo distribution, and therefore the baseline terms are considered most plausible.

**Reply** We have adopted the suggested wording, and Line 365–369 now reads:

"Excessively long (>50 km) or short (<10 km) channels are considered to be invalid on the basis of modern Greenland observations (e.g., Chandler et al., 2013; Dow et al., 2015) and an anomalous $overburden_\%$ is considered invalid on the basis of the conceptual model for murtoo distribution and genesis (e.g., Ahokangas et al., 2021; Hovikoski et al., 2023). Therefore, our baseline conductivity terms are the considered the most plausible parameters."
* * *
**Comment 2.35** 366: "the pressure conditions... are notably different"

**Reply** We have corrected this, now found on Line 375.
* * *
**Comment 2.36** 367: suggest breaking the sentence to make it clear which channels (modelled or observed) you refer to in the final clause. "In Greenland, channels exist at lower pressure..., and the resultant hydraulic potential gradient..."

**Reply** We have adopted the suggested wording, now found on Line 376–379
* * *
**Comment 2.37** 376: typo – FLDIL

**Reply** We have corrected this, now found on Line 385.
* * *
**Comment 2.38** Section 5.2: From your description of sites D& E, I struggle to see the difference in landform/hydrological context. You start with E, and describe it as being representative of channel onset nearby, and conclude that the model drainage behaviour is consistent with murtoo development phases (although murtoos themselves aren't evident here). Then you describe D as being located at the head of two channels, and near murtoos, and interpret that 1 of the 2 modes of drainage here is consistent with murtoo formation. I see that the drainage behaviour differs at sites D& E, but both can allow for murtoo formation and both have some relationship with channel onset, so I think this part of section 5.2 (e.g. lines ~389-406) would be better packaged if you instead: introduce D& E as both being in the zone hypothesised to favour murtoo formation, and that both display drainage behaviour that could accommodate murtoo formation, but they show different interannual behaviour; and note also that one node is close to an observed murtoo while the other is not. I think this would better frame the later discussion of absence of murtoos where they're predicted, and the discussion of the biannual behaviour and its relationship to landform distribution. Here and below: a figure, perhaps plotted in map view, of nodes that display biannual behaviour vs annual behaviour would really help this discussion and that in section 5.3.

**Reply** Thank you for this paragraph suggestion, which we have adopted. Lines 399–409 now read:

"Figure 3D & E demonstrates the seasonal evolution of two nodes between 40–60 km from the ice margin, each nearby to channel systems. Both nodes fall within the hypothesised zone of murtoo formation and both nodes display drainage behaviour which could accommodate murtoo formation, with a seasonal increase in $overburden_\%$ up to a maximum of approximately 120 % and a more gradual decrease thereafter. However, the two nodes show different interannual behaviour, and only one is located close to a murtoo field. At node 3,842, ~54 km from the ice margin and chosen to be representative of surrounding nodes at the onset of a channel (Figure 3E), the pattern of drainage repeats annually—every year the increase and decrease in $overburden_\%$ is accompanied by peaks in $q_s$, $Q_c$, and $V_W$ and the nearby development of channels throughout

the meltwater season. At the onset of channelisation the maximum $Q_c$ approaches but never exceeds 1 $1 m^3 s^{-1}$ . However, although this evolution through time does appear consistent with each of the murtoo developmental phases (Table 1), node 3,842 is not located near to a murtoo field .

At node 16,402, located..."

However, we have not created an additional figure as suggested, given the practical difficulties associated with identifying which of several thousand nodes do or do not display biannual behaviour.
* * *
**Comment 2.39** 392: wording is a bit awkward – I suggest "representative of nodes surrounding the onset of a channel" ?

**Reply** This text was removed as part of the suggested rewrite above.
* * *
**Comment 2.40** 402-5: there's a few awkward phrases here. Consider: "before quickly dropping to an $overburden_\%$ that is elevated relative to the previous winter." "Years with an elevated $overburden_\%$ are associated with lower Qc...". Put commas round "approaching 1 m3 s-1". Replace odd-numbered years with "We consider that the latter case is more consistent with..."

**Reply** We have adopted the suggested wording, now found throughout the paragraph starting on Line 409.
* * *
**Comment 2.41** 410: double "maximum"

**Reply** We have corrected this.
* * *
**Comment 2.42** 411: "the channel remains active over winter".

**Reply** We have amended this text to now end in "over winter". New text now found on Line 423.
* * *
**Comment 2.43** 418-9: suggest you add to the end of the sentence "that have been invoked to explain murtoo sedimentology".

**Reply** We have added the suggested text on Line 431.
* * *
**Comment 2.44** 422: since GLaDS is pervasively connected and therefore can't produce the rapid changes in flow that you describe, is this limitation more widely a problem for other modelling questions? Of course the model is a necessary compromise on certain aspects in order to be manageable, but does your work flag that this is actually important to implement more fully?

**Reply** This is a limitation shared by many recent subglacial drainage models (see Rada Giacaman and Schoof, 2023) and we now state this on Line 434, which reads:

"However, as with other subglacial hydrology models, GlaDS is a model in which the subglacial system is assumed to be pervasively hydraulically connected (see Rada Giacaman and Schoof, 2023)"

Further, we do agree that in order to more faithfully reproduce murtoo formation it is likely that hydraulic isolation is something that is important to implement in future studies. We have included a sentence to this effect on Lines 439–441 which reads:

"Including spatially variable system conductivity is likely to be important in future work which seeks to evaluate the ability of process-based subglacial hydrology models to represent landform formation. "

Reference: Rada Giacaman, C. A. and Schoof, C.: Channelized, distributed, and disconnected: spatial structure and temporal evolution of the subglacial drainage under a valley glacier in the Yukon, The Cryosphere, 17, 761–787, `https://doi.org/10.5194/tc-17-761-2023`, 2023.

**Comment 2.45** Section 5.3 Is the heading appropriate? I think something like "Comparing murtoo and meltwater route hydrology" is a closer description of what this Section considers. Paragraph 1 is overcomplicated, and I think it is mis-framed as an evaluation of GLaDS ability to represent meltwater pathways – you do something more specific than that and it raises an interesting discussion about different types of behaviour. It could be much more succinct and clear if you frame the opening to this Section as: You further explore drainage behaviour in the zone hypothesised to be relevant for murtoo formation. You group all the nodes in this zone according to whether they are located among murtoos, meltwater routes, or neither. Murtoos and meltwater route distribution are based on Ahokangas et al, and you include eskers with meltwater routes, though you acknowledge there is no age control to say if they all form simultaneously (and in the relevant time-slice).

**Reply** We have adopted the suggested subsection heading and have changed the framing sentence so that Line 450–451 now reads:

"We explore drainage behaviour in the area of anticipated murtoo formation by isolating and taking a spatial median of nodes in the baseline model 40–60 km from the ice margin"

**Comment 2.46** Line 448: "winter minima"

**Reply** Amended on Line 468.

**Comment 2.47** 449: "do not intersect mapped glaciofluvial geomorphology"

**Reply** Amended on Line 472.

**Comment 2.48** 450: "...are lower..."

**Reply** Amended on Line 470.

**Comment 2.49** P3 [in section 5.3] could be more impactful, and it's missing a clear discussion of all elements in Fig 6 – in the second half of the paragraph, you overlook Fig 6 and only talk about what's in the Appendix. I suggest, for all parameters, first comment on what is shown in Fig 6, then comment on what differences are apparent during specific months.

I also think there's some interesting behaviour in Fig 6 that you haven't commented on (but here would be the place for it), and I wonder if it would contribute to your discussion of biannual behaviour and/or differences in drainage mode within the same 'murtoo-favouring zone'. For all parameters, the murtoo data has a bimodal distribution, while the channels either have a single peak or are also bimodal – is this something you can comment on?

**Reply** As suggested, we have commented more on the PDF, and the next text (Line 474–480) now reads:

"The probability density functions of murtoo routes and meltwater routes is also clearly distinct from terrain without glaciofluvial landforms (Figure 6). However, the probability density functions of murtoo routes and meltwater routes are also different from one another, particularly at the lower tail of the distributions (Figure 6). Murtoo routes have a $overburden_\%$ distribution with a more tightly constrained lower tail than meltwater routes, with fewer nodes dropping below $overburden_\% = 80\%$. There is a bimodal distribution of both $q_s$ and $V_W$ within murtoo routes that is not evident in meltwater routes at the lower tail of the distribution.

Both meltwater routes and murtoo routes have a bimodal $Q_c$ distribution, but the lower murtoo route peak is offset towards higher channel discharges"
* * *
**Comment 2.50** P4-P6 [in section 5.3] – I still find it hard to work through the arguments relating to the biannual drainage signal. It's an interesting behaviour, but I'm still not really sure what you are saying about it in relation to either space or landforms (or whether it's a model artefact that wouldn't be present in reality). I think there's a missing link in your arguments that relates to the statements on lines 472-3 and 481-2: that there is a spatial component to the biannual signal in murtoo route outputs, and a significant biannual difference between murtoo routes and meltwater routes.

I don't believe you've demonstrated a significant biannual difference between murtoo routes and meltwater routes. Section 5.2/Fig 3 show that two nodes in the hypothesised murtoo forming zone have different signals – one annual, and one biannual. You've concluded that both behaviours are suited to murtoo formation (albeit only one of the two modes in the biannual case). Section 5.3/Fig 6 show that murtoo and melt routes differ in behaviour within a season/year, but you don't demonstrate here that one or the other is characterised by a biannual signal. Therefore, your statements on lines 472-3 and 481-2 combine these two results, without demonstrating that the biannual signal is landform-specific.

I think paragraphs 4-6 (i.e. remainder of Section 5.3, from line 467), ought to be revised to sharpen the discussion about the biannual signal, its spatial distribution, and how it might relate to a specific landform type. As above, I think that a figure that demonstrates the "spatial component to the biannual signal" would really help – whether or not this spatial biannual pattern also has a connection to a specific landform type. This could be a map – nodes with an annual/biannual signal. And/or an equivalent to Fig 6, but plotted for the two lateral or central parts of the lobe instead of by landform type?

**Reply** In seeking to address this comment, and several of those below, we have restructured the end of Section 5.3 from Line 491 onwards. We have trimmed and simplified much of the text with a view to highlighting:

- That there is a statistically significant difference between meltwater routes and murtoo routes (Lines 475–491).

- That this difference is subtle, and both murtoo routes and meltwater routes are consistent with the murtoo formation phases (Lines 492–493).

- We conclude that this statistical difference has a strong spatial component linked to the repetitive presence of overwinter channels, GlaDS is not necessarily resolving differences between the landforms themselves (Lines 493–503).

- The existence of overwinter channels is not necessarily surprising, but their repetitive signal is likely driven by our fixed model forcing. Their spatial expression meanwhile may be linked to the diverging lobe and ice flow vectors (Lines 504–509).

- Winter channels, are certainly less likely to be repetitive in a more realistic model setup, and may well disappear altogether (Lines 509–512).

- Murtoos distribution appears to be a compelx mix of factors, and in reproducing murtoo forming conditions in the centre of the ice lobe (where murtoos are absent 40–60 km from the ice margin) our model is clearly missing key subglacial process (e.g., ice coupling, sediment dynamics) and future work is needed (Lines 512-519).
* * *
**Comment 2.51** In P4 (in section 5.3) , there are far too many ideas all stuffed into one paragraph: I would move the opening to conclude P2 and open P3: murtoo & meltwater routes are both very different to where there are no landforms, and in this way GLaDS faithfully represents FLDIL drainage patterns. This is further evident when plotted as a PDF. Here, however, you also see subtle differences between murtoos and meltwater routes. - Start P4 by posing the problem that murtoos are absent where the model (both

conceptual and physical) suggests they ought to form. Offer your geological/data reasons for this. Then...
- Alternatively, murtoo distribution could be related to biannual channel discharge behaviour (reported in section 5.2, Fig 3). Biannual signal is interpreted as due to channels persisting through winter; they likely influence the nearby system the following summer, when the initial melt input would be discharged by an already established efficient pathway. There is also a spatial component to the biannual signal in our model output. When channels in the central third of the FLDIL persist over winter, those in the outer two thirds do not – and vice versa. Murtoo distribution – absent in the central third – could be a reflection of this spatial control on winter channel operation.

This sets up the whole of the final part of your discussion in the context of murtoo formation (presence/absence), and there is a clear purpose to the biannual discussion, rather than discussing a quirk that seems to have emerged from your modelling whose connection to your research question isn't really apparent.

**Reply** As suggested, we have moved the opening of P4 to conclude P2 and state that GlaDS is faithfully representing FLDIL drainage patterns. However, because we cannot rule out that overwinter channels (which drive the statistically significant difference) are not in fact a model artefact, we have stopped short of linking murtoo formation to winter channels. Instead, in rewriting the end of section 5.3, we have tried to make it more relevant to the paper's aims (how well does a process-based model resolve murtoo formation). To do so, we have reframed this section as one in which we state that GlaDS is clearly distinguishing between meltwater/murtoo routes and the wider terrain, but is failing to resolve differences between murtoo and meltwater routes, and is therefore not able to resolve specific murtoo locations. We go on to suggest future work which may further improve GlaDS ability to represent murtoo formation.
* * *
**Comment 2.52** P5 – I also think this is unnecessarily wordy and buries the point of the argument. Consider re- ordering (and trimming) the ideas: The appearance of winter channels isn't surprising: they are evident on Greenland. Yours operate at very low discharges (below an arbitrary threshold for classifying a "channel") but nonetheless exhibit this behaviour. However, the spatial pattern of winter persistence is unexpected. There is spatial variability in your climate forcing, which translates into spatial variability in meltwater input, though your input has no interannual variability that would explain a biannual signal in channel Q and overburden.

This leads directly into the next paragraph, in which you offer a solution.

**Reply** We have adopted these ideas in our rewrite of Section 5.3, and have used this structure in our new writing (Lines 504–509)
* * *
**Comment 2.53** Line 503 – if this difference between murtoo and meltwater routes that you refer to here is the presence of a biannual signal, then as above, I don't think you have demonstrated this. (If you only mean that they differ, as in Fig 6, then this is ok.)

**Reply** As part of our rewrite in this section to more clearly communicate the significant differences reported on, we have removed this particular text.
* * *
**Comment 2.54** 504: "divergence... appears to..."

**Reply** We have removed this particular text
* * *
**Comment 2.55** 506: be specific – which portions of the model have channels that resist closure during winter? A figure would really help this spatial discussion. Where/how do these portions relate to flow divergence?

**Reply** To address this, we include a new supplementary figure (Figure A2) which illustrates where winter channels form each year and how this relates to murtoo distribution

**Comment 2.56** 508-9: but even if the very regular biannual signal isn't evident any more, do certain sectors have more of a tendency for over-winter channel persistence than others? I'm not sure you intended to conclude this discussion with 'it's a model artefact', but that's how it reads.

**Reply** There is no clear difference between sectors. In our rewrite, we have elected to be more cautious because we ultimately cannot rule out that it is in fact a model artefact, we have removed this specific line as per the above comments.
* * *
**Comment 2.57** 509-514: this is a really interesting idea, that the sub-lobes approximately match the pattern of winter channel persistence. Again, could a figure that illustrates this be combined with one as suggested above, or would the same figure serve both aspects of this discussion? And can you offer a concluding sentence that might explain why large eskers bound zones with different winter persistence of channels? In terms of the writing structure, I would break these lines off into a new, final paragraph.

**Reply** The original text was intended to highlight the close overlap between overwinter channels, murtoo routes, and sub-lobes, without being overly speculative. However, because we cannot rule out that the overwinter channels are in fact a model artefact, we have removed this particular section of text.
* * *
**Comment 2.58** 517: I think you've actually explored drainage in relation to more than one glaciofluvial landform – suggest pluralise

**Reply** We have pluralised as suggested (Now on Line 522).
* * *
**Comment 2.59** 520: the absence of topography is an awkward concept. Suggest amend to "including the absence of any relief"

' **Reply** as suggested, we have amended this to "a total absence of relief" (Line 526).
* * *
**Comment 2.60** 523: the wording here is a bit confusing about the direction/trend of change (up/down or forward/backward in time) – suggest something like "Assuming this area has been uplifted by a maximum of $\sim$100m, the volume of melt... would have been higher during the YD due to higher temperatures at lower altitude."

**Reply** We have adopted the suggested wording, which can now be found on Line 528–531.
* * *
**Comment 2.61** 537-539: this seems very specific and a poor final sentence – is it necessary or can you move it earlier in the paragraph? It seems more natural to end with the future work sentence.

**Reply** As suggested, we have moved the sentence. It can now be found on Line 534–536.
* * *
**Comment 2.62** In this section you focus on all the things that you haven't included. But you have done some fairly extensive sensitivity testing. I think it would be entirely appropriate to note that you haven't included all these factors and this introduces some uncertainty, but that you have tested many parameters and your findings are largely robust.

**Reply** We have not made any change in response to this, as we do go on to state this in the summary and conclusion section, where we feel it is most appropriate.
* * *
**Comment 2.63** 544: "The alternating sedimentological sequence"

**Reply** We have adopted the suggested wording, which can now be found on Line 549.
* * *
**Comment 2.64** 557: delete "extending", you've already said the channels extend

**Reply** We have removed this "extending".
* * *
**Comment 2.65** Make sure the title of the article matches the final title.

**Reply** Thank you for noticing this, which we have now fixed.
* * *
**Comment 2.66** Fig A15. I think the caption ought to read 'comparison of basal melt $7 \times 10^{-3}$... against baseline $1 \times 10^{-3}$ ?

**Reply** Thank you for noticing this in what is now Fig A16, it actually should read $5 \times 10^{-3}$ and we have amended both this and Fig A15.